# Serum coating enables feeder-free culture of naive human pluripotent stem cells preserving developmental potential

Giada Rossignoli [1,16], Michael Oberhuemer [2,3,16], Ida Sophie Brun [4,16], Irene Zorzan [1,15,16], Anna Osnato[5,6], Anne Wenzel[4], Emiel van Genderen[7], Andrea Drusin[1], Giorgia Panebianco [1], Nicolò Magri [1], Moritz Becker[2,3], Mairim Alexandra Solis[5,8], Chiara Colantuono[9], Sam Samuël Franciscus Allegonda van Knippenberg[5,6], Thi Xuan Ai Pham[5,6], Sherif Khodeer [5,6], Paolo Grumati [10,11], Davide Cacchiarelli [10,12,13], Paolo Martini [14], Nicolas Rivron[7], Vincent Pasque [5,6], Jan Jakub Żylicz [4✉], Martin Leeb [2✉] & Graziano Martello [1✉]

## Abstract

**Naive human pluripotent stem cells (hPSCs) represent a pre-implantation epiblast state able to efficiently differentiate into embryonic and extraembryonic pre-implantation lineages and to self-organise in vitro into blastocyst-like structures called blastoids. Naive hPSC maintenance routinely relies on co-culture with mouse embryonic fibroblast (MEFs) as feeder cells, a method prone to variability and analytical confounders. Here, we describe a feeder-free culture system based on serum coating that supports long-term maintenance of naive hPSCs. Across five laboratories, 30 serum batches were evaluated for the expansion of eight naive hPSCs lines for up to 25 passages. Mass spectrometry analysis identified fibronectin and collagens as extracellular matrix proteins consistently present in serum coating. Cells cultured on serum coating displayed growth kinetics, clonogenic capacity, mutation rates, and global gene expression profiles comparable to MEF-based cultures. Importantly, serum-cultured naive hPSCs efficiently underwent germ layer specification, retained trophectoderm competence, and generated blastoids with efficiency similar to MEF-based cultures. Collectively, serum coating provides a scalable, cost-effective, and robust alternative to feeder-based systems, preserving genomic stability and developmental potential while eliminating MEF-associated disadvantages and variability. This platform facilitates large-scale applications of naive hPSCs and enables more reproducible mechanistic studies.**

**Keywords** Human Naive Pluripotent Stem Cells; Blastoids; Extraembryonic Lineages; Extracellular Matrix; Feeder-free
**Subject Categories** Cell Adhesion, Polarity & Cytoskeleton; Methods & Resources; Stem Cells & Regenerative Medicine

See also: M Yagi & K Hochedlinger

## Introduction

The derivation of human pluripotent stem cells (hPSCs) has transformed the stem cell field. Their remarkable capacity to self-renew and to differentiate into all embryonic specialised cell types has generated substantial enthusiasm for their use in biological and medical research. hPSCs were first derived from the inner cell mass (ICM) of in vitro fertilised embryos (Human Embryonic Stem Cells, hESCs), and cultured in serum-based media on a feeder layer of inactivated mouse embryonic fibroblasts (MEFs) (Thomson et al, 1998), using methods developed for mouse ESCs (mESCs) (Evans and Kaufman, 1981). In addition, since the first reprogramming of human induced pluripotent stem cells (hiPSCs) from somatic cells in 2007, by overexpressing a cocktail of transcription factors (OCT4, SOX2, KLF4 and c-MYC from Takahashi et al, 2007, and OCT4, SOX2, NANOG

[1]Department of Biology, University of Padua, Padua, Italy. [2]Max Perutz Labs, Vienna Biocenter Campus, University of Vienna, Vienna, Austria. [3]Vienna BioCenter PhD Program, Doctoral School of the University of Vienna, Medical University of Vienna, Vienna, Austria. [4]Novo Nordisk Foundation Center for Stem Cell Medicine - reNEW, Department of Biomedical Sciences, Faculty of Health and Medical Science, University of Copenhagen, Copenhagen, Denmark. [5]KU Leuven-University of Leuven, Department of Development and Regeneration, Leuven Stem Cell Institute, Leuven, Belgium. [6]KU Leuven Institute for Single Cell Omics (LISCO), Leuven, Belgium. [7]Institute of Molecular Biotechnology of the Austrian Academy of Sciences (IMBA), Vienna BioCenter, Vienna, Austria. [8]Gorgas Memorial Institute for Health Studies, Panama, Panama. [9]NEGEDIA S.r.l., Pozzuoli, Italy. [10]Telethon Institute of Genetics and Medicine (TIGEM), Pozzuoli, Italy. [11]Department of Clinical Medicine and Surgery, University of Naples "Federico II", Naples, Italy. [12]Department of Translational Medicine, University of Naples "Federico II", Naples, Italy. [13]Genomics and Experimental Medicine Program, Scuola Superiore Meridionale (SSM, School of Advanced Studies), Naples, Italy. [14]Department of Molecular and Translational Medicine, University of Brescia, Brescia, Italy. [15]Present address: Epigenetics Programme, Babraham Institute, Cambridge, UK. [16]These authors contributed equally: Giada Rossignoli, Michael Oberhuemer, Ida Sophie Brun, Irene Zorzan. ✉E-mail: jan.zylicz@sund.ku.dk; martin.leeb@univie.ac.at; graziano.martello@unipd.it

and LIN28 from Yu et al, 2007), hundreds of lines, reflecting the high genetic diversity of humans, have been derived.

However, capturing PSCs homogeneously reflecting specific developmental stages and in chemically-defined conditions remains an ongoing challenge. Undifferentiated mESCs were initially maintained in feeder-free conditions in the presence of leukaemia inhibitory factor (LIF) and foetal bovine serum (FBS) on gelatin-coated plates (Smith et al, 1988; Williams et al, 1988). This culture regime captures heterogeneous populations reflecting the pre- and post-implantation stages (Ying et al, 2008; Marks et al, 2012; Kolodziejczyk et al, 2015). Subsequently, chemically-defined protocols were established that capture relatively homogeneous populations reflecting either the pre-implantation (blastocyst stage) or post-implantation (pre-gastrulation stage) state of the epiblast. On the one hand, two inhibitors (2i), the MEK inhibitor PD0325901 and the GSK3 inhibitor CHIR99021, were shown to promote efficient mESC self-renewal that more homogeneously reflects the pre-implantation stage cells—a so-called naive state—and can be cultured in the absence of feeders (Ying et al, 2008; Martello and Smith, 2014). This combination of 2i and the cytokine LIF (2iL) resulted in more robust proliferation, increased oxidative phosphorylation and genome hypomethylation (Dunn et al, 2014; Carbognin et al, 2016; Betto et al, 2021). On the other hand, FGF and Activin A were shown to maintain efficient self-renewal of mEpiSCs that reflect the post-implantation stage cells, a so-called primed state, when cultured on fibronectin coatings (Brons et al, 2007; Tesar et al, 2007; Carbognin et al, 2023).

Conventional hPSCs cultured without feeders on Matrigel or vitronectin-coated plates in Essential 8 (E8) or mTeSR media including FGF2 and TGFβ (Ludwig et al, 2006; Braam et al, 2008; Chen et al, 2011) are in a primed pluripotent state more akin to the post-implantation epiblast. This state is highly similar to the one of mEpiSCs in terms of growth factor dependence, transcriptional and epigenetic regulation (Buecker et al, 2010; Chan et al, 2013; Gafni et al, 2013; Yan et al, 2013). The ability to convert mEpiSCs into mESCs through culture conditions or transient overexpression of naive-specific TFs such as Klf4 provided the paradigm for the derivation of naive hPSCs (Guo et al, 2009). Surprisingly, hPSCs were found not responsive to 2iL (Hanna et al, 2010), leading different research groups to first derive naive hPSCs in 2iL by overexpressing transcription factors associated with naive pluripotency (Hanna et al, 2010; Buecker et al, 2010). Later on, more specific culture conditions and genetic manipulations were implemented (Chan et al, 2013; Gafni et al, 2013; Takashima et al, 2014; Theunissen et al, 2014). Since then, naive hPSCs have been successfully derived from human blastocysts and somatic cell reprogramming by transient gene overexpression or chemical resetting from primed hPSCs in various media (Guo et al, 2016; Liu et al, 2017; Kilens et al, 2018; Wang et al, 2018; Giulitti et al, 2019). Successful maintenance of human naive pluripotency in vitro in a transgene-independent manner and serum-free medium was initially achieved by culture optimisation starting from 2iL conditions (Takashima et al, 2014; Theunissen et al, 2014). The addition of Gö6983, a protein kinase C (PKC) inhibitor, previously shown to also suppress mESC differentiation (Dutta et al, 2011), combined with a lower concentration of CHIR99021 (t2iLGö), was beneficial in maintaining compact colonies with morphology and proliferation of naive hPSCs (Takashima et al, 2014). Subsequently, GSK3 inhibition was shown to be dispensable for the maintenance

and resetting of naive hPSCs, leading to the replacement of CHIR99021 with the tankyrase inhibitor XAV939 (PXGL) to achieve more robust naive cultures (Bredenkamp et al, 2019). A high-throughput chemical screen identified a combination of compounds, including the alternative GSK3 inhibitor IM12, the BRAF inhibitor SB590885, the SRC inhibitor WH-4-023, and the ROCK inhibitor Y-27632, supplemented with FGF and Activin A that synergise with PD0325901 and LIF to support the expansion of naive hPSCs (5i/L/AF) (Theunissen et al, 2014). The addition of the JNK inhibitor SP600125 (6i/L/A) has been reported to increase the efficiency of naive hPSCs induction from the primed state (Theunissen et al, 2014). In addition, a defined condition termed NHSM (naive human stem cell medium) was developed, in which 2iLGö was supplemented with the p38 inhibitor SB203580 or BIRB796, the JNK inhibitor SP600125, ROCK inhibitor Y-27632, bFGF and TGF-β1 (Gafni et al, 2013; Bayerl et al, 2021). The derivation of naive hPSCs has broadened the potential applications of stem cells. Consistent with the fact that their transcriptomic state reflects a developmental state similar to the blastocyst stage epiblast rather than post-implantation epiblast, naive hPSCs exhibit higher expression of pluripotency genes such as *NANOG*, *KLF4*, *KLF17*, *DPPA3*, and *DPPA5* when compared to the human primed post-implantation epiblast (Yan et al, 2013) and are characterised by global genomic hypomethylation compared to primed cells (Theunissen et al, 2016), like mESCs (Betto et al, 2021), and biallelic expression of multiple imprinted genes (Pastor et al, 2016; Bar et al, 2017; Martini et al, 2022). Naive hPSCs also use a bivalent metabolic system with a greater reliance on oxidative metabolism, whereas primed hPSCs are almost exclusively glycolytic (Takashima et al, 2014; Theunissen et al, 2014; Gu et al, 2016; Zhou et al, 2012), as reported for mouse naive and primed PSCs, respectively (Carbognin et al, 2016). Importantly, this naive state resembles a preimplantation-stage - and thus broader - differentiation potential compared to primed hPSCs, enabling efficient differentiation into extraembryonic tissues, including trophectoderm (TE) (Castel et al, 2020; Dong et al, 2020; Guo et al, 2021; Io et al, 2021b), primitive endoderm (PrE) (Linneberg-Agerholm et al, 2019; Okubo et al, 2024), and extraembryonic mesoderm (Pham et al, 2022), three lineages previously inaccessible for in vitro studies. Furthermore, naive hPSCs can aggregate and generate 3D blastocyst-like structures - so-called blastoids - composed of the three founding lineages as a model to study human implantation and peri-implantation development (Yanagida et al, 2021; Kagawa et al, 2022; Yu et al, 2021; Proks et al, 2025; Zhao et al, 2025).

A major hurdle for studying the human naive state is that, unlike the mouse naive and human primed states, it is still routinely cultured on MEFs. In addition to the experimental cost and time involved, the quality of culture on feeders has been shown to depend on numerous factors such as embryo age, passage number of MEFs and their genetic background. All of these affect MEF features and thereby, hPSCs proliferation, colony formation potential and cell state (Xie et al, 2004; Schnabel et al, 2012; Azizi et al, 2019; Choupani et al, 2022). The exchange of signals between feeders and hPSCs, some of which are still unknown, also prevents a stringent control of the experimental system and makes it more difficult to define ideal growth conditions and underlying biological pathways necessary for maintaining pluripotency in vitro, with significant implications for downstream analyses and applications of these cells. There have been extensive efforts to eliminate feeders

from naive hPSC cultures. However, to this date, they have not been widely adopted for multiple reasons. A 3D culture system based on Matrigel allows for a robust expansion of naive hPSCs in PXGL (Cesare et al, 2022). While it allows for efficient 3D differentiation, it is time- and resource-intensive as a routine culture method. Reduced proliferation was reported if MEFs were replaced with Matrigel or laminin-511 coating in t2iLGö medium in 2D (Takashima et al, 2014). Media formulations with extensive addition of inhibitors allowed for robust expansion of hPSCs on Matrigel or Vitronectin-coated plates (FINE, NHSM and RSeT, a commercial medium based on NHSM) (Gafni et al, 2013; Szczerbinska et al, 2019). However, these inhibitors affect the morphology (Szczerbinska et al, 2019) or stabilise a transcriptional state of hPSCs distinct from naive cells routinely used in blastoid formation (Gafni et al, 2013; Liu et al, 2017). Consequently, the routinely used conditions for naive hPSCs expansion and blastoid generation all rely on the use of MEFs (Yu et al, 2021; Yanagida et al, 2021; Kagawa et al, 2022).

The addition of serum to cell culture media has been used to improve cell adhesion since the first half of the last century (Puck et al, 1958; Holmes, 1967; Hayman et al, 1985). Analysis of serum composition led to the discovery of the extracellular matrix (ECM) proteins vitronectin (Hayman et al, 1983) and fibronectin (Hayman and Ruoslahti, 1979). Murine ESCs and EpiSCs can be expanded efficiently on FBS-coated plates (Brons et al, 2007; Murray et al, 2013). Moreover, while mESCs strongly attach when converted from 2iL to serum/LIF, cells grown in serum-containing media frequently exhibit attachment problems when converted to serum-free 2iL. In the latter case, the addition of a small amount of FBS in 2iL generally enhances cell adhesion (Balbasi et al, 2022). In addition, primed hPSCs have been successfully grown on serum-coated substrates without feeders (Vallier et al, 2005).

In this study, we observed, across 5 laboratories, that naive hPSCs of both induced and embryonic origins can be easily adapted to a feeder-free, serum-based coating in PXGL medium. Cells grown on serum-coated culture dishes retain a naive pluripotency gene expression signature, and a proliferation rate and clonogenic capacity similar to the original lines on MEFs, without acquiring pathogen-associated mutations. In addition, naive hPSCs cultured on serum coating retain full differentiation potential towards both extra-embryonic and embryonic tissues as well as the capacity to self-assemble into blastoids.

# Results

## Naive hPSCs spontaneously adapt to serum coating

Naive hiPSCs directly reprogrammed from somatic cells (Giulitti et al, 2019) (HPD06 and HPD03) were plated in PXGL medium on dishes coated with 10% FBS diluted in DMEM (hereafter referred to as serum coating) without MEFs. Naive hiPSCSs retained their distinctive dome-shaped morphology (Fig. 1A) and expression of general pluripotency (POU5F1/OCT4 and NANOG) and naive-specific (KLF17, TFCP2L1, and KLF4) markers at comparable or even higher levels relative to cells cultured on MEFs, as well as low or undetectable expression of primed genes (Figs. 1B and EV1A–C).

Prompted by these results, we decided to test extensively whether serum coating could be used for naive hPSC lines of different origin (i.e. reprogramming of fibroblasts, resetting of primed PSCs and embryo-derived), of both sexes and in different laboratories. Furthermore, we reasoned that an extensive test of the differentiation potential should be performed, including both embryonic and extraembryonic lineages, to test if serum coating allowed for the maintenance of bona fide functional naive hPSCs (Fig. 1C). We successfully converted 8 naive hPSC lines in 5 different laboratories to feeder-free conditions (Table 1). Converted lines were characterised by dome-shaped morphology (Fig. 1A), comparable expression of general (OCT4 and NANOG) and naive pluripotency markers (KLF17 and SUSD2) (Figs. 1B–D and EV1B–D), and robust expansion. All cell lines were maintained under these conditions for at least ten passages (see Table 2 for detailed information on each cell line and experiments). In total, we tested over 30 different batches of FBS from 7 different suppliers, and all but one serum batch allowed expansion of naive PSCs (Table 3, see Methods for details of the batch testing).

We have not experienced any cell line failing to convert to serum coating in PXGL. Only two lines (HPD06 and HPD03), which were plated at the same density as the maintenance of MEFs for starting the conversion, exhibited a transient decrease in proliferation and clonogenicity (Fig. EV1E,F), which were recovered to levels comparable to the same cell line cultured on MEFs by passage 8 (Fig. 1E,F). All lines subsequently were plated at higher density during the first couple of passages of the adaptation to the feeder-free condition, which occurred swiftly.

Converted naive hPSCs could be efficiently transfected, leading to stable expression of a transgene from a transposon-based vector (Fig. EV1G).

We conclude that serum coating allows for the maintenance of naive pluripotency markers and the long-term expansion of naive hiPSCs, without application of additional inhibitors or genetic manipulation.

## Proteomics analysis of serum composition reveals abundant ECM proteins

We wondered how serum coating could sustain the attachment and growth of naive hPSCs. Therefore, we assessed the composition of the coating through a label-free quantitative proteomics analysis on 5 batches extensively used in this study (Tables 2 and 3). The entire list of identified proteins was screened and annotated for matrisome and matrisome-associated macromolecules using the Matrisome AnalyseR tool from The Matrisome Project (Petrov et al, 2023). On average, we detected 672 proteins in each serum batch. 451 proteins were detected in all 5 serum batches, of which 92 (20.4%) were classified as "Core matrisome" or "Matrisome-associated" hits. ECM glycoproteins and ECM regulators accounted for the majority of the hits in these two categories (Fig. 2A). Among the ECM proteins detected in all 5 batches, we detected several Collagens and Fibronectin (FN1) (Fig. 2B), which have been previously implicated in the expansion of hPSCs (Cesare et al, 2022; Kitajima and Niwa, 2010; Kim and Kino-oka, 2014). Of note, these ECM proteins were among the top 200 most abundant proteins in all analysed batches (Fig. EV2A).

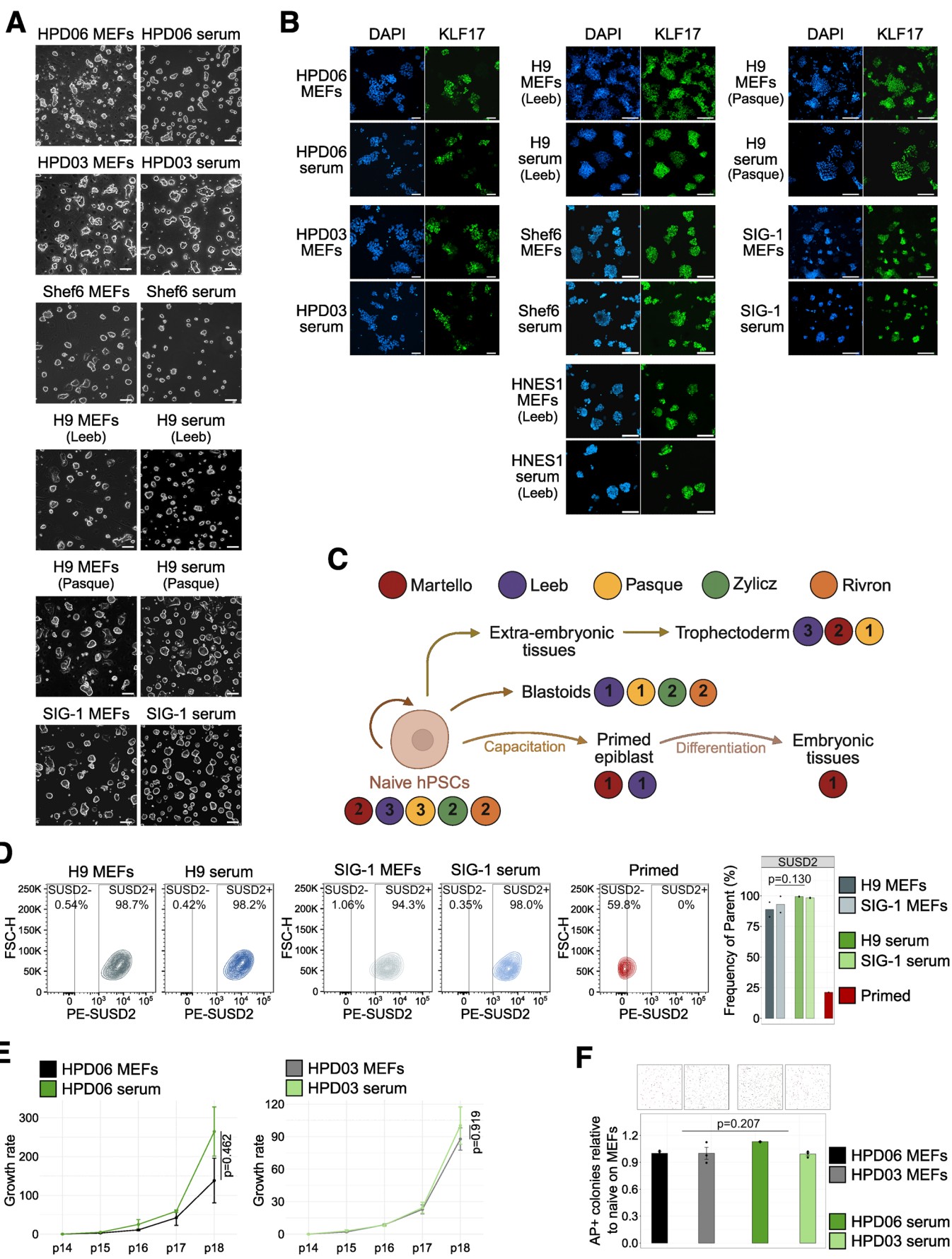

**Figure 1. Serum coating allows for the maintenance of naive hPSCs.**

(A) Morphologies of different naive hPSC lines (HPD06, HPD03 and SIG-1 iPSCs, and H9 and Shef6 ESCs) stably cultured on MEFs or serum coating. Scale bars: 100 μm. Representative images of two independent experiments are shown. (B) Immunostaining for KLF17 of different naive hPSC lines (HPD06, HPD03 and SIG-1 iPSCs, and H9, HNES1 and Shef6 ESCs) stably cultured on MEFs or serum coating. See also Fig. EV1B,C. Scale bars: 100 μm. Representative images of two independent experiments are shown. (C) Schematic representation of the analyses performed in this study. Coloured dots identify the laboratories involved, with numbers indicating cell lines used for each analysis. (D) Flow cytometry analysis of naive H9 hESCs and SIG-1 hiPSCs stably cultured on MEFs or serum coating. Left: Representative contour plots of SUSD-PE *versus* forward scatter. Right: Quantification of SUSD2-positive cells as the frequency of live cells. Technical replicates from $n = 1$ independent experiment are shown as dots. Two-sided unpaired Student's t-test. (E) Growth rate of naive HPD06 and HPD03 hiPSCs stably cultured on MEFs or serum coating. Bars indicate the mean ± SEM of technical replicates shown as dots from $n = 2$ independent experiments. Two-way repeated measures ANOVA. (F) Top: Representative AP staining images after clonal assay of naive HPD06 and HPD03 hiPSCs stably cultured on MEFs or serum coating. Bottom: Quantification of the relative number of AP-positive pluripotent colonies counted per well. Bars indicate the mean ± SEM of technical replicates shown as dots from $n = 3$ independent experiments. Two-sided unpaired Student's t-test. Source data are available online for this figure.

**Table 1. Naive hPSC lines used in this study.**

| Cell line | Origin | Sex | Derivation | Group |
|---|---|---|---|---|
| HPD06 | Induced from human fibroblasts | Male | Somatic cell reprogramming (Giulitti et al, 2019) | Martello |
| HPD03 | Induced from human fibroblasts | Male | Somatic cell reprogramming (Giulitti et al, 2019) | Martello |
| Shef6 (Aflatoonian et al, 2010) | Embryonic | Female | Epigenetic resetting (Guo et al, 2017) | Leeb |
| SIG-1 | Induced from human fibroblasts | Female | mRNA KLF4 reprogramming (Liu et al, 2017) | Pasque |
| KOLF2.1 J | Induced from human fibroblasts | Male | Episomal Sox2-17 and KLF4 reprogramming (MacCarthy et al, 2024) | Rivron |
| SCTi003-A | Induced from human fibroblasts | Female | Episomal Sox2-17 and KLF4 reprogramming (MacCarthy et al, 2024) | Rivron |
| H9 (Thomson et al, 1998) | Embryonic | Female | Epigenetic resetting (Guo et al, 2017) | Leeb |
| | | | | Pasque |
| | | | | Zylicz |
| HNES1 (Guo et al, 2016) | Embryonic | Male | From blastocyst | Leeb |
| | | | | Pasque |
| | | | | Zylicz |

We conclude that serum contains a mixture of ECM proteins previously reported to support the expansion of hPSCs.

A recent study reported that the expansion of naive primate PSCs on commercial basement membrane matrices coating (i.e. Geltrex) resulted in elevated H3K27me3 levels, which needed to be reduced by adding a PRC2 inhibitor to PXGL medium to facilitate stable propagation of naive hPSCs (Huang et al, 2025). Basement membranes are characterised by the presence of Collagen IV and Laminins, which we could not detect consistently in our serum batches. We could not detect an increase in H3K27me3 in naive hPSC expanded on serum coating (Fig. EV2B), suggesting that the composition of the ECM substrate can stabilise the epigenetic state of hPSCs.

## Serum coating does not induce genetic variations in naive hPSCs

Several studies demonstrated that conventional hPSCs can acquire genetic changes during long-term in vitro culture, which commonly include mutations in cancer-associated genes, especially *TP53* (Merkle et al, 2017; Lezmi et al, 2024). We analysed two naive iPSC lines (HPD06 and HPD03), asking whether they acquired mutations in hotspot genes during the adaptation to serum. We collected genomic DNA from HPD06 and HPD03 cultured for more than 2 months after conversion to serum coating and from the same lines kept on MEFs (Fig. 3A) and deep sequenced exomes (more than 100x independent base coverage). The comparison with the reference genome identified

a total of 1641 shared variants (SNVs, indels, and CNVs) in protein-coding regions between naive HPD06 hPSCs cultured on MEFs and serum, likely already present in the donor genome, or accumulated during derivation and previous culture on MEFs (Fig. 3B, left top Venn diagram). HPD06 continuously cultured on MEFs accumulated 936 variants, contrary to the 356 variants specifically detected in the same line cultured on serum. Similarly, more variants were detected in naive HPD03 hPSCs on MEFs than in the same line converted to serum coating (509 and 311, respectively; Fig. 3B, left bottom Venn diagram). We then grouped protein-coding variants according to their clinical impact (tier I, strong significance; tier II, potential clinical significance; tier III, unknown clinical significance; and tier IV, benign or likely benign) (Li et al, 2017). For both HPD06 and HPD03, no Tier I alterations were identified, and Tier IV represented the vast majority of detected variants (Fig. 3B, right pie charts). Interestingly, for both naive lines, the culture on MEFs showed a larger fraction of Tier III variants compared to serum coating (Fig. 3B, pie charts). Few cancer-related genes previously reported as commonly mutated in conventional hPSCs showed only Tier IV benign alterations (*FAT1*, *ASXL1* and *NF1*) (Lezmi et al, 2024). No genetic alteration was found for *TP53*, a predominantly mutated gene during in vitro culture (Merkle et al, 2017; Lezmi et al, 2024).

Overall, culturing naive hPSCs on serum coating did not lead to pathogenic mutations. Moreover, the mutation rates of naive lines cultured on serum were lower than those of cells cultured on MEFs.

**Table 2.** Complete overview of experiments and experimental details of serum-cultured cells.

| Experiment | Figure(s) and panel(s) | Line(s) | Passage(s) on serum coating | FBS batch(es) | Group |
|---|---|---|---|---|---|
| Low-density conversion | 1A,B EV1A–C,E,F | HPD06 | p0-p4 | Gibco 10270106 (2342201) | Martello |
| | | HPD03 | p0-p4 | | |
| Long-term maintenance | 1A,B,D–F EV1B EV2B | HPD06 | >14 | Gibco 10270106 (2342201, 2412072) and Gibco A5256701 (2749488, 2740171, 2740173, 2453915) | Martello |
| | | HPD03 | >14 | | |
| | | H9 | >20 | Sigma Aldrich F7524 (19C111) and Biowest S1600 (S00KI20001) | Leeb |
| | | Shef6 | >20 | | |
| | | HNES1 | >10 | | |
| | | H9 | >10 | Gibco A5256701 (B2873995RP) | Pasque |
| | | SIG-1 | >10 | | |
| | | H9 | >10 | Sigma Aldrich F7524 (0001669689) | Zylicz |
| | | HNES1 | >17 | | |
| Genetic engineering | EV1G | HPD06 | >20 | Gibco A5256701 (B2772471RP) | Martello |
| Proteomics on serum batches | 2 EV2A | / | / | Gibco 10270106 (2412072) and Gibco A5256701 (B2772471RP) | Martello |
| | | | | Biowest S1600 (S00KI20001) | Leeb |
| | | | | Sigma Aldrich F7524 (0001669689) | Zylicz |
| | | | | Gibco A5256801 (2575650H) | Rivron |
| Exome-Seq in maintenance | 3B | HPD06 | p17–18 | Gibco A5256701 (2740173) | Martello |
| | | HPD03 | p17–18 | | |
| RNA-Seq in maintenance | 3C,D EV2C,D EV3 | HPD06 | p14 | Gibco 10270106 (2342201) | Martello |
| | | HPD03 | p14 | | |
| | | Shef6 | >20 | Sigma Aldrich F7524 (19C111) | Leeb |
| Capacitation | 4 EV4 | HPD06 | >20 | Gibco A5256701 (2740173, 2453915) | Martello |
| | | Shef6 | >20 | Sigma Aldrich F7524 (19C111) | Leeb |
| EBs differentiation from capacitated cells | 5A | HPD06 | >20 | Gibco A5256701 (2740173) | Martello |
| TE induction | 5B,C EV5B | H9 | p6 | Biowest S1600 (S00KI20001) | Leeb |
| | | Shef6 | 25 | | |
| | | HNES1 | >15 | | |
| | | HNES1 GATA3::mKO | p8 | | |
| TSC differentiation | 5D,E EV5C–E | HPD06 | >20 | Gibco A5256701 (2749488, 2740171, 2740173, 2453915) | Martello |
| | | HPD03 | >20 | | |
| | | SIG-1 | >10 | Gibco A5256701 (B2873995RP) | Pasque |
| Blastoid generation | 6 EV6 | H9 | >20 | Biowest S1600 (S00KI20001) | Leeb |
| | | H9 | p4–11 | Sigma Aldrich F7524 (0001669689) | Zylicz |
| | | HNES1 | p8–18 | | |
| | | KOLF2.1 J | p5–10 | Gibco A5256801 (2575650H) | Rivron |
| | | STCi003-A | p5–10 | | |
| | | HNES1 | p6 | Gibco A5256801 (B2873995RP) | Pasque |

## Serum coating preserves naive hPSC gene expression

To further characterise the naive features of hPSCs on serum, we compared the transcriptome of naive hPSCs cultured on MEFs and serum coating with a panel of published transcriptomes spanning human naive and primed PSCs, TSCs and fibroblasts. RNA-seq data confirmed a distinct global expression profile of our naive hPSC lines compared with primed hPSCs, TSCs and fibroblasts, as they fall together within the main naive cluster (Fig. 3C). Gene expression analysis confirmed robust induction of several core and

**Table 3. FBS batches used in this study.**

| Supplier | Catalog number | Lot number | Comment | Group |
|---|---|---|---|---|
| Gibco | 10270106 | 2342201 | Used for adaptation & experiments | Martello |
| | | 2426974 | Tested for cell growth and attachment | |
| | | 2412072 | Used for adaptation & experiments | |
| | | 2534380 | Tested for cell growth and attachment | |
| | | 2534396 | Tested for cell growth and attachment | |
| | A5256701 | 2749488 | Used for adaptation & experiments | Martello |
| | | 2740171 | Used for adaptation & experiments | |
| | | 2740173 | Used for adaptation & experiments | |
| | | 2453915 | Used for adaptation & experiment | |
| | | B2695490RP | Tested for cell growth and attachment | |
| | | B2741829RP | Tested for cell growth and attachment | |
| | | B2724133RP | Tested for cell growth and attachment | |
| | | B2772462RP | Tested for cell growth and attachment | |
| | | B2772471RP | Used for adaptation & experiments | |
| | | B2802149RP | Tested for cell growth and attachment | |
| | | B3065189RP | Tested for cell growth and attachment | Leeb |
| | | 2563330 | Tested for cell growth and attachment | |
| | | 2575659H | Used for adaptation & experiments | Rivron |
| | | B2873995RP | Used for adaptation & experiments | Pasque |
| Sigma Aldrich | F7524 | 0001669689 | Used for adaptation & experiments | Zylicz |
| | | 19C111 | Used for adaptation & experiments | Leeb |
| | | 0001670544 | Tested for cell growth and attachment | |
| | | 0001689071 | Tested for cell growth and attachment | |
| | | 0001671877 | Tested for cell growth and attachment | |
| | F4135 | 15D353 | Fails to support attachment and growth | Leeb |
| Biowest | S1600 | S00KI20001 | Used for adaptation & experiments | Leeb |
| | S1810 | S00Q3 | Tested for cell growth and attachment | |
| Cytiva | SV30160 | SV30160.03 | Tested for cell growth and attachment | Leeb |
| | | CK20240003 | Tested for cell growth and attachment | |
| PAN Biotech | P22110 | P30-3306 | Tested for cell growth and attachment | Leeb |
| | P221003 | P30-3302 | Tested for cell growth and attachment | |
| Clontech | 631106 | A13001 | Tested for cell growth and attachment | Leeb |
| Seradigm | 76314 | 102F24 | Tested for cell growth and attachment | Leeb |

naive-specific pluripotency genes (Figs. 3D and EV2C), and low to undetectable expression of primed-specific markers, in line with published transcriptomes.

Primed and naive hPSCs cultured on MEFs showed variable expression of fibroblast markers, possibly due to MEF contamination in collected cells (Figs. 3D and EV2C,D). Instead, fibroblast markers were barely detectable in naive lines cultured on serum coating. To further explore the potential contamination from MEFs mRNA, we aligned the reads against both human and mouse genomes (Fig. EV3A) and observed elevated expression of fibroblast markers only in naive hPSCs cultured on MEFs, which was further increased after alignment to the mouse genome (Figs. EV3B and EV2D). These analyses indicated the presence of

MEFs mRNA contaminations in naive hPSC expanded on feeders, which were therefore filtered out (Fig. EV3A and Methods).

After filtering, we asked whether we could detect significant transcriptional differences between naive hPSCs cultured on MEFs or serum coating. Differential expression analyses in 2 naive hPSC lines revealed a small number of differentially expressed genes (DEGs), most of which were pseudogenes (Fig. EV3C), with no significant enrichment for any biological process. Notably, we could not detect differences in the expression of any pluripotency or differentiation markers.

It has been observed that mitochondrial activation and metabolic rewiring occur concurrently with the generation and stabilisation of naive hPSCs (Takashima et al, 2014). Mouse ESCs

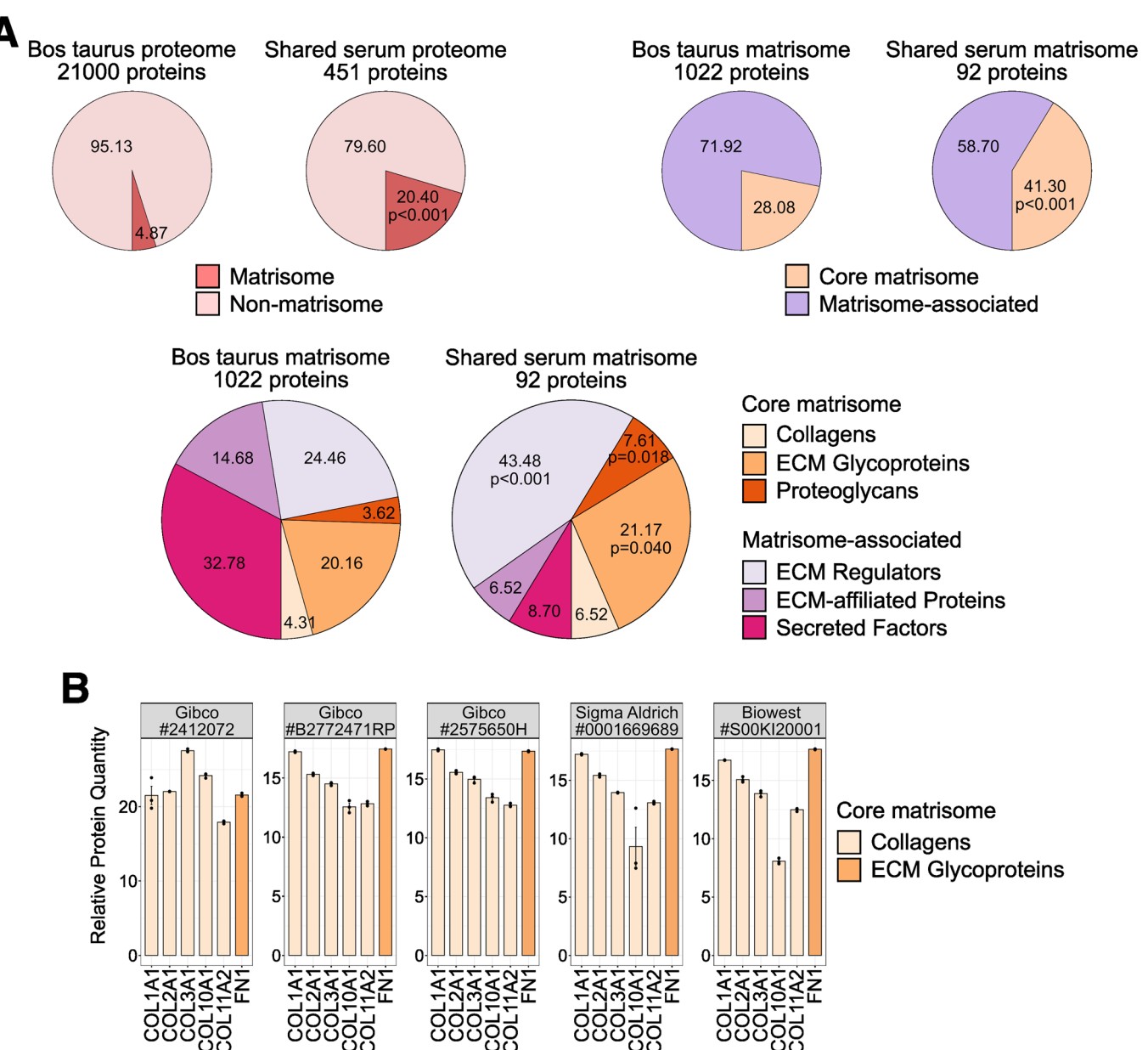

**Figure 2. Serum contains a mixture of ECM proteins.**

(A) Matrisome proteins enrichment analysis of the proteome shared between 5 different serum coating batches in comparison to the reference Bos taurus ECM proteome (Petrov et al, 2023). The obtained list of proteins was analysed using the Matrisome AnalyseR tool (https://sites.google.com/uic.edu/matrisome/tools/matrisome-analyzer) (Petrov et al, 2023) from The Matrisome Project. The composition of matrisome proteins is expressed in percentage. Binomial test for increasing percentages compared to the reference. (B) Protein abundance of collagens and fibronectin (FN1) shared between 5 different serum coating batches (identified by supplier and lot number). Bars indicate the mean ± SEM of n = 3 technical replicates shown as dots. Source data are available online for this figure.

use oxidative phosphorylation, while mEpiSCs and hPSCs are mostly glycolytic and have low ability for mitochondrial respiration (Zhou et al, 2012). We collected the list of oxidative phosphorylation genes from KEGG (ko00190 pathway; https://www.genome.jp/kegg/pathway.html) and added the complex IV cytochrome c oxidase (COX) genes (Takashima et al, 2014). In naive hPSCs on MEFs and on serum coating, we observed a comparable increase in the expression of most genes involved in oxidative phosphorylation (Fig. EV3D). Furthermore, in agreement with previously published

data (Takashima et al, 2014; Zhou et al, 2012), the expression of most of the COX genes was decreased in primed hPSCs compared to naive hPSCs, both on MEFs and serum coating.

Previous studies reported biallelic expression of some imprinted genes in naive hPSCs (Pastor et al, 2016; Bar et al, 2017; Martini et al, 2022). Therefore, we analysed our bulk RNA-Seq samples with BrewerIX (Martini et al, 2022). We detected a significant biallelic expression of *H19* and *ZIM2/PEG3* specifically in all naive hiPSCs, either on serum or MEFs, but not in primed hiPSCs

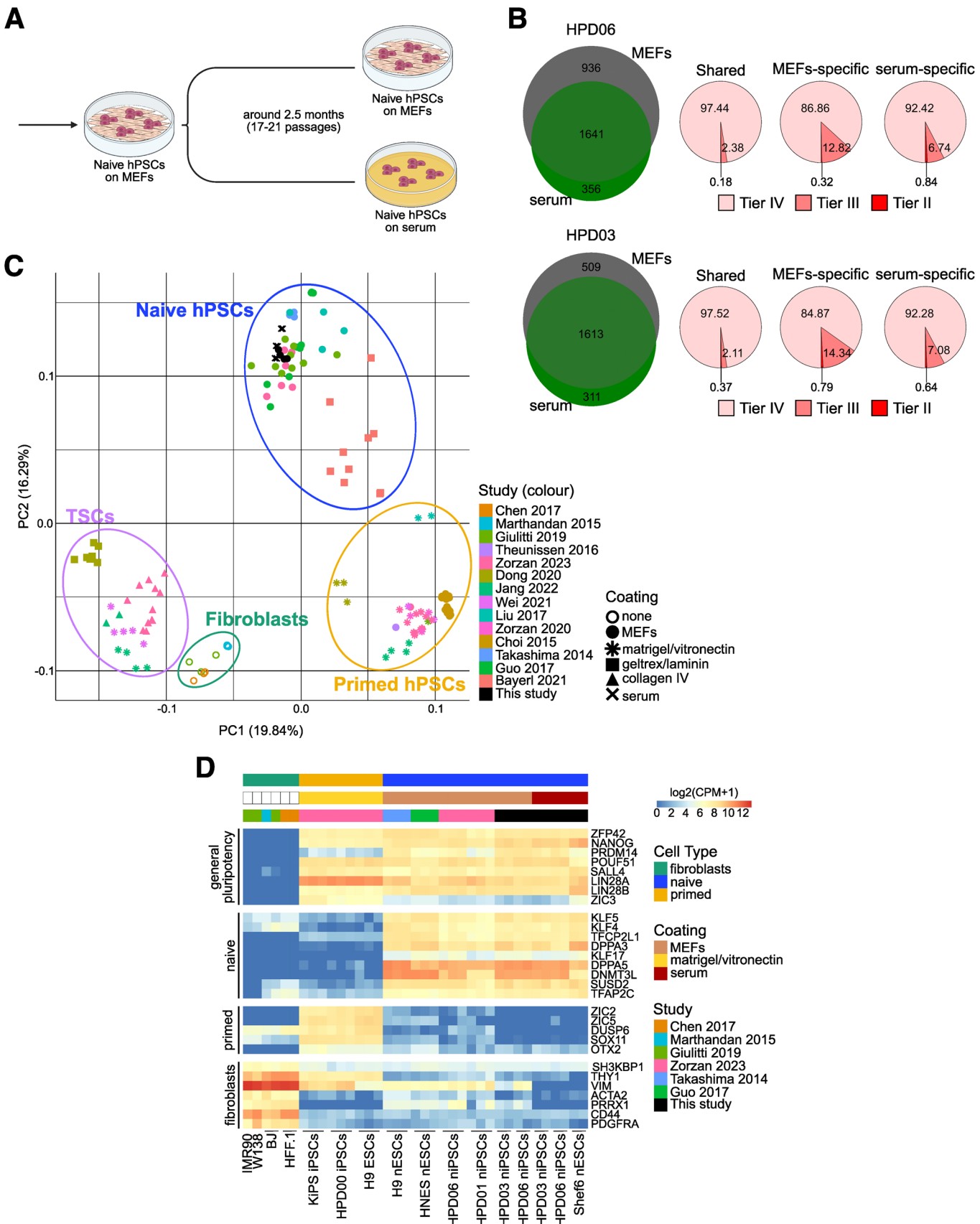

**Figure 3. Naive hPSCs on serum coating do not accumulate pathogenic mutations and maintain their transcriptomic profile.**

(A) Schematic representation of the experimental setting for the Exome-Seq analysis of naive HPD06 and HPD03 hiPSCs stably cultured on MEFs or serum coating. (B) Analysis of detected genetic variations in coding genes of naive HPD06 and HPD03 hiPSCs stably cultured on MEFs or serum coating compared to the reference human genome. Left: Venn diagrams showing the quantification of shared and culture-specific (MEFs or serum coating) detected variants. Right: Pie charts of the composition of shared and culture-specific (MEFs or serum coating) detected variants classified by Tier and expressed in percentage. (C) Principal component analysis of naive HPD06 and HPD03 hiPSCs and Shef6 hESCs stably cultured on MEFs or serum coating with published naive and primed hPSCs, TSCs and fibroblast lines performed on the top 5000 most variable genes identified through RNA-seq. (D) Heatmap of general pluripotency, naive, primed, and fibroblasts genes in naive HPD06 and HPD03 hiPSCs and Shef6 hESCs stably cultured on MEFs or serum coating and in published fibroblasts, primed and naive hPSCs. See also Fig. EV2C. Source data are available online for this figure.

(Fig. EV3E), as previously reported (Nazor et al, 2012; Pastor et al, 2016). Overall, naive cells displayed similar allelic expression of imprinted genes on serum coating and MEFs.

We concluded that naive hPSCs cultured on serum coating maintain their distinctive transcriptome profile without significant alterations compared to naive hPSCs cultured on MEFs.

## Serum-adapted naive hPSCs faithfully recapitulate embryonic differentiation trajectories

Upon withdrawal of self-renewal cues, naive hPSCs exit the naive pluripotent state. Before gaining the ability to commit to embryonic cell fates, they need to undergo a process termed capacitation (Rostovskaya et al, 2019). Capacitation requires approximately eight to ten days, consistent with the timing between the blastocyst and the pre-gastrulation post-implantation embryo in vivo. Hence, to assess the capacitation and differentiation potential of serum-cultured naive hPSCs, we subjected them to a variety of differentiation conditions. Changes in morphology were equivalent between naive hPSCs originally grown on MEFs or serum coating, across multiple differentiation media (Fig. EV4A). Furthermore, expression of naive-specific markers (*TFCP2L1*, *KLF4* and *NANOG*) was downregulated while expression of primed markers (*SALL2*, *CDH2* and *ZIC2*) was induced during the exit from naive pluripotency, irrespective of whether naive hPSCs were maintained on MEFs or serum before the induction of differentiation (Figs. 4A,B and EV4B).

To further characterise gene expression behaviour during capacitation, we explored transcriptome changes throughout an eight-day differentiation time course in various conditions: unsupplemented N2B27, N2B27 supplemented with TNKS1/2 inhibition (XAV) (Rostovskaya et al, 2019) or FGF2-ActivinA-XAV (FAX), or E8 or mTeSR, which contain FGF and Activin A. All differentiation was induced from naive Shef6 hESCs cultured on serum. We first noted that differentiation time points, rather than medium conditions, had the largest impact on clustering (Fig. 4C) and that the Spearman correlation between the same time points of different conditions was high (Fig. EV4C). Consistently, expression kinetics of *KLF4* and *ZIC2* were equivalent between conditions (Fig. 4D). Comparison with published data for MEF-cultured H9 and HNES1 naive hESC lines differentiating with XAV (Rostovskaya et al, 2019), showed that Shef6 naive hESCs cultured on serum coating followed the same trajectory as H9 and HNES1 hESCs cultured on MEFs before capacitation (Fig. 4E), and have similar expression patterns for known pluripotency genes (Fig. 4F). We further noted that the presence of XAV had no discernible effect on gross gene expression or proliferation compared to capacitation in

N2B27 alone. We surmise that the gene expression programme accompanying capacitation is not substantially affected by the substrate used for naive culture.

We next benchmarked our in vitro capacitation system starting from serum-coated naive hPSCs by comparing our RNA-seq time course data to data of human ex vivo cultured pre- and peri-implantation embryos (Zhao et al, 2025). Principal component analysis showed a striking alignment between our in vitro capacitation system and embryo data (Fig. 4G). This is reflected in a clear correlation between differentiation-induced gene expression changes in vitro and in vivo at both the global transcriptome (Fig. EV4D) and the individual pluripotency marker level (Fig. EV4E,F).

Taken together, these findings demonstrate that serum-adapted hPSCs progress through the developmental states of pluripotency at least as well as feeder-dependent hPSCs and that this model can generate results compatible with in vivo embryonic development.

## Naive hPSCs on serum coating efficiently differentiate to the three primary germ layers and trophectoderm cell fates

After confirming that naive hPSCs cultured on serum can exit pluripotency under different conditions with a progression similar to that of the same cell lines cultured on MEFs, we wondered if naive hPSCs cultured in feeder-free conditions still retain the capacity to efficiently differentiate into embryoid bodies (EBs) composed of the three germ layers. Capacitated hPSCs from naive cells previously cultured on MEFs or serum coating were aggregated in 3D and let spontaneously differentiate for 14 days (Fig. EV5A). EBs derived from both conditions showed an upregulation of germ layer markers compared to capacitated cells (Fig. 5A). We conclude that naive hPSCs cultured on serum differentiate to the three germ layers with an efficiency similar to naive hPSCs previously cultured on MEFs.

Naive hPSCs possess the ability to directly enter the trophecto-derm lineage when exposed to dual MEK and TGF-β inhibition (Guo et al, 2021; Io et al, 2021b). This potential is the prerequisite for the generation of blastoid models and a key feature of naive human pluripotency. We hence compared the ability to initiate TE-differentiation of three distinct hPSC lines cultured on serum or on MEFs. Our results showed that serum-cultured hPSCs induced the TE transcription factor GATA3 and the TE-specific surface marker TROP-2 to the same extent as their MEF-cultured counterparts (Figs. 5B,C and EV5B).

Naive hPSCs cultured on MEFs can also promptly differentiate toward TSCs when exposed to TSC medium (Cinkornpumin et al,

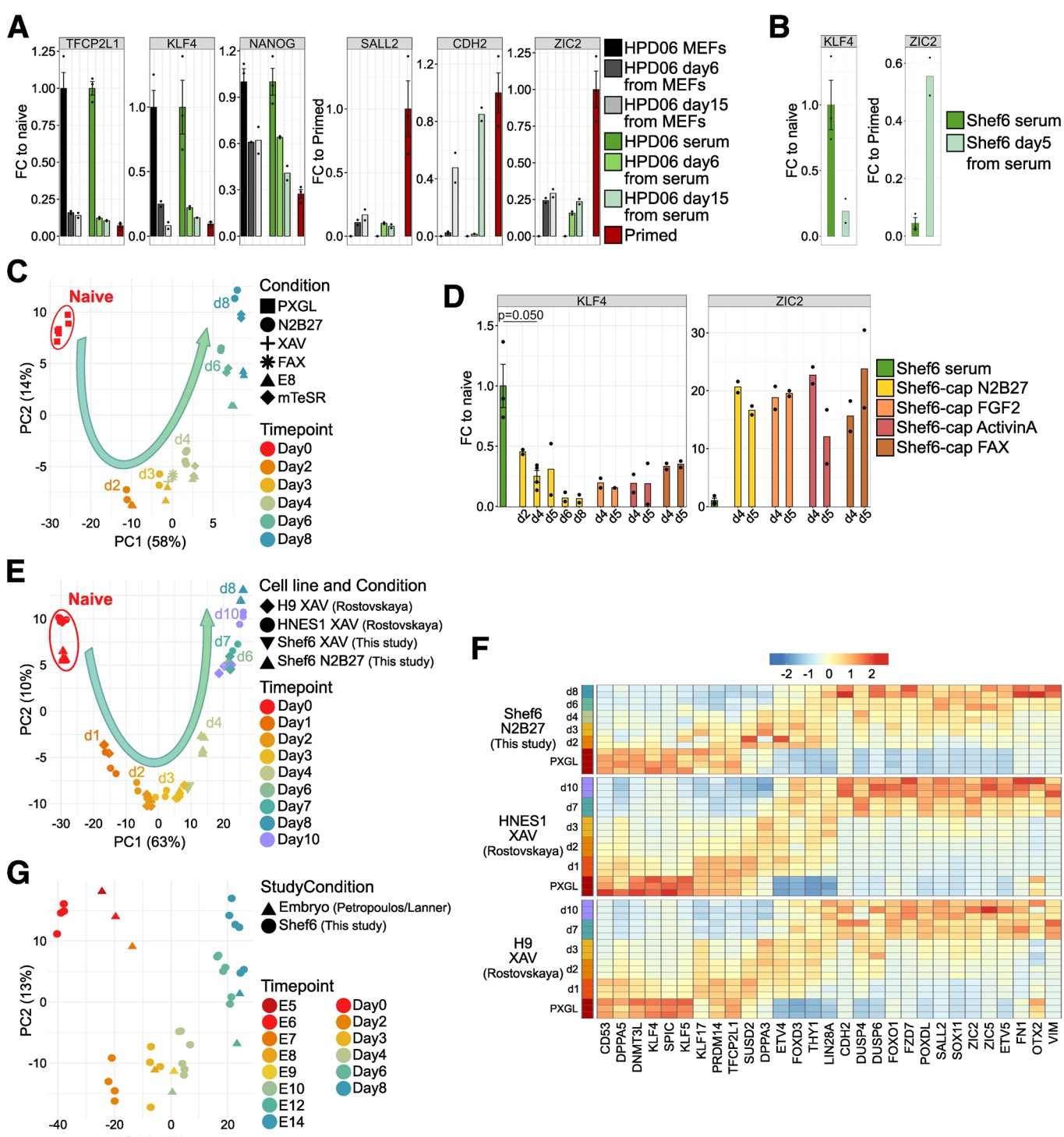

2020; Dong et al, 2020; Guo et al, 2021). Therefore, we assessed whether naive hPSCs cultured on serum coating still retain this capacity. We obtained TSCs within a few passages from naive hiPSCs stably cultured in both conditions, which showed indistinguishable morphology (Fig. EV5C) and similar downregulation of naive genes *TFCP2L1* and *KLF17* and upregulation of TSC genes *GATA3* and *GATA2* (Fig. 5D). Consistently, the GATA3 signal was detected by immunofluorescence at a comparable level to stable naive-derived TSCs (passage 15), while KLF17 and OCT4 were

undetectable (Figs. 5E and EV5D). Several genes (e.g. *TFAP2A*, *KRT7*, and *GATA3* and *GATA2* to a lesser extent) are shared between TSCs and amnion, as early amniogenesis has been reported to occur via a trophectoderm-like route (Rostovskaya et al, 2022). Therefore, we wondered if TSCs derived from naive hiPSCs cultured on serum retain the same low expression of amnion markers as naive hiPSCs cultured on MEFs. We previously reported that BAP medium-treated primed hPSCs showed increased expression of *GATA3* together with amnion markers

**Figure 4.   Feeder-free naive hPSCs can recapitulate embryonic differentiation trajectories.**

(A) Gene expression analysis by RT-qPCR of naive (*TFCP2L1* and *KLF4*), general (*NANOG*) and primed (*SALL2*, *CDH2* and *ZIC2*) pluripotency markers in HPD06 hiPSCs capacitated from naive lines stably cultured on MEFs or serum coating. Bars indicate the mean ± SEM of technical replicates shown as dots from $n = 3$ independent experiments for naive hPSCs in PXGL and primed. Technical replicates from $n = 2$ independent experiments for capacitated cells at days 6 and 15 are shown as dots. (B) Gene expression analysis by RT-qPCR of naive (*KLF4*) and primed (*ZIC2*) pluripotency markers in Shef6 hESCs and HPD06 hiPSCs capacitated from naive lines stably cultured on serum coating. Bars indicate the mean ± SEM of technical replicates shown as dots from $n = 2$ independent experiments for naive hPSCs. Technical replicates from $n = 2$ independent experiments for capacitated cells at day 5 are shown as dots. (C) Principal component analysis of all genes identified via RNA-seq of an eight-day differentiation time course starting from naive Shef6 hESCs stably cultured on serum coating in PXGL. Differentiation was induced by culture in basal medium (N2B27) with or without supplementation of TNKS1/2 inhibition (XAV) or FGF2-ActivinA-XAV (FAX) or commercial media containing FGF2 and Activin A signalling modulators (E8 or mTeSR). Technical replicates include $n = 6$ from two independent experiments for d0, $n = 4$ from two independent experiments for N2B27 at day 4, and $n = 2$ for all other combinations of conditions and time points. (D) Gene expression analysis by RT-qPCR of naive (*KLF4*) and primed (*ZIC2*) pluripotency markers of naive Shef6 hESCs stably cultured on serum coating and capacitated in basal medium (N2B27) or supplemented with FGF2, Activin A or a combination of FGF2, Activin A and XAV. Bars indicate the mean ± SEM of technical replicates shown as dots from $n = 2$ independent experiments for naive hPSCs and N2B27 capacitation at day 4. Technical replicates from $n = 1$ independent experiments for all other combinations of conditions and time points are shown as dots. Two-sided unpaired Student's t-test. (E) Principal component analysis of the abovementioned feeder-free eight-day differentiation time course in N2B27 with or without supplementation of XAV integrated with already published RNA-seq data of naive H9 and HNES1 hESCs cultured on MEFs and capacitated in N2B27 supplemented with XAV for up to 10 days (Rostovskaya et al, 2019). Data was batch-corrected for datasets. Technical replicates include $n = 6$ from two independent experiments for d0, $n = 4$ from two independent experiments for N2B27 at day 4, $n = 2$ for all other combinations of conditions and time points, and $n = 3$ for each combination of cell line, condition, and time point in the published dataset. (F) Heatmaps showing the expression of general pluripotency, naive, and primed markers. Data includes serum-adapted naive Shef6 hESC lines capacitated in N2B27 and published data for naive H9 and HNES1 hESCs cultured on MEFs and capacitated in N2B27 supplemented with XAV (Rostovskaya et al, 2019). Biological replicates include $n = 6$ from two independent experiments for PXGL, $n = 4$ from two independent experiments for N2B27 at day 4, $n = 2$ for all other combinations of conditions and time points, and $n = 3$ for each combination of cell line, condition, and time point in the published dataset. Expression levels are row-wise z-transformed DESeq2-normalised counts. (G) Principal component analysis of the abovementioned feeder-free eight-day differentiation time course (N2B27, E8 and mTeSR) integrated with pseudo bulk data derived from the human embryo reference dataset (subset for embryonic cells spanning E5-E14). Data was batch-corrected for data sets and then filtered by the 9376 DEGs (|log2FC| > 1, padj < 0.05) when comparing the naive state with any of the differentiation conditions and time points. Technical replicates include $n = 4$ from two independent experiments for d0 and $n = 2$ for all other combinations of conditions and time points. Source data are available online for this figure.

*IGFBP3* and *BAMBI* (Zorzan et al, 2023). Therefore, we exposed primed Keratinocytes induced Pluripotent Stem Cells (KiPS) to BAP medium and observed a cell morphology consistent with previous reports (Yang et al, 2015; Io et al, 2021b; Zorzan et al, 2023) and completely distinct from our naive hiPSCs-derived TSCs (Fig. EV5E, left). We further analysed the expression of the amnion markers (Cinkornpumin et al, 2020; Guo et al, 2021) and detected lower expression of *IGFBP3* and undetectable expression of *BAMBI* in TSCs derived from naive hiPSCs cultured both on MEFs and serum coating compared to BAP-treated cells (Fig. EV5E, right).

We concluded that naive hiPSCs cultured on MEFs or serum coating readily differentiate to bona fide trophectoderm cells with similar efficiency.

## Naive hPSCs on serum coating efficiently self-organise into blastoids

Previous studies have reported that naive hPSCs are able to self-organise into blastoids upon aggregation. Blastoids are stem cell-based embryo models reminiscent of the human blastocyst (Yu et al, 2021; Yanagida et al, 2021; Kagawa et al, 2022). Such structures robustly specify an outer layer of TE-like cells encompassing the pluripotent epiblast (EPI)-like cells and rare PrE-like cells. To test if naive hPSCs cultured on serum retain this unique feature, we optimised the Kagawa et al (Kagawa et al, 2022; Heidari Khoei et al, 2023) blastoid induction protocol, by passaging naive hPSCs at high density on serum coating, before aggregation in AggreWells, but without the need for a time-consuming plastic attachment step typically used to deplete MEFs from the cultures.

Multiple naive hPSCs (Tables 1 and 2) cultured on serum coating were able to aggregate and form blastoid structures with comparable cavitation efficiency, diameter, and rate of single cavities to blastoids derived from naive hPSCs cultured on feeders (Figs. 6A and EV6A,B). Immunofluorescence analysis validated

that these structures contain TE-like (GATA2/3 or CDX2), EPI-like (NANOG, OCT4 and KLF17), and rare PrE-like cells (GATA4 and SOX17) (Figs. 6B and EV6B,C). This analysis confirmed that naive hESCs and hiPSCs cultured on serum-coated dishes are able to robustly self-organise into blastoids.

To determine cellular identities, we performed 10x scRNAseq on day 5 blastoids derived from naive H9 hESCs cultured using serum-coating. This analysis showed two main populations of cells on the UMAP (Fig. 6C). Based on the expression of key marker genes, the two populations were annotated as TE-like and EPI-like cells (Fig. 6D,E), with TE-like cells being underrepresented due to their fragility and loss at the single-cell dissociation stage. Notably, TE-like cells expressed a wide range of trophoblast markers (e.g., *GATA3*, *NR2F2*, *GATA2*) but not those of the amnion (e.g., *ISL1*, *TFAP2B*) or other off-target lineages (Figs. 6D,E and EV6D). Similarly, EPI-like cells expressed both core (e.g., *POU5F1*, *NANOG*) and naive pluripotency markers (e.g., *DPPA5*, *DNMT3L*) but not those associated with primed pluripotency (e.g., *OTX2*, *ZIC2*) (Figs. 6D,E and EV5D). Previous reports have shown that PrE-like cells are rarely specified during blastoid induction (Kagawa et al, 2022). Consistent with this finding, we only observed sporadic individual cells expressing the PrE marker *GATA4*, and no Seurat cluster contained a significant number of PrE-like cells (Fig. EV6D,E). To further validate these annotations, we integrated our data with embryo reference datasets (Fig. EV6F) (Yan et al, 2013; Petropoulos et al, 2016; Xiang et al, 2020; Meistermann et al, 2021; Tyser et al, 2021; Yanagida et al, 2021; Zhao et al, 2025). Our analysis further confirmed that cluster 4 is predominantly composed of TE-like cells, while the remaining clusters retain EPI identity (Figs. 6F and EV6F,G). This analysis confirmed that the blastoids contain bona fide TE-like and EPI-like cells. Overall, the formation of blastoids was achieved using 4 lines in 4 different laboratories, demonstrating the robustness of the protocols and their potential for wide adoption in the community.

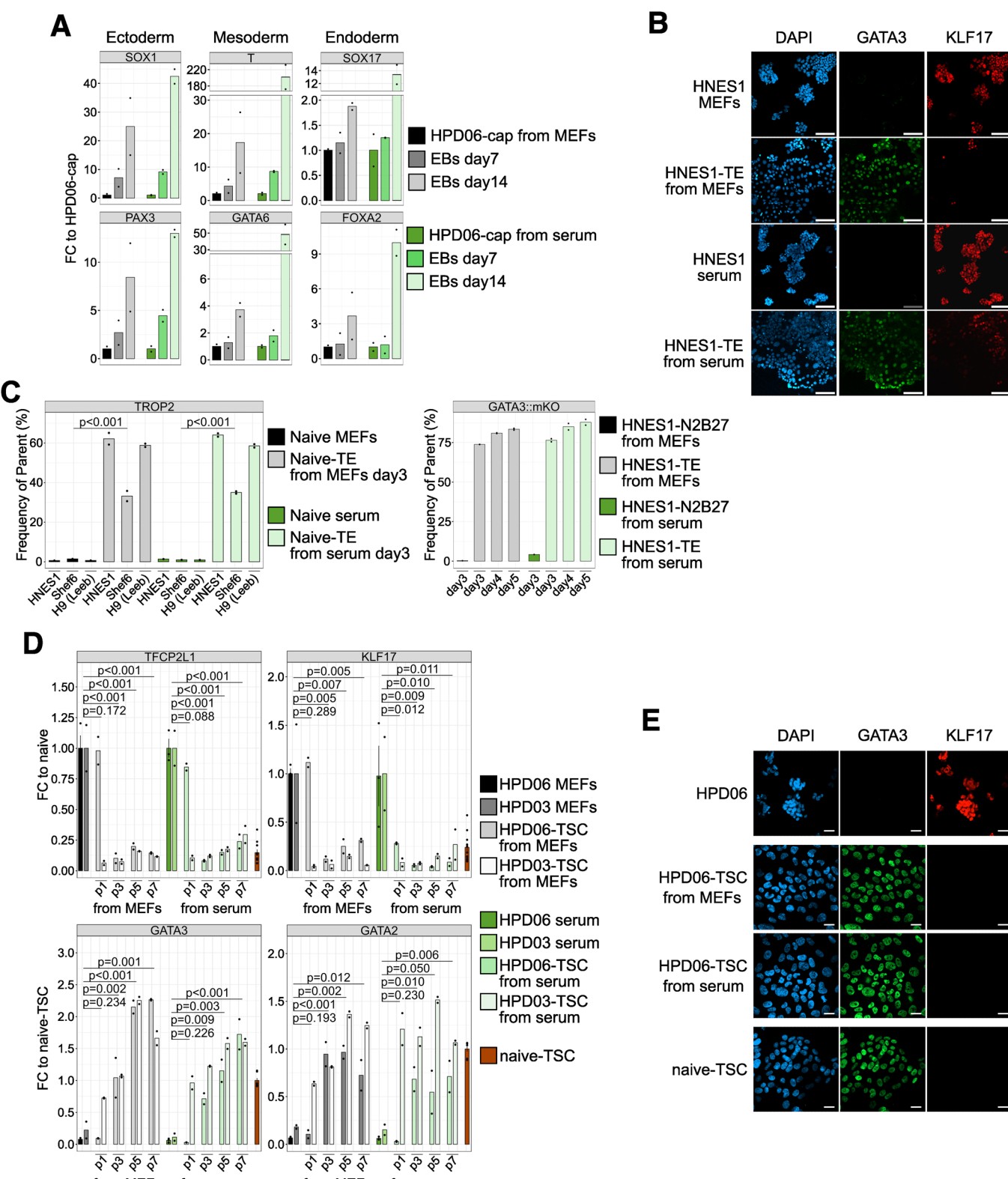

**Figure 5. Feeder-free naive hPSCs efficiently differentiate to embryonic and extraembryonic lineages.**

(A) Gene expression analysis by RT-qPCR of germ layers markers (ectoderm: *SOX1* and *PAX3*, mesoderm: *T* and *GATA6*, and endoderm: *SOX17* and *FOXA2*) in EBs derived from naive HPD06 hiPSCs capacitated from MEFs or serum coating. Technical replicates from $n = 2$ independent experiments are shown as dots. (B) Immunostaining for TE/TSCs (GATA3) and naive pluripotency (KLF17) markers after 5 days of TE differentiation from naive HNES1 hESCs stably cultured on MEFs or serum coating. Scale bars: 100 μm. Representative images of two independent experiments are shown. (C) Flow cytometry analysis of TE derived from naive HNES1, Shef6, and H9 hESCs stably cultured on MEFs or serum coating. Left: Quantification of TROP2-positive cells as the frequency of live cells at Day 3 of differentiation. Technical replicates from $n = 1$ independent experiment for naive hPSCs and $n = 2$ independent experiments for differentiated cells are shown as dots. Two-sided unpaired Student's t-test. Right: Quantification of GATA3-positive cells as the frequency of live cells at day 3 of differentiation in N2B27 and at different time points during differentiation to TE. Technical replicates from $n = 1$ independent experiment are shown as dots. (D) Gene expression analysis by RT-qPCR of naive pluripotency (*TFCP2L1* and *KLF17*) and TE/TSCs (*GATA3* and *GATA2*) markers in TSCs derived from naive HPD06 and HPD03 hiPSCs stably cultured on MEFs or serum coating. Bars indicate the mean ± SEM of technical replicates shown as dots from $n = 3$ independent experiments for naive hPSCs and naive-TSCs. Technical replicates from $n = 2$ independent experiments for the differentiation time points are shown as dots. Two-sided unpaired Student's t-test. (E) Immunostaining for TE/TSCs (GATA3) and naive (KLF17) or general pluripotency (OCT4) markers of trophectoderm derived from naive HPD06 hiPSCs stably cultured on MEFs or serum coating. Scale bars: 50 μm. Representative images of two independent experiments are shown. Source data are available online for this figure.

Overall, feeder-free naive hESCs, when aggregated, robustly self-assemble into blastoids containing cells with appropriate lineage identities.

## Discussion

Expansion of human naive PSCs on a layer of inactivated MEFs has several limitations. Production of MEFs requires the sacrifice of mice, which contrasts with the 3Rs principles for the ethical use of animals in research. MEF production is also time-consuming, expensive, and variable, limiting the range of application of human naive PSCs. Indeed, both mESCs and conventional hPSCs became widespread models only once feeder-free conditions were developed (Smith et al, 1988; Ludwig et al, 2006; Braam et al, 2008; Chen et al, 2011).

The use of MEFs brings an additional hurdle, which is the co-culture of non-human cells with hPSCs. The former might appear as contaminants in global analyses of the latter, as suggested by our analyses of several RNA-sequencing datasets revealing widespread expression of fibroblast markers. Furthermore, the two cell populations might affect each other in a non-controllable manner.

Conventional hPSCs can be expanded without feeders on plates coated with recombinant ECM proteins, such as fibronectin and vitronectin (Braam et al, 2008; Chen et al, 2011). Alternatively, solubilised basement membrane matrices, commercialised as Matrigel or Geltrex, can be used (Ludwig et al, 2006). However, these substrates have been tested for human naive PSC expansion with limited success. In some cases (i.e. laminin), only short-term expansion could be achieved, while in others (i.e. Matrigel) stable expansion could be achieved only with additional inhibitors in the media (Takashima et al, 2014; Gafni et al, 2013; Szczerbinska et al, 2019; Huang et al, 2025).

Importantly, commercially available defined ECM substrates are expensive and range from 0.5 to 2 Euros/cm². In contrast, the cost of using serum coating is reduced by two orders of magnitude to around 0.5 cents/cm², while allowing for long-term expansion without requiring any additional changes in media formulation or culture regime. Implementing our approach requires batch testing of serum, a standard procedure for over 40 years, also required for the culture of mESCs with serum-based media (Behringer et al, 2017). We tested over 30 lots of FBS from 7 different suppliers throughout this study and found that all but one lot supported the expansion of naive hPSCs; thus, batch testing should be a straightforward procedure. Serum can be ordered and tested in batches of over 150 L, drastically reducing the total amount of time invested in batch testing compared to MEF production. Notably, one single 500 ml bottle of FBS is sufficient to generously coat more than 1000 full 6-well plates; hence, a single batch of 100 bottles will sustain 100,000 6-well plates. Of note, a single round of MEF production will take considerably more time and effort than serum batch testing and typically involves the expansion of MEFs to more than 100 15-cm dishes to generate sufficient amounts to coat only 400 6-well plates. Therefore, using serum coating provides a significant advantage for culturing human naive hPSCs.

We performed stable genetic manipulations on naive PSCs expanded on serum, without the need to re-optimise protocols previously used on MEFs. We could swiftly generate lines stably expressing constructs of interest, without the need for multi-resistant MEFs, which are not easily available to all laboratories. We should mention that the recovery of colonies after fluorescence-activated cell sorting was higher on feeders, as also observed during genetic engineering of murine ESCs (Lazzarano et al, 2018; Yamanishi et al, 2018; Acosta et al, 2018; Tamura et al, 2021; Tsai et al, 2023). We note that shuttling serum-adapted naive hESCs between serum coating and MEFs works without the need for re-adaptation to feeder-free conditions. Therefore, MEFs could be used only for the recovery of clonal colonies after genetic manipulations.

For all these reasons, serum-coating represents a significant improvement over current methodologies, allowing for the widespread use of naive hPSCs for several applications, such as large-scale screenings, epigenetic and metabolic profiling, and the generation of embryo models such as blastoids. Expansion of naive hPSCs on serum resulted in a streamlined blastoid protocol, which worked in multiple laboratories with multiple cell lines, with an efficiency as high as feeder-based protocols.

A recent study showed that human naive PSC colonies are surrounded by laminins and collagens (Cesare et al, 2022), which are, at least in part, produced by naive hPSCs themselves. However, this autocrine ECM protein production is not sufficient for hPSC expansion, as either MEFs or exogenous substrates are needed. Our proteomic analysis of the sera coating allowing for naive hPSCs expansion revealed the shared presence of several Collagens together with Fibronectin. We, therefore, speculate that this combination of ECM proteins is crucial for naive hPSC expansion,

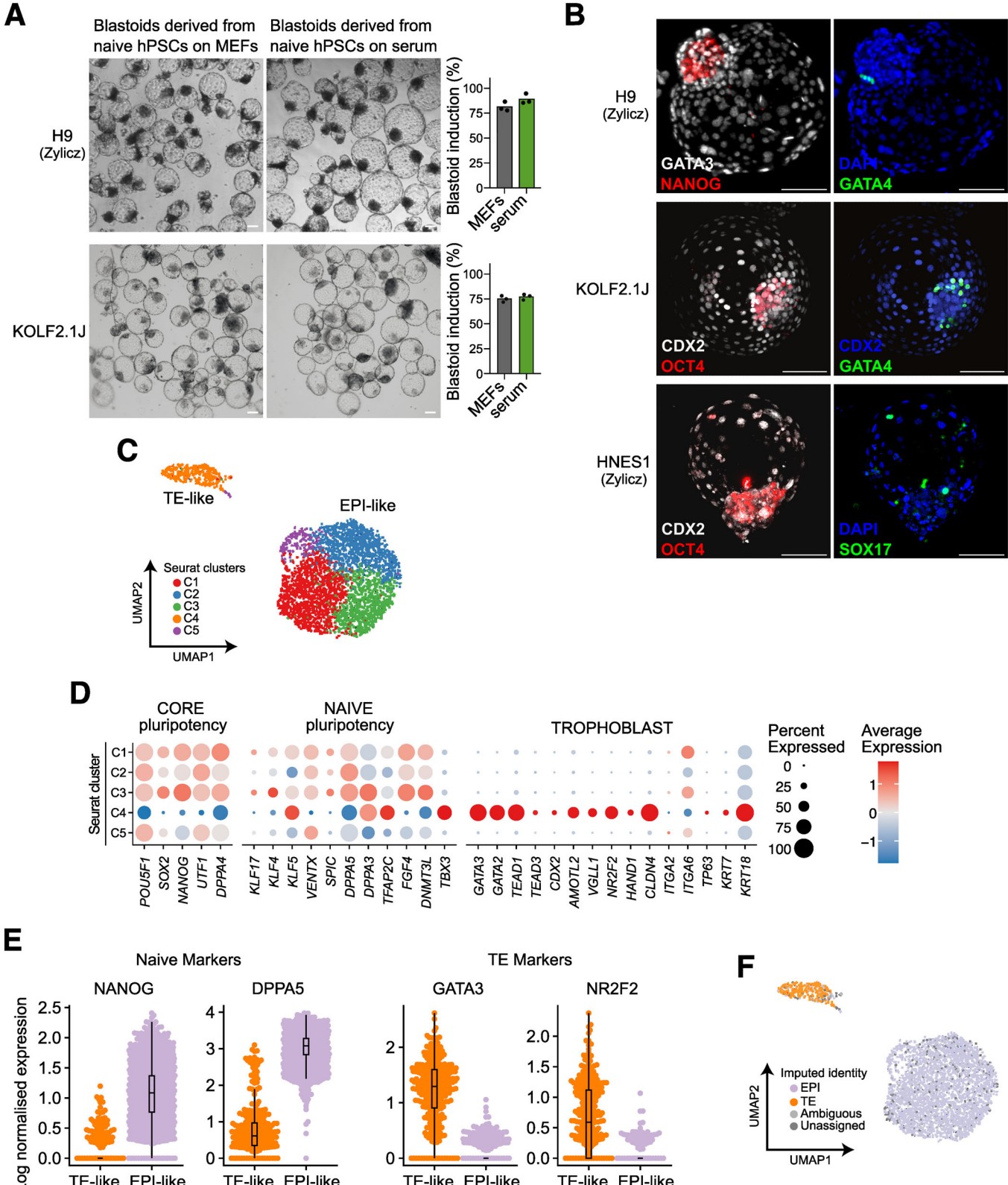

**Figure 6. Feeder-free naive hPSCs efficiently form blastoids.**

(A) Left: Representative brightfield images of blastoids induced from naive H9 hESCs and KOLF2.1J hiPSCs cultured on MEFs or serum coating. Scale bar: 100 μm. Right: quantification of induction efficiency from naive H9 hESCs and naive KOLF2.1J hiPSCs cultured on MEFs or serum-coated dishes. Data is based on brightfield images of d5 blastoids. Technical replicates from $n = 3$ independent experiments for naive H9 hESCs and $n = 1$ independent experiment for naive KOLF2.1J hiPSCs are shown as dots. (B) Immunostaining of d5 blastoids derived from naive H9 and HNES1 hESCs and KOLF2.1J hiPSCs cultured on serum coating. Blastoids were stained for TE (GATA3 or CDX2), PrE (GATA4 or SOX17) and EPI (NANOG and OCT4) markers. Shown is the maximum projection. Scale bars: 100 μm. (C) UMAP plot from scRNA-seq analysis of blastoids from naive H9 hESCs cultured on serum coating for 4 passages. Cells are coloured according to the sample Seurat cluster ($n = 3260$). Highlighted are TE-like (cluster 4) and EPI-like clusters. (D) Expression of selected lineage-specific marker genes across Seurat clusters from (C). The size of the dots represents the proportion of cells in the indicated group expressing the given gene, and colour encodes the scaled average expression. (E) Combined beeswarm and box plots of gene expression of naive pluripotency markers (*NANOG* and *DPPA5*) and TE markers (*GATA3* and *NR2F2*) from (C). Cells were separated based on their EPI or TE signature. Single cells are plotted as dots; boxes span the interquartile range (IQR), the centre line indicates the median, and whiskers extend to values within 1.5 times the IQR. (F) UMAP plots as in (C) but with cells based on the imputed annotation of in vitro samples from a reference in vivo dataset (Zhao et al, 2025). Unassigned and ambiguous labels refer to cells with either none or with more than two imputed annotations, respectively. Source data are available online for this figure.

but future individuation and analysis of serum batches that do not support naive hPSCs and functional studies will be needed to tease out the contribution of each individual ECM protein.

This manuscript represents a joint effort of several laboratories routinely working with naive hPSCs. Once a methodology was identified by one lab, reagents and protocols were shared with other labs, resulting in extensive testing of multiple lines for different applications before publication. This constructive and cooperative approach allowed us to quickly build trust in our methodology, thanks to the testing performed in parallel by 5 laboratories.

## Methods

### Reagents and tools table

| Reagent/Resource | Reference or Source | Identifier or Catalog Number |
|---|---|---|
| **Experimental models** | | |
| Naive hPSCs | This study | Listed in Table 1 |
| SCTi003-A naive hPSCs | STEMCELL Technologies | 200-0511 |
| SIG-1 naive hPSCs | Sigma-Aldrich | EPITHELIAL-1-IPSC0028 |
| H9 hESCs | WiCell | WA09 |
| MEFs DR4 | ATCC | SCRC-1045 |
| | C57BL/6 | SCRC-1002 |
| **Recombinant DNA** | | |
| EGFP-piggyBAC hygomycin | in-house | |
| pBase transposase | In-house | |
| **Antibodies** | | |
| Listed in Table EV2 | | |
| **Oligonucleotides and other sequence-based reagents** | | |
| PCR primers | This study | Listed in Table EV1 |
| **Chemicals, Enzymes and other reagents** | | |
| Fetal Bovine Serum | Several | Listed in Table 3 |
| DMEM high glucose | Gibco | 41965039 |
| | | 41966029 |
| | Merck Sigma Aldrich | D5671 |

| Reagent/Resource | Reference or Source | Identifier or Catalog Number |
|---|---|---|
| DMEM/F12 | Gibco | 11320074 |
| | | 31330038 |
| Neurobasal | Gibco | 21103049 |
| E8 flex | Gibco | A2858501 |
| N2 Supplement | Gibco | 17502048 |
| B27 Supplement | Gibco | 17504044 |
| L-glutamine | Gibco | 25030024 |
| 2-mercaptoethanol | Merck Sigma Aldrich | M3148 |
| | Gibco | 31350- 010 |
| PD0325901 | Axon Medchem | 1408 |
| | MedChem Express | HY-10254 |
| XAV939 | Axon Medchem | 1527 |
| Gö6983 | Axon Medchem | 2466 |
| LIF | Qkine | Qk036 |
| Y27632 | Axon Medchem | 1683 |
| | Tocris | 1254 |
| Activin A | Qkine | Qk001 |
| | PeproTech | 120-14E |
| FGF2 | Qkine | Qk002 |
| | PeproTech | 100-18B |
| A83-01 | Axon Medchem | 1421 |
| | Peprotech | 9094360 |
| | Merck Sigma Aldrich | SML0788 |
| | MedChem Express | HY-10432 |
| TrypLE | Gibco | 12563029 |
| | Thermo Fisher Scientific | A1217701 |
| Accutase | Merck Sigma Aldrich | A6964 |
| Bovine serum albumin | Gibco | 15260-037 |
| | Merck Sigma Aldrich | A3059 |
| | Merck Sigma Aldrich | A7979 |
| ITS-X | Gibco | 51500056 |
| L-ascorbic acid | Merck Sigma Aldrich | A4544 |
| | Merck Sigma Aldrich | A8960 |

| Reagent/Resource | Reference or Source | Identifier or Catalog Number |
| --- | --- | --- |
| EGF | Qkine | Qk011 |
| | Miltenyi Biotec | 130-097-750 |
| CHIR99021 | Axon Medchem | 1386 |
| SB431542 | Axon Medchem | 1661 |
| Valproic acid | Merck Sigma Aldrich | P4543 |
| | Merck Sigma Aldrich | V0033000 |
| LPA | Merck Sigma Aldrich | L7260 |
| | Peprotech | 2256236 |
| | Tocris | 3854 |
| **Software** | | |
| QuantStudio™ 6&7 Flex | Applied Biosystems | |
| ZEN 2012 | Carl Zeiss MicroImaging, Inc. | |
| Cell Profiler | https://cellprofiler.org/ | |
| Fiji (from ImageJ) | National Institutes of Health | |
| FlowJo | BD Biosciences | |
| Spectronaut | Biognosys | |
| R | https://ropensci.org/ | |
| **Other** | | |
| Zeiss LSN700 | Carl Zeiss MicroImaging, Inc. | |
| Zeiss LSM 980 | Carl Zeiss MicroImaging, Inc. | |
| Axio Observer Z1 | Carl Zeiss MicroImaging, Inc. | |
| Nikon TiE A1R | Nikon Instruments Inc. | |
| Leica Stellaris 5 | Leica Microsystems | |
| Olympus IX83 | Olympus Corporation | |
| LSR Fortessa | BD Biosciences | |
| Symphony A5 | BD Biosciences | |
| FACSymphony S6 | BD Biosciences | |
| NextSeq500 | Illumina Inc. | |
| NextSeq2000 | Illumina Inc. | |

## Culture of hPSCs

Naive HPD06 and HPD03 hiPSCs were previously generated by direct reprogramming from somatic cells as described in Giulitti et al (2019) were cultured on mitotically inactivated mouse embryonic fibroblasts (MEFs) or serum coating in PXGL medium (Bredenkamp et al, 2019) at 37 °C, 5% $CO_2$, 5% $O_2$. The PXGL medium was prepared as follows: N2B27 (DMEM/F12 and Neurobasal in 1:1 ratio, with 1:200 N2 Supplement and 1:100 B27 Supplement, 2 mM L-glutamine, 0.1 mM 2-mercaptoethanol) supplemented with 1 µM PD0325901, 2 µM XAV939, 2 µM Gö6983 and 10 ng/ml human LIF (hLIF). Serum coating medium was composed of cold 10% Fetal Bovine Serum (FBS) in DMEM high

glucose. After adding 2 ml of serum coating solution to each well of a 6-well plate, plates were incubated overnight at 37 °C, rinsed once with PBS without $MgCl_2$/$CaCl_2$ (Merck Sigma Aldrich D8537) before plating the cells. Cells were passaged as single cells every 3–4 days at a split ratio of 1:3 or 1:4 following dissociation with TrypLE for 3 min at room temperature. A 10 µM ROCK inhibitor (Y-27632, ROCKi) was added to the naive medium for 24 h after passaging. For the feeder-free conversion, naive hiPSCs HPD06 and HPD03 stably cultured on MEFs were collected with ReLeSR (STEMCELL Technologies 100-0483). Cells were first incubated for 2 min at room temperature, followed by another 5 min at 37 °C after the complete aspiration of the reagent. Cells were collected by carefully washing the well with fresh PXGL medium with 10 µM ROCKi and were plated on feeder-free plates coated with serum at a 1:1 to 1:2 ratio.

Naive H9 (Thomson et al, 1998) and HNES1 (Guo et al, 2016) from the Leeb group, HNES1 GATA3::mKO2 (Guo et al, 2021) and Shef6 (Aflatoonian et al, 2010) hESCs were cultured on mitotically inactivated C57BL/6 or DR4 MEFs or serum coating in PXGL at 37 °C, 5% $CO_2$, 5% $O_2$. Cells were passaged every 3–4 days at a split ratio up to 1:12 using Accutase for 5 min at 37 °C. A 10 µM ROCKi was added to the naive medium for 24 h after passaging. Naive H9 and HNES1 hESCs from the Zylicz group were instead passaged at a split ratio up to 1:6 using TrypLE for 3 min at 37 °C.

Naive KOLF2.1 J (Pantazis et al, 2022) and SCTi003-A hiPSCs from the Rivron group were cultured on 0.1% gelatin-coated plates with a feeder layer of irradiated MEFs or serum coating in PXGL at 37 °C, 5% $CO_2$, 5% $O_2$. Cells were passaged every 3–4 days in a ratio of 1:6 following dissociation with Accutase for 5 min at 37 °C. 10 µM ROCKi was added for 24 h after passaging.

Naive H9 and HNES1 hESCs from the Pasque group, and SIG-1 hiPSCs were cultured on MEFs or serum coating in PXGL under hypoxic conditions in 5% $O_2$ and 5% $CO_2$ incubator under humidified conditions at 37 °C. Cells were passaged every 3 days at a split ratio up to 1:10 by single-cell dissociation with Accutase for 5 min at 37 °C. 10 µM ROCKi was added to the naive medium for 24 h after passaging.

Primed human KiPS (Takashima et al, 2014) were cultured on pre-coated plates with 0.5% growth factor-reduced Matrigel in E8 medium, made in-house according to Chen et al (2011), at 37 °C, 5% $CO_2$, 5% $O_2$. Cells were passaged every 3–4 days at a split ratio of 1:8 following dissociation with 0.5 mM EDTA (Invitrogen AM99260G) in PBS without $MgCl_2$/$CaCl_2$.

Primed H9 from the Leeb group and re-primed Shef6 hESCs were maintained feeder-free on plates pre-coated with 1% (v/v) Geltrex (Gibco A1413302) in E8 flex at 37 °C, 5% $CO_2$, 5% $O_2$. Following dissociation with Versene (Gibco 15040-033) at room temperature for 5 min, cells were passaged every 3–4 days at a split ratio of 1:6.

Primed H9 hESCs from the Pasque group were grown in pre-coated geltrex tissue culture under normoxic conditions and under humidified conditions at 37 °C in complete E8Flex medium. Cells were dissociated into clumps every 5–6 days by incubating for 5 min at room temperature in Accutase and passaged at a split ratio of 1:12. 10 µM ROCKi was added to the naive medium for 24 h after passaging.

All cell lines were routinely checked for mycoplasma and tested negative (Euroclone EMK090020 or Mycoplasma check Service from Eurofins).

All FBS batches tested and used to perform experiments in this study are listed in Table 3. Each new batch was tested as a coating in parallel with MEFs and/or previous working batches for morphology and proliferation rate retention, and stable expression or protein levels of general pluripotency and naive pluripotency markers over 4/5 passages.

## Genetic engineering of naive hPSCs

HPD06-GFP stable line was generated by incubating 1 μg of an EGFP-piggyBAC and 1 μg of pBase diluted in 250 μL of Opti-MEM (Gibco 31985062) with 6 μL of Lipofectamine 2000 (Thermo Fisher 11668027) diluted in an equal volume of Opti-MEM for 20 min. The mix was added to a suspension of $1 \times 10^5$ cells/cm$^2$ dissociated as previously reported and resuspended in PXGL with 10 μM ROCKi and plated directly into a single well of a 6-well plate. Medium was changed after 24 h, and antibiotic selection (150 μg/mL hygromycin) was applied 48 h post-transfection. Cells were long-term maintained as previously reported in selection.

## Capacitation of naive hPSCs

Naive hiPSCs HPD06 stably cultured on MEFs or serum coating were capacitated following the Rostovskaya et al (2019) protocol with minor modifications. Briefly, cells were dissociated with TrypLE, and $5 \times 10^4$ cells/cm$^2$ were seeded on a 6-well plate pre-coated with 0.5% growth factor-reduced Matrigel in PXGL medium with 10 μM ROCKi (day 0). From day 1 to day 6, cells were cultivated in N2B27 with 2 μM XAV939. From day 6 onward, the medium was changed to XAF (N2B27 supplemented with 2 μM XAV939, 20 ng/ml Activin A and 10 ng/ml FGF2).

Naive hESCs Shef6 and H9 stably cultured on MEFs or serum coating were capacitated following a modified version of the Rostovskaya et al (2019) protocol. Briefly, cells were dissociated with Accutase, and $1 \times 10^4$ cells/cm$^2$ or $1.5 \times 10^4$ cells/cm$^2$ were seeded on a 6-well plate pre-coated with Geltrex directly into N2B27 medium supplemented with 10 μM ROCKi (day 0). From day 1, cells were cultivated in N2B27 without ROCKi. For the additional conditions, the N2B27 was supplemented with 2 μM XAV939, 12 ng/mL FGF2 and/or 10 ng/mL Activin A or replaced with E8 or mTeSR.

## Conversion of human naive PSCs into trophectoderm/TSCs

Induction of trophectoderm identity in naive Shef6, HNES1, HNES1 GATA3::mKO2 and H9 hESCs was performed as previously described (Guo et al, 2021). In brief, cells were dissociated with Accutase and $2 \times 10^4$ cells/cm$^2$ were seeded on a 12-well plate pre-coated with Geltrex into PXGL with 10 μM ROCKi (day −1). From the next day (day 0), cells were switched to N2B27 supplemented with 1 μM PD0325901 and 1 μM A83-01 and fed daily.

For TSCs differentiation, naive HPD06 and HPD03 hiPSCs stably cultured on MEFs or serum coating were pre-treated on MEF with TS medium (DMEM/F12 supplemented with 0.1 mM 2-mercaptoethanol, 0.2% FBS, 0.3% Bovine serum albumin [BSA], 1% insulin-transferrin-selenium-ethanolamine-X [ITS-X], 1.5 μg/ml L-ascorbic acid, 50 ng/ml EGF, 2 μM CHIR99021, 0.5 μM A83-01,

1 μM SB431542, 0.8 mM Valproic acid [VPA], and 5 μM ROCKi) for 24 h. Cells were dissociated with TrypLE, and $5 \times 10^4$ cells/cm$^2$ were seeded on a 6-well plate pre-coated with 5 μg/mL Collagen IV and further cultured in TS medium. Cells were cultured in 5% CO$_2$ and 5% O$_2$, changing medium every 2 days, and were passaged upon 80% confluency at a 1:6 to 1:8 ratio.

Naive SIG-1 hiPSCs to trophoblast differentiation was done as previously published (Pham et al, 2022), using the following previously described protocols for hTSCs (Cinkornpumin et al, 2020; Dong et al, 2020; Guo et al, 2021; Io et al, 2021b). Cells were cultured on MEFs or serum-coated plates until subconfluency and dissociated into single cells using TrypLE for 5 min at 37 °C in PXGL supplemented with 10 mM Y-27632. At day 2 of culture, cells were washed with PBS (Gibco, 10010-015), and the media was switched from PXGL to ASECRiAV medium (Okae et al, 2018) consisting of DMEM/F12 supplemented with 0.3% BSA, 0.2% FBS, 1% ITS-X, 1.5 mg/mL L-ascorbic acid, 0.5 mM A83-01, 1 mM SB431542, 50 ng/ml EGF, 2 μM CHIR99021, 0.8 mM VPA, 0.1 mM 2-mercaptoethanol and 5 mM Y-27632. The medium was changed every day and supplemented with 5 mM Y-27632. From passage 1 onwards, hTSCs were cultured and maintained on 1.5 mg/mL iMatrix-silk-E8-Laminin overnight-coated plates in hypoxia conditions 5% O$_2$ and 5% CO$_2$ at 37 °C, and passaged every 5 days at a 1:3 splitting ratio.

Naive SIG-1 hiPSCs differentiation to trophectoderm was performed as previously described (Io et al, 2021a). Briefly, cells were dissociated in single cells using Accutase at 37 °C for 6 min, followed by gentle mechanical dissociation. $3 \times 10^4$ cells/cm$^2$ were plated onto iMatrix Silk laminin 511-coated plate (Amsbio AMS.892 012) in nTE1 media (N2B27 supplemented with 2 μM A83-01, 2 μM PD0325901, 10 ng/ml BMP4 [R&D Systems 314-BP] and 10 μM Y-27632). After 24 h, cells were washed with PBS, and the medium was switched to nTE2 for 48 h and refreshed daily (N2B27 supplemented with 2 μM A83-01, 2 μM PD0325901 and 1 μg/ml JAKi [Merck Sigma Aldrich 420099]). Cells were fixed on day 3.

## Culture of BAP-treated primed pluripotent stem cells

MEFs were fed with DMEM/Ham's F-12 medium containing 0.1 mM 2-mercaptoethanol, 1% ITS-X, 1% non-essential amino acids (NEAA; Gibco 11140050), 2 mM L-glutamine, and 20% KnockOut™ Serum Replacement (KSR; Gibco 10828028) for 24 h. The supernatant was collected and used as MEF-conditioned medium. KiPS were dissociated into single cells with TrypLE. The cells were cultured on pre-coated plates with 0.5% growth factor-reduced Matrigel at a density of $2 \times 10^4$ cells/cm$^2$ with MEF-conditioned medium supplemented with 10 ng/ml BMP4 (Peprotech 120-05ET), 1 μM A83-01, 0.1 μM PD173074 (Axon Medchem 1673), and 10 μM ROCKi, as described previously (Io et al, 2021b). The medium was changed daily.

## Spontaneous differentiation with embryoid bodies

Capacitated hiPSCs HPD06 from stable cultures on MEFs or serum coating were single-cell dissociated with TrypLE and seeded at $1.5 \times 10^4$ cells in v-bottom 96-well plates (Greiner, M9686) pre-treated with anti-adherence AggreWell™ Rinsing Solution (STEM-CELL Technologies 7010) in E8 medium with 10 μM ROCKi (day

0). From day 1, aggregates were cultivated in spontaneous differentiation medium (DMEM/F12 with 20% FBS, 2 mM L-glutamine, 1% NEAA and 0.1 mM 2-mercaptoethanol). Half of the medium was changed every 2 days.

## Blastoid formation

Blastoid experiments from naive H9 and HNES1 hESCs from the Zylicz group were performed according to Kagawa et al (2022), with minor modifications. Briefly, naive hESCs were collected by incubation for 3 min with Accutase. In the presence of MEFs, cells were incubated for at least 60 min on 0.1% gelatin-coated plates, after which the non-attached were collected. Single cells were plated at a density of 98–102 cells per micro-well of a 24-well AggreWell™400 plate in N2B27 supplemented with 10 µM ROCKi to aggregate for 3 h. Subsequently, half medium was changed with 2x PALLY consisting of N2B27 supplemented with 0.3% BSA and 2x concentration of the PALLY medium containing 1 µM PD0325901, 1 µM A83-01, 10 ng/mL hLIF, 1 µM oleoyl-L-α-lysophosphatidic acid sodium salt (LPA) and ROCKi. 24 h later, half-medium was changed with 1x PALLY, which was repeated 24 h later. After a total of 72 h of PALLY culture, blastoids were maintained for 48 h in LY medium consisting of N2B27 supplemented with LPA and ROCKi. Blastoids from H9 naive hESCs were processed for single-cell RNA sequencing at day 5.

Blastoid experiments from naive H9 hESCs from the Leeb group and KOLF2.1J and SCTi003-A hiPSCs were performed according to Kagawa et al (2022), with minor modifications. Naive KOLF2.1J and SCTi003-A hiPSCs were collected by incubation with Accutase for 5 min. In the presence of MEFs, cells were incubated for 70 min on 0.1% gelatin-coated plates, after which the non-attached hiPSCs were collected. Single cells were plated at a density of 20,000 cells per well of a 96-well plate comprising microwells of 200 µm diameter, made as previously described (Rivron et al, 2012). Cells were cultured in N2B27 supplemented with 0.3% BSA, 1 µM PD0325901, 1 µM A83-01, 10 ng/ml hLIF, 1 µM LPA and 1:1000 CEPT cocktail (Thermo Fisher A56799). After 24 h, the medium was changed to N2B27 supplemented with 1 µM PD0325901, 1 µM A83-01, 10 ng/ml LIF and 1 µM LPA for 48 h with daily medium changes. Blastoids were maintained for an additional 48 h in N2B27 medium supplemented with 1 µM LPA.

Blastoid experiments from naive HNES1 hESCs from the Pasque group were performed as previously described (Kagawa et al, 2022) in AggreWells™400 (STEMCELL Technologies 34415). Briefly, naive HNES1 hESCs were treated with Accutase at 37 °C for 6 min, followed by gentle mechanical dissociation. In the presence of MEFs, cells were incubated for 70 min on 0.1% gelatin-coated plates, after which the non-attached hESCs were collected. The cell pellet was resuspended in Aggregation medium composed of N2B27 with 0.3% BSA and 10 µM ROCKi. Single cells were counted and viability assessed via trypan blue staining using ThermoFisher Countess II, and $1.0/1.2 \times 10^5$ live cells/well (~80/100 cells/µwell) were seeded into AggreWells™400 treated with an anti-adherent solution (STEMCELL Technologies 07010) and washed 3 times (2x N2B27 and 1x aggregation medium) before cell plating (day −1). The cells were allowed to form aggregates inside the microwell for 24 h. On day 0, the aggregation medium was replaced with PALLY

medium: N2B27 with 0.3% BSA, 1 µM PD0325901, 1 µM A83-01, 0.5 µM LPA, 10 ng/ml hLIF and 10 µM ROCKi. PALLY medium was refreshed on day 1. After 48 h (day 2), PALLY medium was replaced with N2B27 containing 0.3% BSA, 0.5 µM LPA and 10 µM Y-27632 (LY medium). LY medium was refreshed on day 3, and blastoids fully formed on day 4.

## Proliferation assay and AP staining

Cell proliferation was assessed by plating $5 \times 10^4$ cells/cm² in a 6-well plate. Cells were single-cell dissociated with TrypLE, counted and re-plated every 3 days.

For AP staining, cells were fixed with a citrate–acetone–formaldehyde solution and stained using the Alkaline Phosphatase kit (Merck Sigma Aldrich 86 R-1KT). Plates were scanned using a Nikon Scanner and scored with the Analyse Particles plugin of the software Fiji (from ImageJ) 1.0 (pixels 5–500, circularity 0.5–1).

## Gene expression analysis by quantitative PCR with reverse transcription

Total RNA from naive hiPSCs HPD06 and HPD03 was isolated using Total RNA Purification Kit (Norgen Biotek 37500), and complementary DNA (cDNA) was made from 500 ng using M-MLV Reverse Transcriptase (Invitrogen 28025-013) and dN6 primers. Data were acquired with the QuantStudio™ 6&7 Flex Software 1.3 (Applied Biosystems).

Total RNA from naive hESCs H9 and Shef6 was isolated with the ExtractMe kit (Blirt EM15) following the manufacturer's instructions, and cDNA was made from 100 ng using the SensiFAST cDNA Synthesis Kit (Bioline BIO-65054). Real-time PCR was performed on the CFX384 Touch real-time PCR detection system (Bio-Rad).

SYBR Green Master mix (Bioline BIO-94020) was used for real-time qPCR, and primers are listed in Table EV1. Three technical replicates were carried out for each biological replicate of all RT-qPCR analyses, and *β-ACTIN* or *GAPDH* were used as endogenous controls to normalise expression.

## Immunofluorescence

Immunofluorescence analysis of naive hiPSCs HPD06 and HPD03 was performed on 1% Matrigel-coated glass coverslip in wells. Cells were fixed in 4% Formaldehyde (Merck Sigma Aldrich 78775) in PBS for 10 min at room temperature, washed in PBS, and permeabilised and blocked in PBS with 0.3% Triton X-100 (PBST) and 5% of horse serum (ThermoFisher 16050-122) for 1 h at room temperature. Cells were incubated overnight at 4 °C with primary antibodies in PBST with 3% horse serum. After washing with PBS, cells were incubated with secondary antibodies (Alexa, Life Technologies) for 45 min at room temperature. Nuclei were stained using Fluoroshield™ with DAPI (Merck Sigma Aldrich F6057). Images were acquired with a Zeiss LSN700 confocal microscope using ZEN 2012 software. Images were processed using the software Fiji (from ImageJ) 1.0. Fluorescence intensity was quantified using Cell Profiler software (v4.2.1) using identical conditions for all images. DAPI staining was used to identify individual nuclei of cells. At least 700 nuclei from five randomly

selected fields from two independent experiments were analysed for each condition. Single-cell measured mean intensity values were considered for downstream data analysis.

For immunofluorescence analysis of naive hESCs H9, HNES1 and Shef6 from the Leeb group, cells were plated onto glass coverslips (Marienfeld YX03.2) or µ-Slide 8 Well Glass Bottom Chamber Slides (Ibidi 80827) coated with Geltrex. Cells were fixed in 4% paraformaldehyde in PBS for 20 min at room temperature, washed in PBS, and permeabilised with 0.5% Triton X-100 in PBS for 10 min. After two washes with 0.2% Tween-20 in PBS (PBST), cells were incubated in PBST with 5% BSA for 1 h at room temperature. Cells were incubated overnight at 4 °C with primary antibodies diluted in PBST with 5% BSA. After washing with PBST, cells were incubated with secondary antibodies (Alexa, Life Technologies) diluted in PBST with 5% BSA for 1 h in the dark at room temperature. After washing twice with PBST, the nuclei were stained with 20 ng/mL DAPI (Invitrogen D1306) in PBST for 5 min in the dark at room temperature. Samples were mounted using Vectashield with DAPI (Vector Laboratories VECH-1200). Images were acquired with an Axio Observer Z1 microscope or a Zeiss LSM 980 with Airyscan 2 and processed using the software Fiji (from ImageJ) 1.0.

The immunofluorescence staining protocol was performed in naive H9 hESCs from the Pasque group and SIG-1 hiPSC as previously published (Pham et al, 2022; Pasque et al, 2014). Cells were grown on 0.1% gelatinised 18 mm round coverslips on MEFs or serum coating. Cells were fixed in 4% paraformaldehyde for 10 min at room temperature in the dark and permeabilised with 0.5% Triton X-100 in PBS for 5 min and washed twice with 0.2% Tween 20 in PBS (PBST) for 5 min each before proceeding to the staining. Primary and secondary antibodies were diluted in a blocking buffer containing mainly 2% Tween-PBS with 5% normal donkey serum and 0.2% fish skin gelatin. Cells on coverslips were incubated overnight at 4 °C with the specific primary antibodies in blocking solutions, then washed three times with PBST for 5 min. After washing, the samples were incubated with secondary antibodies diluted in blocking buffer for 1 h in the dark at room temperature. The samples were then washed with PBST three times for 5 min each and afterwards washed with 0.002% DAPI (Sigma-Aldrich D9542) solution in PBST. The coverslips were mounted in Prolong Gold reagent with DAPI after a final wash in PBST. Naive hPSCs immunofluorescence images were taken in a Nikon TiE A1R inverted microscope and were processed using the software Fiji (from ImageJ) 1.0.

Blastoids from naive H9 and HNES1 hESCs from the Zylicz group were fixed with 4% paraformaldehyde for 20 min at room temperature. Subsequently, the samples were washed 3 times for 10 min with PBST (PBS containing 0.1% Tween20) supplemented with BSA (0.3%). The samples were then permeabilised for 20 min using 0.2% Triton X-100 in PBS and afterwards blocked using a blocking buffer containing 0.1% Tween 20, 1% BSA and 10% normal donkey serum in PBS for at least 3 h. The samples were then incubated overnight at 4 °C with primary antibodies diluted in blocking buffer. The next day, samples were washed with PBST at least three times for 10 min each. After washing, the samples were incubated with secondary antibodies diluted in blocking buffer for 3 h in the dark at room temperature. The samples were then washed with PBST three times for 10 min each, stained with DAPI during the second wash, and afterwards mounted using PBS. Image acquisition was performed using a Leica Stellaris 5 fluorescence

confocal microscope and was processed using the software Fiji (from ImageJ) 1.0.

Blastoids from naive H9 hESCs from the Leeb group and KOLF2.1 J and SCTi003-A hiPSCs were fixed with 4% PFA for 30 min at room temperature and rinsed three times with PBS. Samples were permeabilised and blocked using 0.3% Triton X-100 and 10% normal donkey serum (Merck Sigma Aldrich S30) in PBS for at least 60 min and incubated overnight at room temperature with primary antibodies diluted in fresh blocking/permeabilisation buffer. The following day, samples were washed with PBS containing 0.1% Triton X-100 (PBST) at least three times for 10 min each. Samples were then incubated in Alexa Fluor-tagged secondary antibodies (Abcam or Thermofisher Scientific) diluted in PBST for at least 30 min in the dark at room temperature and washed three times with PBST for 10 min each. Images were captured using an Olympus IX83 confocal microscope. Each image consisted of 30 optical sections, with an average thickness of approximately 3–5 µm per section. Images were processed using the software Fiji (from ImageJ) 1.0.

Blastoids from naive HNES1 hESCs from the Pasque group were fixed with 4% PFA for 30 min at room temperature and rinsed three times with PBS. About 10–20 structures per condition were then moved to Thermo Scientific™ Nunc™ MiniTrays with Nunclon™ Delta surface and permeabilised for 30 min at RT using PBS/0.3% Triton X-100 (Merck Sigma Aldrich). Cells were blocked for 4–6 h at RT with blocking solution (PBS 0.3% Triton X-100 with 10% Normal Donkey Serum [Merck Sigma Aldrich S30]). Primary antibodies were diluted in blocking solution and incubated overnight at 4 °C in a humidified chamber. Cells were washed three times with PBS 0.1% Triton X-100, and stained with secondary antibodies for 1 h at RT in a humidified chamber. Cells were washed three times with PBS 0.1% Triton X-100, and stained with DAPI for 15 min at RT during the second wash (0.1 µg/ml DAPI in PBS 0.1% Triton X-100). Finally, blastoids were moved into IBIDI µ-Slide 15 Well 3D Glass Bottom (81507) and imaged using a C2 or TiE A1R confocal microscope (Nikon). Images were processed using the software Fiji (from ImageJ) 1.0, Z-stacks were shown as Max Intensity projections and denoised using the "despeckle" tool equally on all images.

All used primary antibodies are listed in Table EV2.

## Flow cytometry

For flow cytometry analysis of naive hESCs H9 and HNES1 from the Leeb group, and HNES1 GATA3::mKO and Shef6 during capacitation or trophectoderm induction, cells were detached as reported and resuspended in DMEM supplemented with 0.5% BSA. For surface marker staining, the cells were washed 1x in FACS buffer (1x PBS and 1% BSA), stained with the fluorophore-conjugated antibodies diluted in FACS buffer for 30 min on ice in the dark and washed again 1x with FACS buffer. For live/dead discrimination, 5 µg/mL DAPI (Invitrogen D1306) was used. Antibody and reporter signal levels were measured using the LSR Fortessa (BD Biosciences) or the ZE5 (Bio-Rad) and then analysed with the FlowJo software (v10.10, BD Biosciences).

Naive H9 hESCs from the Pasque group and SIG-1 hiPSCs were first stained for viability in 1:300 Live/Death Zombie Aqua UV in PBS. Fluorophore-conjugated antibody was diluted at a ratio of 1:50 in fluorescence-activated cell sorting (FACS) buffer with PE-

conjugated anti-human SUSD2 (Biolegend, 327406) for 20 min at room temperature, then washed with FACS buffer, and fixed in 4% paraformaldehyde. Cells were passed through a 40 μm cell strainer (Corning 352340). The fluorescence intensity of 20,000 cells was recorded on a flow cytometer Symphony A5 (BD Biosciences) and analysed using FlowJo software (v10.10, BD Biosciences). Single-stained controls were used for compensation and gating in the flow cytometer. Only live cells were used for data analysis. The frequency of the parent population has been used to plot the SUSD2-positive cells.

## Proteomics

Samples were processed using the PreOmics iST sample preparation kit (PreOmics P.O. 00027). LC-MS/MS analysis consisted of a NanoLC 1200 coupled via a nano-electrospray ionisation source to the quadrupole-based Q Exactive HF benchtop mass spectrometer (Michalski et al, 2011). For the chromatographic separation, a binary buffer system consisting of solution A: 0.1% formic acid, and B: 80% acetonitrile, 0.1% formic acid was used. Peptides were separated according to their hydrophobicity on an analytical column (75 μm) in-house packed with C18-AQ 1.9 μm C18 resin with a gradient of 7–32% solvent B in 45 min, 32–45% B in 5 min, 45–95% B in 3 min, 95–5% B in 5 min at a flow rate of 300 nl/min. MS data acquisition was performed in DIA (Data Independent Acquisition) mode using 32 variable windows covering a mass range of 300–1650 $m/z$. The resolution was set to 60,000 for MS1 and 30,000 for MS2. The AGC was 3e6 in both MS1 and MS2, with a maximum injection time of 60 ms in MS1 and 54 ms in MS2. NCE was set to 25%, 27.5%, and 30%. All acquired raw files were processed using Spectronaut software (17.0). For protein assignment, spectra were correlated with the Rattus Norvegicus/Bos Taurus database (v. 2023). Searches were performed with tryptic specifications and default settings for mass tolerances for both MS and MS/MS spectra.

The other parameters were set as follows. Fixed modifications: Carbamidomethyl (C); variable modifications: Oxidation, Acetyl (N-term); digestion: Trypsin, Lys-C; min. peptide length = 7 Da; max. peptide mass = 470 Da; false discovery rate for proteins and peptide-spectrum = 1%. Perseus software (1.6.2.3) was used to logarithmise, group and filter the protein abundance.

The obtained list of proteins was analysed using the Matrisome AnalyseR tool (https://sites.google.com/uic.edu/matrisome/tools/matrisome-analyzer (Petrov et al, 2023)) from The Matrisome Project.

## Exome sequencing

NEGEDIA S.r.l. performed the Exome Sequencing service. Genomic DNA was extracted with a paramagnetic bead technology using the Negedia DNA extraction service, quantified using the Qubit 4.0 fluorimetric Assay (Thermo Fisher Scientific) and sample integrity, based on the DIN (DNA integrity number), was assessed using a Genomic DNA ScreenTape assay on TapeStation 4200 (Agilent Technologies). Libraries were prepared from 100 ng of total DNA using the NEGEDIA Exome sequencing service, which included library preparation, target enrichment using Agilent V8 probe set, quality assessment with FastQC v0.11.9 (https://www.bioinformatics.babraham.ac.uk/projects/fastqc/) and sequencing on a NovaSeq

6000 system using a paired-end, $2 \times 150$ cycle strategy (Illumina Inc.). The raw data were analysed by NEGEDIA Exome pipeline (v1.0), which involves alignment to the reference genome (hg38, GCA_000001405.15), removal of duplicate reads and variant calling with Sentieon 202308 (Aldana and Freed, 2022). Variants were annotated by the Ensembl Variant Effect Predictor (VEP) tool3 (v. 104) (McLaren et al, 2016). Annotated variants were selected based on: impact on protein High/Moderate, base depth $\geq$30, alternate depth $\geq$5 and alt/base depth $\geq$5%. Selected variants were finally classified with Varsome Clinical (https://eu.clinical.varsome.com/).

## Bulk RNA sequencing and analysis

Total RNA from HPD06 and HPD03 naive hiPSCs was isolated as previously described, and Quant Seq 3' mRNA-seq Library Prep kit (Lexogen) was used for library construction. Before sequencing, library quantification was performed by fluorometer (Qubit) and bioanalyser (Agilent). Sequencing was performed on NextSeq500 ILLUMINA instruments to produce 5 million reads (75 bp SE) for each sample. For the comparison with published datasets, transcript quantification was performed from raw reads using Salmon (v1.6.0) (Patro et al, 2017) on transcripts defined in Ensembl 106. Gene expression levels were estimated with the tximport R package (v1.26.1) (Soneson et al, 2016). Batch correction was performed using the ComBat_seq function from the sva R package (v3.50.0). Batches have been defined following library preparation: batch 1 for full-length libraries and batch 2 for Quant Seq 3' mRNA-seq Library Prep kit. Counts-per-million (CPM) on batch corrected counts were computed using the CPM function of the edgeR package (v4.0.12) (Robinson et al, 2010). PCA was performed using the svd R function on log-transformed CPM. All plots except the heatmaps have been done using ggplot2 (v3.5.1). Heatmap was done using the pheatmap R package (v1.0.12). All analyses were performed using R v4.3.2.

For the analysis of MEFs and serum differentially expressed genes (DEGs), the reads were trimmed using BBDuk (BBMap v. 37.87), with parameters indicated in the Lexogen data analysis protocol. Trimmed reads were aligned to the Homo sapiens genome (GRCh38.p13) or the mouse genome (GRCm38.p6) using STAR (v. 2.7.6a), gene expression levels were quantified using featureCounts (v. 2.0.1) and quantified genes with total number of counts above 10 in at least 2 samples were considered. Differential expression analysis was performed using the DESeq2 R package (v. 1.28.1). Transcripts with an absolute value of log2 Fold Change | log2FC| > 1 and an adjusted $P$-value < 0.05 (Benjamini–Hochberg adjustment) were considered significant and defined as DEGs. All RNA-seq analyses were carried out in the R environment (v. 4.0.0) with Bioconductor (v. 3.7). Volcano plots were computed with log2FC and $-\log10$ adjusted $P$-value from DESeq2 differential expression analysis output using the ggscatter function from the ggpubr R package (v. 0.4.0).

Imprinted genes analysis of HPD00 primed and HPD06 and HPD03 naive hiPSCs has been performed by submitting the .fastq raw files to BrewerIX (https://brewerix.bio.unipd.it) (Martini et al, 2022), implementing the complete pipeline and evaluating the significant genes.

Total RNA from naive hESCs Shef6 was isolated as previously described. Library preparation with the Quant Seq 3' mRNA-seq Library Prep kit (Lexogen) and sequencing with the Illumina

NextSeq2000 P3 platform were carried out at the VBCF NGS facility, producing 5–10 million reads (50 bp SE). For analysis of the fastq files, the Nextflow 23.04.1.5866/nf-core/rnaseq v3.10.1 pipeline was used. This included quality control with fastQC (v0.11.9), pseudo-alignment to the human reference genome hg38 with Salmon (v1.9.0) and alignment with STAR (v2.7.10a). Gene expression levels were estimated with the tximport R package (v1.26.1) (Soneson et al, 2016). DESeq2 (v1.38.3) was used for further analysis, including differential expression analyses. For comparison with already published data (Rostovskaya et al, 2019), the batch correction was performed using the ComBat_seq function from the sva R package (v3.46.0). For PCAs, counts were transformed with the regularised log transformation (rlog) function (for own time course) or the variance-stabilising transformation (vst) function (integration with Rostovskaya data) from DESeq2. All plots except heatmaps have been done using ggplot2 v3.5.1. Heatmaps were done with the pheatmap R package (v1.0.12) using row-wise z-transformed DESeq2-normalised counts. All analyses were performed using R v4.3.2.

## Single-cell RNA sequencing, analysis and comparison with bulk RNA sequencing

For the analyses of the pluripotency exit, the human embryogenesis reference dataset (Zhao et al, 2025) was derived by the integration of six published human datasets (Yan et al, 2013; Petropoulos et al, 2016; Xiang et al, 2020; Tyser et al, 2021; Yanagida et al, 2021; Meistermann et al, 2021) covering developmental stages from the zygote to the gastrula was kindly provided by the Petropoulos lab. Analysis was performed using Seurat v5.1.0, and the data were subset for embryonic cells covering E5-E14. For the PCA, pseudobulk data were generated with Seurat's AggregateExpression function and integrated with the bulk RNA-seq data using DESeq2 v1.38.3. Batch correction for the datasets was performed using the ComBat_seq function before applying variance stabilising transformation with the vst function from DESeq2. Heatmaps and boxplots were generated using the Shiny app (http://petropoulos-lanner-labs.clintec.ki.se) from the reference dataset (Zhao et al, 2025).

For blastoids validation, blastoids were dissociated into a single cell suspension by incubation with a mixture of TrypLE and Accutase. The sample was incubated with 0.5 µg of unique hashtag antibody for 20 min on ice. TotalSeq hashtag antibodies (Biolegend) were used to multiplex the samples (Hansen et al, 2023). The sample was sorted on a BD FACSymphony S6 (BD Biosciences) and then loaded onto a Chromium Next GEM chip (10x Genomics). Further steps of library preparation were performed according to the Chromium Next GEM Single Cell 3' v3.1 user guide, with the addition of the hashtag library for demultiplexing. Combined libraries were sequenced using the NextSeq2000 P2-100 kit (Illumina) with Paired-end sequencing. Initial processing of scRNA-seq data was performed using Cell Ranger (v6.1.2, 10X Genomics). Dual-indexed RNA and single-indexed hashtag oligo (HTO) libraries were processed in separate instances of cellranger mkfastq, using Illumina's bcl2fastq (v2.20.0.422). FASTQ files were aligned to the human reference genome (GRCh38, v 2020-A as provided by 10x Genomics) with cellranger multi (expect-cells 16500, min-assignment-confidence 0.9), assigning 4251 cells to the sample of interest. The resulting filtered feature-barcode matrix was loaded into Seurat (v 4.3.0) (Hao et al, 2021), excluding features

that were detected in less than three cells. Next, RNA data were normalised with the default LogNormalize method, and the HTO assay was normalised with centered log-ratio (CLR) transformation. Based on QC plots, cells with more than 15% mitochondrial counts or less than 7000 UMIs (nCount_RNA) were removed, retaining 3260 high-quality cells. These were subjected to standard Seurat processing using mostly default parameters unless indicated: FindVariableFeatures, ScaleData, RunPCA, RunUMAP (dims=1:15), FindNeighbors (dims=1:15), FindClusters (resolution=0.5). To characterise the cells' cycling status, the function CellCycleScoring was run using the built-in lists of S and G2M phase markers (cc.genes.updated.2019) derived from Tirosh et al (2016). Cells separated by cell cycle phase on the UMAP, and this source of heterogeneity was then regressed out using ScaleData (vars.to.regress = c("S.Score", "G2M.Score"), features = rownames(object)), followed by RunPCA, RunUMAP(dims=1:15), FindNeighbors(dims=1:15), and FindClusters(resolution=0.5), resulting in 5 clusters. The raw counts of the quality-filtered matrix were uploaded to the Early Embryogenesis Projection Tool (v2.1.1) (https://petropoulos-lanner-labs.clintec.ki.se/shinys/app/ShinyEmbryoProjP, accessed on 2024-09-26) to project the cells on an integrated reference UMAP of human embryo development and get annotation of predicted cell identities (Zhao et al, 2025).

## Statistics and reproducibility

All RT-qPCR experiments were performed in three technical replicates. For each dataset, sample size $n$ refers to the number of biological or technical replicates, shown as dots and stated in the figure legends. All error bars indicate the standard error of the mean (SEM). Significance has been reported as $p$-values, calculated using the test reported in the figure legends.

## Ethics declarations

Our research complies with all relevant ethical regulations, including ISSCR guidelines.

HPD06 and HPD03 naive hiPSCs used in G.M.'s laboratory were checked by the European ethics committee and registered in the human pluripotent stem cells registry (link: https://hpscreg.eu/cell-line/UNIPDi004-B). Experiments with hPSCs and blastoids in J.J.Z.'s laboratory were approved by the Scientific Ethics Committee for Hovedstaden (H-21043866/94634 and H-24048289). Shef6 line use is with the agreement of the Steering Committee of the UK Stem Cell Bank (SCSC16-09). The WiCell line H9 (WA09) was used under the agreements 23-W0460 (JJZ) and 21-W0002 (ML). Work with human embryonic and induced pluripotent stem cells, including blastoids in the V.P. Laboratory was approved by the UZ/KU Leuven ethics committee (S64962, S66595 and S68981). Work with mouse embryonic fibroblast was approved by the UZ/KU Leuven ethics committee (P170/2019). The Austrian Academy of Sciences (the local ethical body) has given N.R.'s laboratory a license to perform blastoid experiments, following expert legal advice that concluded these are not in conflict with Austrian laws. This license is in the shape of a statement of the Commission for Science Ethics of the Austrian Academy of Sciences concerning the project 'Modeling human early development using stem cells'. There is no approval number on that document. This license conforms to the ethical standards suggested by the International

Society for Stem Cell Research (ISSCR). This work did not exceed a developmental stage normally associated with 14 consecutive days in culture after fertilisation, nor did it entail any implantation in vivo.

All collaborators of this study have fulfilled the criteria for authorship and have been included as authors, as their participation was essential for the design and implementation of the study. Roles and responsibilities were agreed among collaborators ahead of the research. This research was not severely restricted or prohibited in the setting of the researchers and does not result in stigmatisation, incrimination, discrimination or personal risk to participants.

## Data availability

This study did not generate any unique reagents and does not report original code. Bulk and single-cell RNA-seq data for this study have been deposited in the Gene Expression Omnibus (GEO) database under the accession code GSE284370. We also included available RNA-Seq data for naive and primed hPSCs, hTSCs and fibroblasts from GSE110377 (Giulitti et al, 2019), GSE63577 (Marthandan et al, 2015), GSE93226 (Chen et al, 2017), GSE75868 (Theunissen et al, 2016), GSE184562 (Zorzan et al, 2023), GSE138688 (Dong et al, 2020), GSE178162 (Jang et al, 2022), GSE135695 (Wei et al, 2021), PRJNA397941 (Liu et al, 2017), GSE133630 (Zorzan et al, 2020), GSE73211 (Choi et al, 2015), PRJEB7132 (Takashima et al, 2014), GSE150772 (Bayerl et al, 2021), E-MTAB-5674 (Guo et al, 2017); for capacitation from GSE123055 (Rostovskaya et al, 2019); for embryonic development from the human embryogenesis reference dataset (Zhao et al, 2025), comprehensive of GSE36552 (Yan et al, 2013), E-MTAB-3929 (Petropoulos et al, 2016), GSE136447 (Xiang et al, 2020), E-MTAB-9388 (Tyser et al, 2021), GSE171820 (Yanagida et al, 2021), PRJEB30442 (Meistermann et al, 2021). Proteomics data have been deposited in the PRIDE Proteomics database under the accession codes PXD059820 and PXD072053. Any additional information required to reanalyse the data reported in this paper is available from the lead contact upon request.

The source data of this paper are collected in the following database record: biostudies:S-SCDT-10_1038-S44318-026-00714-2.

## Peer review information

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

## Acknowledgements

We thank the KU Leuven FACS Core team for supporting flow cytometry experiments, Alejandra Purcell from the Gorgas Institute for FACS analysis, and the VIB BioImaging Core Leuven for their contribution to imaging performed in VP's lab. We thank the Vienna Biocenter Core Facilities (VBCF) for support in NGS analysis, and the Max Perutz Labs FACS and imaging core facilities for experimental support. JJZ's team thanks Antar Drews and Sandra Bages Arnal for their input, the reNEW platforms for technical expertise, support and use of equipment in particular: H. Wollmann, M. Michaut, J. Bulkescher, G. Dela Cruz and A. Kalvisa. GM's laboratory is supported by grants from: Giovanni Armenise–Harvard Foundation, Telethon Foundation (GJC21157), European Research Council Starting Grant (716910 - MetEpiStem), Italian national project 'PRIN 2022' (2022RA8E3T) and La Caixa Foundation (LCF/PR/HR24/52440015). GR is supported by a HORIZON MSCA Postdoctoral Fellowship (101108873 - PLURImet). ML's laboratory is supported by grants from the ML's laboratory is supported by grants from the Austrian Science Fund (FWF; 10.55776/PAT3246524, 10.55776/I5958). MO is a member of the FWF-funded doctoral programme 'Signalling Molecules in Cellular Homeostasis' (SMICH; 10.55776/W1261), and ML is a faculty member and speaker of SMICH and a 'Wiener Wissenschafts- Forschungs- undTechnologiefonds (WWTF)' Vienna Research Group Leader (VRG14-006). VP's laboratory is supported by the Research Foundation-Flanders (FWO grants G0C9320N and G0B4420N to VP), KU Leuven Research Fund (C1 grant C14/21/119 to VP), and Pandarome project 40007487 (G0I7822N) (funded by the FWO and F.R.S.-FNRS) under the Excellence of Science (EOS) program. MAS is supported by the Gorgas Memorial Institute for Health Studies and Fundación Sus Buenos Vecinos in Panama. SSFAvK (11I1523N) and TXAP (11N3122N) are supported by FWO PhD fellowships. JJZ's laboratory is supported by grants from: Novo Nordisk Fonden (NNF21CC0073729), Lundbeckfonden (R345-2020-14979), Danmarks Frie Forskningsfond (0169-00031B) and the European Research Council Starting Grant (101077271 - ChroMeta). NR's laboratory is supported by the European Research Council (ERC) under the European Union's Horizon 2020 research and innovation programme (ERC-Co grant agreement no. 101002317, "BLASTOID: a discovery platform for early human embryogenesis"). DC is supported by Fondazione Telethon Core Grant, Italian Ministry of Health (Piano Operativo Salute Traiettoria 3, T3-AN-09, "Genomed"; Ricerca Finalizzata 2021, "genOMICA"; MCNT2 2023, "EUCARDIS"), Italian Ministry of University and

Research and European Union (Next Generation EU - MUR-PRIN-2022, Project PNC 0000001 D3 4 Health). PG is supported by Fondazione Telethon Core Grant, AIRC (MFAG 2020), PNRR (PNRR-MR1-2022-12376821), PRIN-2022-PNRR (P2022JLNZ7), PRIN (20224FL9T5). PM is supported by the Italian national project 'PRIN 2022' (2022LJZRBY).

## Author contributions

**Giada Rossignoli**: Conceptualization; Data curation; Formal analysis; Supervision; Validation; Investigation; Visualization; Methodology; Writing—original draft; Project administration. **Michael Oberhuemer**: Data curation; Formal analysis; Investigation; Visualization; Methodology; Writing—original draft. **Ida Sophie Brun**: Data curation; Formal analysis; Investigation; Writing—review and editing. **Irene Zorzan**: Investigation; Methodology; Writing—review and editing. **Anna Osnato**: Resources; Investigation. **Anne Wenzel**: Data curation; Formal analysis; Visualization. **Emiel van Genderen**: Resources; Investigation. **Andrea Drusin**: Investigation. **Giorgia Panebianco**: Investigation. **Nicolò Magri**: Investigation. **Moritz Becker**: Investigation. **Mairim Alexandra Solis**: Investigation. **Chiara Colantuono**: Data curation; Formal analysis. **Sam Samuël Franciscus Allegonda van Knippenberg**: Investigation. **Thi Xuan Ai Pham**: Investigation. **Sherif Khodeer**: Investigation. **Paolo Grumati**: Data curation; Formal analysis. **Davide Cacchiarelli**: Data curation; Formal analysis. **Paolo Martini**: Data curation; Formal analysis. **Nicolas Rivron**: Supervision; Funding acquisition; Writing—review and editing. **Vincent Pasque**: Supervision; Funding acquisition; Writing—review and editing. **Jan Jakub Żylicz**: Supervision; Funding acquisition; Visualization; Writing—original draft. **Martin Leeb**: Supervision; Funding acquisition; Writing—original draft. **Graziano Martello**: Conceptualization; Supervision; Funding acquisition; Writing—original draft; Project administration.

Source data underlying figure panels in this paper may have individual authorship assigned. Where available, figure panel/source data authorship is listed in the following database record: biostudies:S-SCDT-10_1038-S44318-026-00714-2.

## Disclosure and competing interests statement

Davide Cacchiarelli is founder, shareholder, and consultant of NEGEDIA S.r.l. Chiara Colantuono is an employee of NEGEDIA S.r.l.

# Expanded View Figures

**Figure EV1. Naive hPSCs on serum coating express key markers and can be genetically engineered.**

(A) Top: Immunostaining for general pluripotency (OCT4 and NANOG) and naive (KLF17 and TFCP2L1) markers of naive HPD06 and HPD03 hiPSCs cultured on MEFs or serum coating at the 4th passage. Complete view of Fig. 1B. Scale bars: 100 μm. Representative images of two independent experiments are shown. Bottom: Mean fluorescence intensity quantification for general pluripotency (OCT4 and NANOG) and naive (KLF17 and TFCP2L1) markers of naive HPD06 and HPD03 hiPSCs cultured on MEFs or serum coating at the 4th passage. At least 700 nuclei from five randomly selected fields from two independent experiments were analysed for each cell line under different conditions. The box plot indicates the 25th, 50th and 75th percentiles. Two-sided unpaired Student's t-test of the means of independent experiments. (B) Immunostaining for general pluripotency (OCT4 and NANOG) and naive (KLF17 and SUSD2) markers of naive H9 and Shef6 hESCs, and SIG-1 hiPSCs stably cultured on MEFs or serum coating. Complete view of Fig. 1B. Scale bars: 100 μm. Representative images of two independent experiments are shown. (C) Gene expression analysis by RT-qPCR of general (*POU5F1* and *NANOG*), naive (*TFCP2L1*, *KLF4* and *KLF17*), and primed (*OTX2* and *ZIC2*) pluripotency markers in naive HPD06 and HPD03 hiPSCs on MEFs or serum coating at the 4th passage when plated at a low density. Bars indicate the mean ± SEM of technical replicates shown as dots from $n = 4$ independent experiments for primed hiPSCs. Technical replicates from $n = 2$ independent experiments for naive hPSCs are shown as dots. Two-sided unpaired Student's t-test. (D) Representative gating strategy to evaluate marker positivity in naive hPSCs by flow cytometry. Selected sub-populations are shown from left to right. First, the cell population was distinguished from cell debris (left panel). Singlets were chosen from the cell population, and live cells among singlets were selected, followed by the gating of marker-positive cells using the unstained negative control (right panel). This gating strategy corresponds to Figs. 1D and 5C, and EV4F. (E) Growth rate of naive HPD06 and HPD03 hiPSCs cultured on MEFs or serum coating when plated at a low density over the first 4 passages of the conversion. Bars indicate the mean ± SEM of technical replicates shown as dots from $n = 2$ independent experiments. Two-way repeated measures ANOVA. (F) Top: Representative AP staining images after clonal assay of naive HPD06 and HPD03 hiPSCs cultured on MEFs or serum coating when plated at a low density at the 4th passage. Bottom: Quantification of the relative number of AP-positive pluripotent colonies counted per well. Technical replicates from $n = 2$ independent experiments are shown as dots. Two-sided unpaired Student's t-test. (G) Morphologies (top) and fluorescence (bottom) of naive HPD06 hiPSCs transfected with an EGFP-piggyBAC 24 h post-transfection (left) and after selection and stable culture (bottom). Scale bars: 200 μm. Representative images of two independent experiments are shown. Source data are available online for this figure.

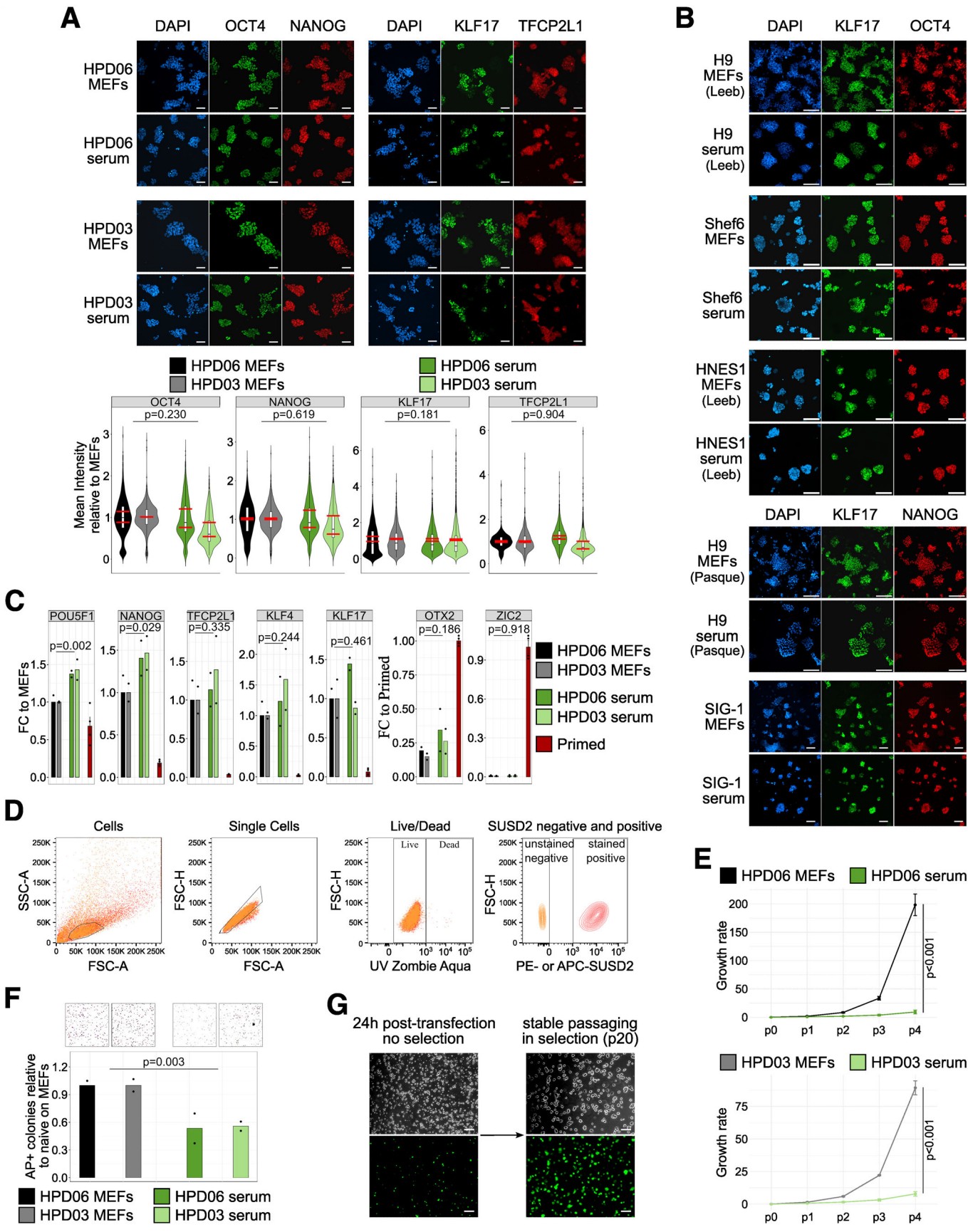

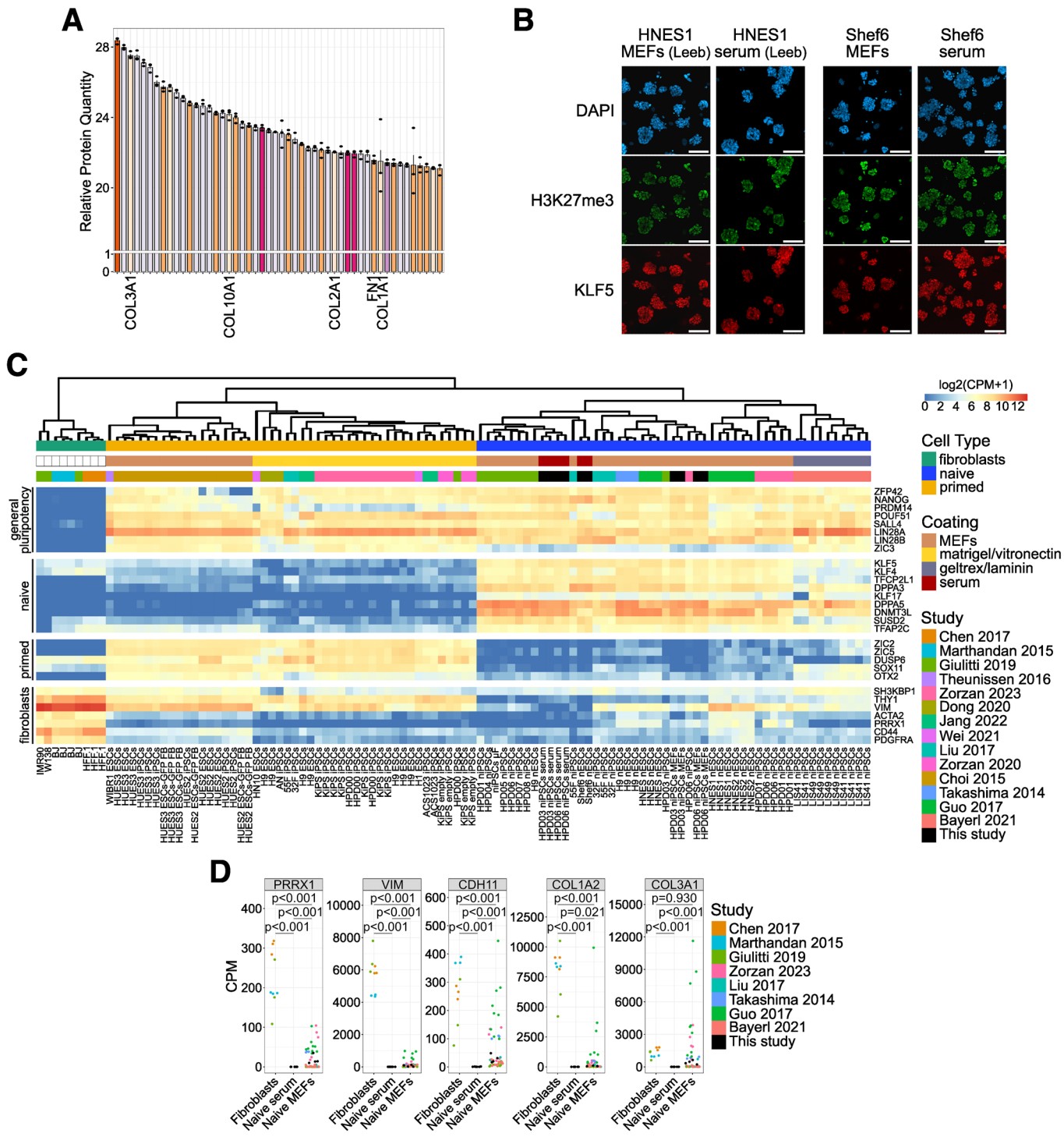

Figure EV2. Naive hPSCs on serum coating retain H3K27me3 marks and do not show contamination of MEFs transcripts.

(A) Top 50 most abundant proteins in one representative batch of serum coating. Detected Collagens and Vitronectin shared between 5 different serum coating batches are highlighted. Bars indicate the mean ± SEM of $n = 3$ technical replicates shown as dots. (B) Immunostaining for H3K27me3 and KLF5 of naive HNES1 and Shef6 hESCs stably cultured on MEFs or serum coating. Scale bars: 100 μm. Representative images of two experiments are shown. (C) Heatmap of general pluripotency, naive, primed, and fibroblasts genes in naive HPD06 and HPD03 hiPSCs lines and Shef6 hESCs stably cultured on MEFs or serum coating and in published fibroblasts, primed hPSCs and naive hPSCs. Extended version of Fig. 3D. (D) Absolute expression (CPM) of fibroblasts markers from Fig. 3D and others in naive HPD06 and HPD03 hiPSCs, and Shef6 hESCs stably cultured on MEFs or serum coating and in published naive hPSCs and fibroblast lines. Source data are available online for this figure.

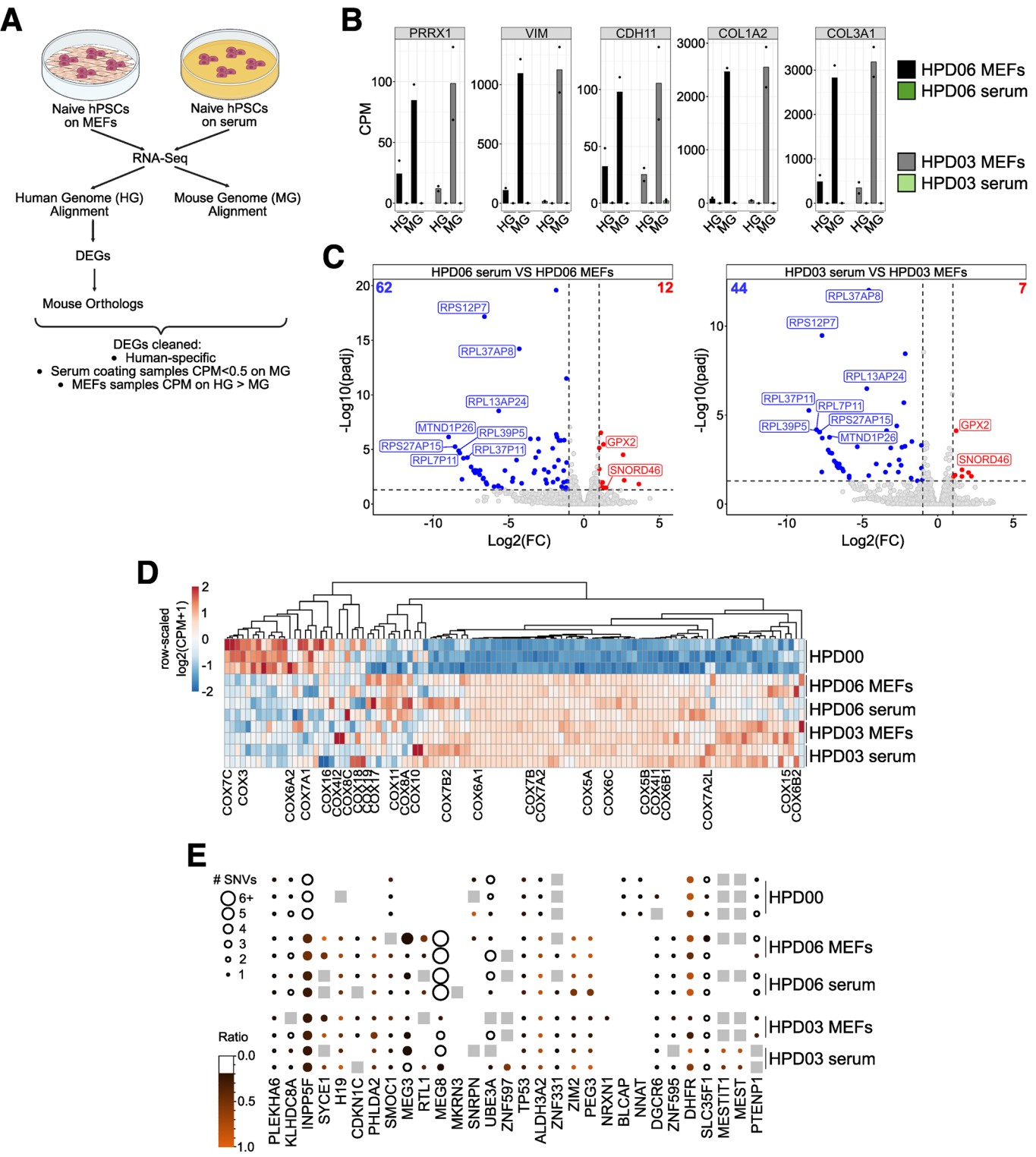

◄ **Figure EV3.   Naive hPSCs on serum coating do not show expression of MEFs genes and retain the expression of OXPHOS and imprinted genes.**

(A) Schematic representation of the pipeline followed for the evaluation and cleaning of MEFs DEGs between naive hPSCs stably cultured on serum coating and on MEFs.
(B) Absolute expression (CPM) of some fibroblasts markers from Fig. 3D and other DEGs with similar behaviour identified between naive HPD06 and HPD03 hiPSCs stably cultured on serum coating and on MEFs. For each gene, expression is reported for each cell line and condition aligned against the human or mouse genome. (C) Volcano plot representing DEGs (|log2FC| > 1 and an adjusted *P*-value < 0.05, Benjamini–Hochberg adjustment, as indicated by dashed lines) between naive HPD06 (top) and HPD03 (bottom) hiPSCs stably cultured on MEFs and serum coating. Blue and red represent down- and up-regulated DEGs, respectively. Labels highlight the top 8 most significant shared down-regulated DEGs and the only two shared up-regulated DEGs between cell lines. (D) Heatmap of oxidative phosphorylation genes from the KEGG PATHWAY Database (https://www.genome.jp/kegg/pathway.html) and COX genes (Takashima et al, 2014) in primed HPD00 hiPSCs and naive HPD06 and HPD03 hiPSCs stably cultured on MEFs or serum coating. (E) BrewerIX gene summary panel results on RNA-Seq data from primed HPD00 hiPSCs and naive HPD06 and HPD03 hiPSCs stably cultured on MEFs or serum coating. Empty dots indicate detected genes with no evidence of biallelic expression, grey squares indicate genes detected but not reaching the thresholds, and the absence of any symbol indicates that the gene was not detected. Source data are available online for this figure.

    

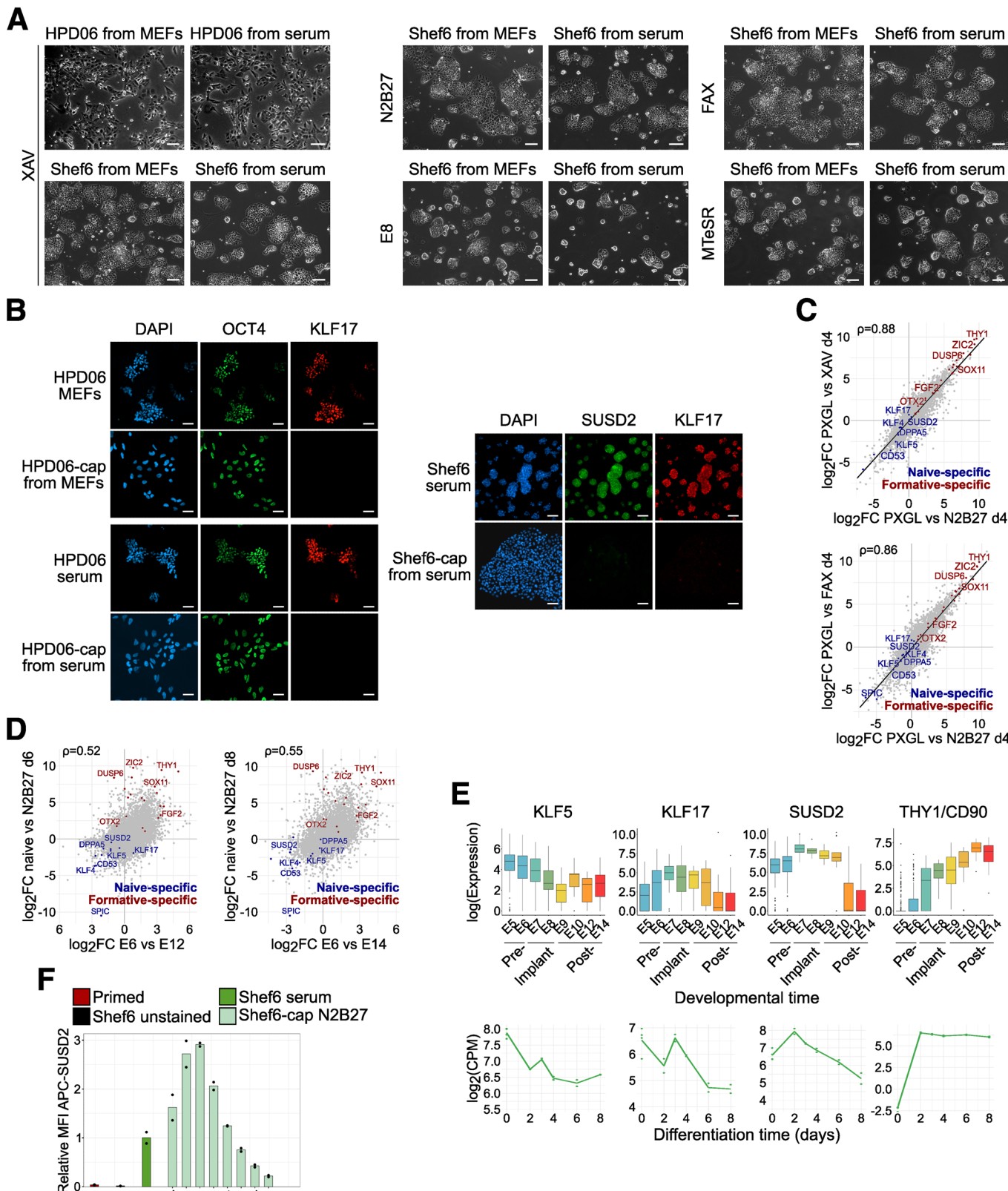

◀  **Figure EV4.  Feeder-free naïve hPSCs can efficiently exit naïve pluripotency.**

(A) Morphologies of HPD06 hiPSCs and Shef6 hESCs capacitated for 10 days from naive lines stably cultured on MEFs or serum coating under different media compositions. Scale bars: 100 μm. Representative images of two independent experiments are shown. (B) Immunostaining for general (OCT4) and naive (KLF17 and SUSD2) pluripotency markers of HPD06 hiPSCs and Shef6 hESCs capacitated for 10 days from naive lines stably cultured on MEFs or serum coating. Scale bars: 100 μm. Representative images of two independent experiments are shown. (C) Scatter Plots showing log2 fold changes (log2FC) of naive Shef6 hESCs stably cultured on serum coating differentiated for 4 days in unsupplemented N2B27, N2B27 supplemented with XAV, or N2B27 supplemented with XAV, FGF2, and Activin A (FAX) compared to the naive state. Technical replicates include $n = 6$ from two independent experiments for d0, $n = 4$ from two independent experiments for N2B27 at day 4, and $n = 2$ for all other combinations of conditions and time points. Data was filtered for the 9376 DEGs ($|log2FC| > 1$, padj < 0.05) identified in any comparison with the naive state. Selected naive- and primed-specific markers are labelled. Spearman correlation ($ρ$, $n = 9376$) is overlaid, with a linear regression line for visualisation fitted. (D) Scatter Plots showing log2 fold changes (log2FC) of naive Shef6 hESCs stably cultured on serum coating in the naive state versus differentiation for 6 or 8 days in N2B27, compared to corresponding developmental times (E6 vs. E12 or E14) from the human embryonic reference dataset (subset for embryonic lineages only) (Petropoulos et al, 2016). Technical replicates include $n = 4$ from two independent experiments for d0 and $n = 2$ for all other combinations of conditions and time points. Data was filtered for the 8508 DEGs ($|log2FC| > 1$, padj < 0.05) identified in any comparison between naive hESCs in PXGL and any differentiation timepoint in N2B27, as well as between E5 or E6 and any later embryonic day until E14. Spearman correlation ($ρ$, $n = 8508$) is overlaid, with selected naive- and primed-specific genes labelled. (E) Gene expression of naive (*KLF5*, *KLF17* and *SUSD2*), and primed (*THY1*) pluripotency markers in the human embryonic reference dataset (subset for embryonic lineages only, created with the https://petropoulos-lanner-labs.clintec.ki.se/shinys/app/ShinyEmbryoRef app applying default parameters and subsetting based on Reannotation (Prelineage, ICM, Epiblast only from (Petropoulos et al, 2016)) and for naive Shef6 hESCs stably cultured on serum coating differentiated for 6–8 days in N2B27, measured by RNA-seq. Lines indicate inferred trends based on mean values, with $n = 4$ replicates from 2 independent experiments for d0 and $n = 2$ technical replicates for any other time point shown as dots. (F) Expression of the naive-specific surface marker SUSD2 in naive Shef6 hESCs stably cultured on serum coating during capacitation for 8 days in N2B27, measured by flow cytometry using an APC-conjugated anti-SUSD2 antibody. The y-axis represents the relative median fluorescence intensity (MFI) normalised to the naive state. Bars represent the mean ± SEM of $n = 3$ technical replicates shown as dots for primed hPSCs. Technical replicates from $n = 1$ independent experiment for naive hPSCs and capacitated cells at different timepoints are shown as dots. Source data are available online for this figure.

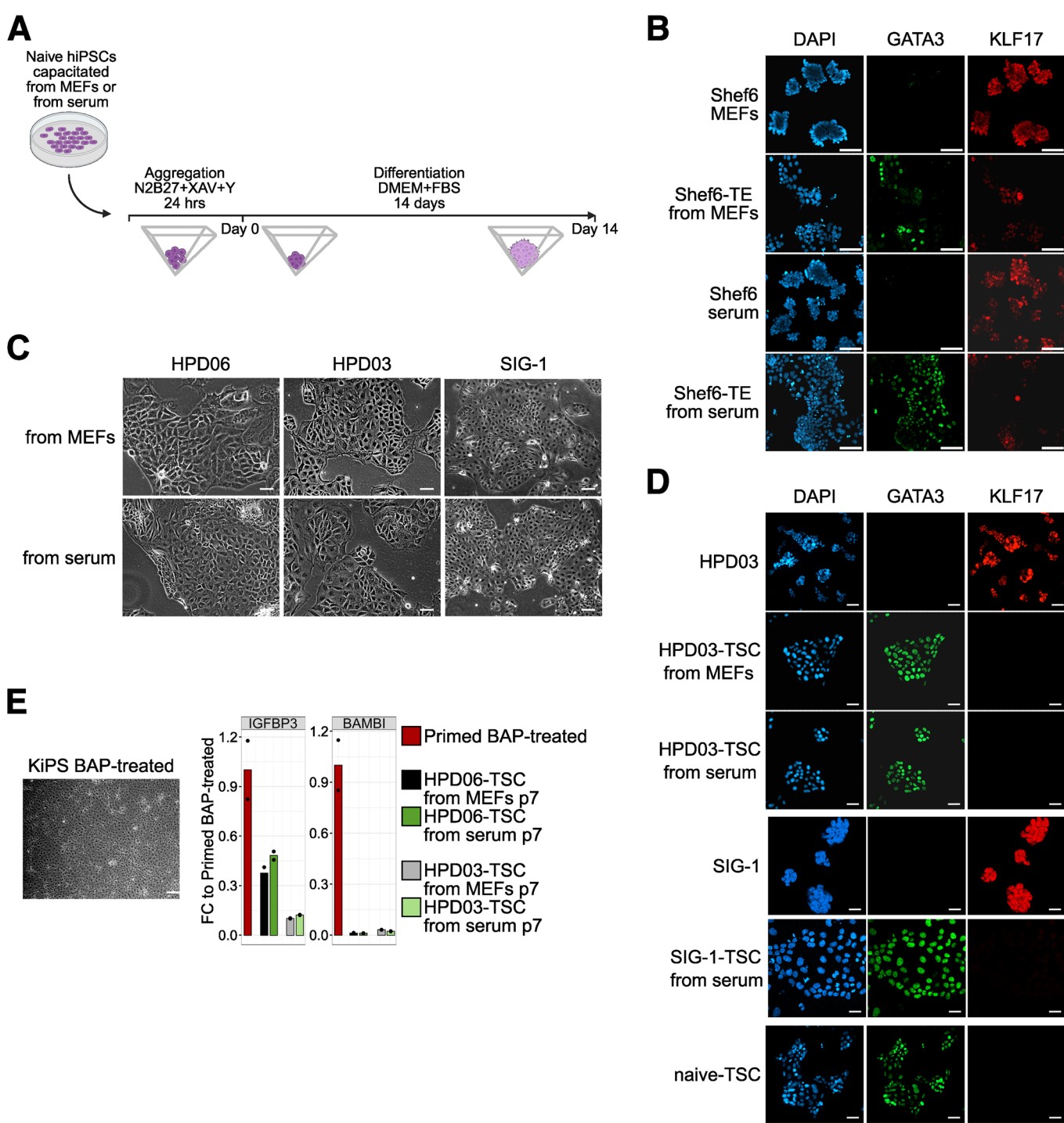

**Figure EV5. Feeder-free naive hPSCs generate bona fide extraembryonic lineages.**

(A) Schematic representation of the experimental setting for the EBs differentiation of naive HPD06 hiPSCs capacitated from MEFs or serum coating. (B) Immunostaining for TE/TSCs (GATA3) and naive pluripotency (KLF17) markers after 5 days of TE differentiation from naive Shef6 hESCs stably cultured on MEFs or serum coating. Scale bars: 100 μm. Representative images of two independent experiments are shown. (C) Morphologies of TSCs derived from naive HPD06, HPD03 and SIG-1 hiPSCs cultured on MEFs or serum coating. Scale bars: 100 μm. Representative images of two independent experiments are shown. (D) Immunostaining for TE/TSCs (GATA3) and naive pluripotency (KLF17) markers in TSCs derived from naive HPD03 and SIG-1 hiPSCs stably cultured on MEFs or serum coating. Scale bars: 100 μm. Representative images of two independent experiments are shown. (E) Left: Morphology of KiPS primed hiPSCs treated with BAP medium for 4 days. Scale bar: 100 μm. Right: Gene expression analysis by RT-qPCR of amnion markers (*IGFBP3* and *BAMBI*) in TSCs derived from naive HPD06 and HPD03 hiPSCs stably cultured on MEFs or serum coating. Technical replicates from $n = 2$ independent experiments are shown as dots. Source data are available online for this figure.

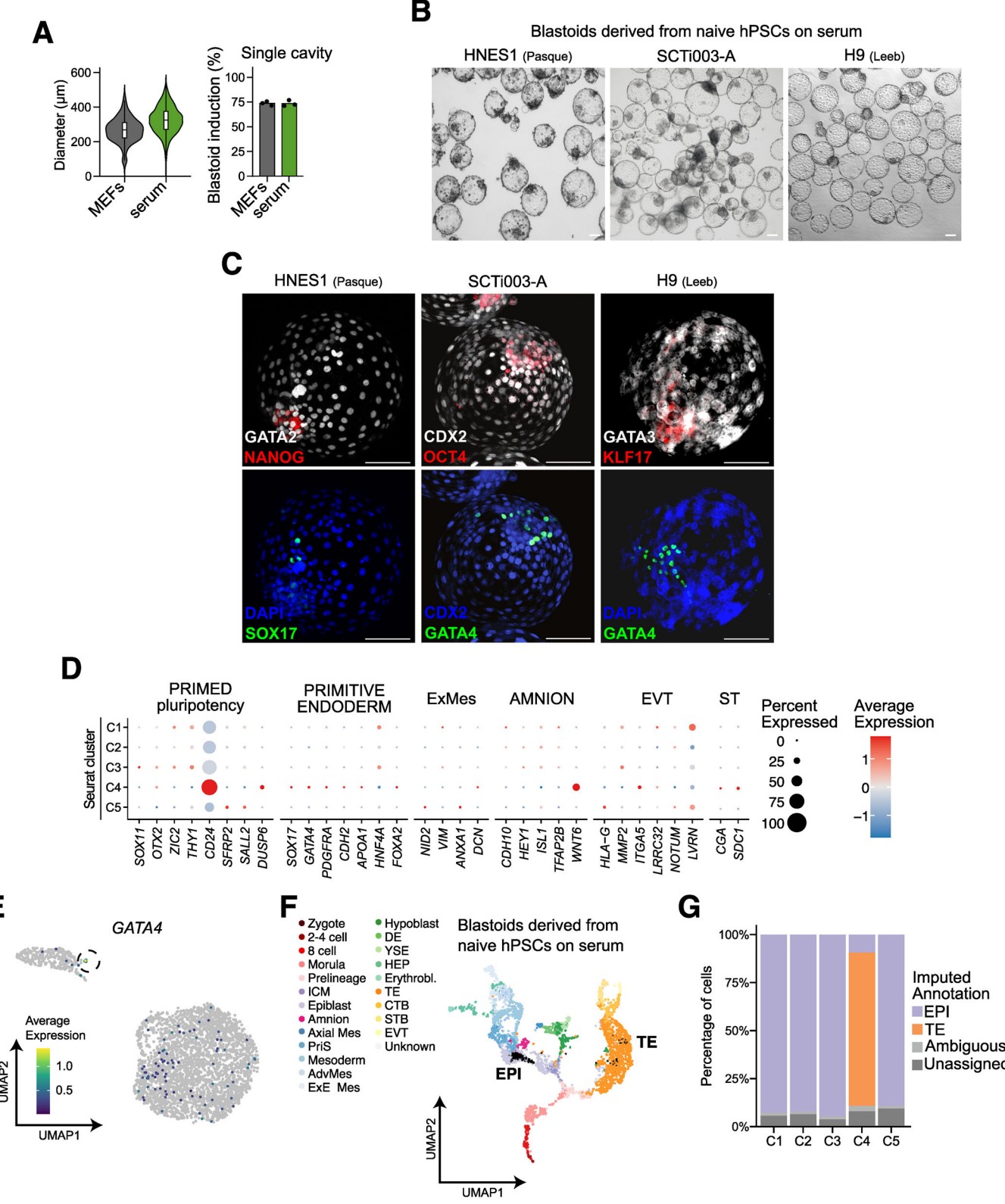

◀  **Figure EV6.   Blastoids from feeder-free naive hPSCs are transcriptionally similar to pre-implantation human embryos.**

(A) Quantification of blastoid diameter (left) and blastoids with a single cavity (right) in structures formed from naive H9 hESCs cultured on MEFs or serum coating. Left: Data is presented as violin plots with the median, 25, 75 percentiles (± min-max) of $n = 407$ (MEFs) and $n = 806$ (serum coating) across $n = 3$ independent experiments. Right: Data shows the mean percentages of technical replicates from $n = 3$ independent experiments for naive H9 hESCs. (B) Brightfield images of blastoids induced from naive HNES1 and H9 hESCs, and SCTi003-A hiPSCs cultured on serum coating. Scale bar: 100 μm. (C) Immunostaining of blastoids derived from naive HNES1 and H9 hESCs and naive SCTi003-A hiPSCs cultured on serum coating. Blastoids were stained for TE (GATA2, CDX2 or GATA3), PrE (SOX17 or GATA4) and EPI (NANOG, OCT4 or KLF17) markers. Shown is the maximum projection. Scale bars: 100 μm. (D) Expression of selected lineage-specific marker genes across Seurat clusters from Fig. 6C. The size of the dots represents the proportion of cells in the indicated group expressing the given gene, and colour encodes the scaled average expression. (E) UMAP plot from scRNA-seq analysis as in Fig. 6C. Cells are coloured according to *GATA4* expression ($n = 3260$). (F) UMAP projection of in vitro d5 blastoids from naive H9 hESCs cultured on serum coating for 4 passages on the human pre-implantation and post-implantation embryos with annotation for scRNA-seq data integration from (Zhao et al, 2025). Cells are coloured by cell type, and black dots show neighbourhoods of in vitro-generated cells projected onto a reference UMAP. (G) Imputed cell annotation across different Seurat clusters as in Fig. 6F. Unassigned and ambiguous labels refer to cells with either none or with more than two imputed stages, respectively. Source data are available online for this figure.

