## [Peer Review File · The EMBO Journal]

Serum coating enables feeder-free culture of naive human pluripotent stem cells preserving developmental potential

Giada Rossignoli, Michael Oberhuemer, Ida Brun, Irene Zorzan, Anna Osnato, Anne Wenzel, Emiel van Genderen, Andrea Drusin, Giorgia Panebianco, Nicolò Magri, Moritz Becker, Mairim Solis, Chiara Colantuono, Sam van Knippenberg, Thi Pham, Sherif Khodeer, Paolo Grumati, Davide Cacchiarelli, Paolo Martini, Nicolas Rivron, Vincent Pasque, Jan Zylicz, Martin Leeb, and Graziano Martello

Corresponding authors: Graziano Martello (graziano.martello@unipd.it) , Martin Leeb (martin.leeb@univie.ac.at), Jan Zylicz (jan.zylicz@sund.ku.dk)

Review Timeline:

Submission Date:	11th Nov 25
Editorial Decision:	12th Dec 25
Editor's Correspondence:	16th Dec 25
Revision Received:	15th Jan 26
Accepted:	27th Jan 26

Editor: Daniel Klimmeck

Transaction Report:

(Please note that the manuscript was previously reviewed at another journal. As EMBO Press has a transfer agreement with that journal, revision was invited based on the reports from that previous external submission. With the exception of the correction of typographical or spelling errors that could be a source of ambiguity, letters and reports are not edited. Depending on transfer agreements, referee reports obtained elsewhere may or may not be included in this compilation. Referee reports are anonymous unless the Referee chooses to sign their reports.)

Point-by-point response to Reviewers' comments

Reviewer #1

In this manuscript, Rossignoli et al. describe the use of a feeder-free protocol for the culture of naive human pluripotent stem cells (PSCs). They demonstrate how serum coating can be used for the long-term maintenance of naive hPSCs. The authors employ a large range of assays to analyze the transcriptome, the genomic stability, the pluripotency status and capability of naive hPSCs to differentiate, including blastoid formation. The authors reason that the routine culture of naive hPSCs on MEF constitutes a major hurdle for studying the human naive state. MEFs are indeed per se variable and depend on embryo age, passage number, genetic background. Moreover, the “exchange of signals between feeders and PSCs prevents a stringent control of the system and makes it difficult to understand the best conditions for maintaining pluripotency”. The overall idea of using serum coating as an alternative to feeder is interesting. I agree that chemically-defined culture conditions should permit to circumvent MEF issues, as highlighted by recent reports proving the interest of such stem cell research (refs 52, 53, Zheng et al., Stem Cell reports 2021).

However, the use of the serum coating proposed by the authors will not provide the stem cell community with a stringently controlled system, as the precise signals provided by the serum for naive pluripotency maintenance are not precisely identified. Moreover, as evoked by the authors, differences of serum batches might impact the hPSCs naive state, but this critical point is not addressed precisely by the authors. In that sense, I am not convinced that the proposed approach will have a significant impact on the stem cell community.

We appreciate the Reviewer's thoughtful comments, but respectfully disagree with the assessment that our study will only have a limited impact on the stem cell community.

Our aim was to provide the field with a robust, simple, and cost-effective alternative to feeder-based systems for the culture of naive hPSCs. The challenges inherent to feeder-based systems—such as cost, labour intensity, variability, and limited accessibility—continue to be a major barrier to the widespread adoption of naive hPSCs research.

Our protocol represents an impactful step forward for the stem cell community in three key ways:

- 1. Accessibility and Scalability:**

Serum-coating dramatically reduces the cost, time and effort of maintaining naive hPSCs by several orders of magnitude. We now discuss this in the discussion section for clarity. Briefly, our serum-coating approach offers a ~100-fold cost reduction compared to commercial ECM substrates, enabling long-term naive hPSC expansion without altering existing media or culture regimes. Batch testing of serum is a well-established, straightforward process—far less labour-intensive than MEFs production—and can be performed at scale. A single batch can support the coating of up to 100,000 6-well plates, making this method not only cost-effective but also highly practical for widespread use. Thereby, naive human culture systems become

more accessible to laboratories that do not have the resources or infrastructure to routinely generate and validate MEFs.

2. **Robustness Across Lines and Laboratories:**

We demonstrate that over 30 serum batches from 7 suppliers, used across 5 independent laboratories and 8 hPSC lines, consistently support the maintenance of naive hPSCs. This high success rate of more than 96% attests to the reproducibility and reliability of the system.

3. **Enabling Downstream Research:**

By simplifying the culture requirements, our protocol lowers the technical threshold to enter the field. This has the potential to enable new labs, expand collaborative networks, and stimulate innovation in areas such as human development, disease modelling, and regenerative medicine. This is an important enabling step that allows more researchers to access powerful models of human development.

Concerning the criticism about *“serum coating proposed by the authors will not provide the stem cell community with a stringently controlled system, as the precise signals provided by the serum for naive pluripotency maintenance are not precisely identified”* we would like to point out that the standard culture conditions for mouse naive hPSCs since the 1980s are based on gelatin, whose composition and potential signalling are yet unclear. Concerning primed hPSCs, the current research standard is the use of undefined ECM mixtures, commercialised as Matrigel or Geltrex, whose composition and potential signalling effects are far from being precisely identified after over 20 years of use across hundreds of labs around the world. We therefore think that our discovery fully matches the standards in the field of hPSCs research.

Designing a fully chemically defined culture regime was not the goal of this study (and there are also no alternative working naive culture systems that are fully chemically defined). However, our work sets the stage for future progress: it facilitates identifying the extracellular matrix (ECM) components, alone or in synergy, that underpin naive pluripotency. The discovery of Collagens and Fibronectins as major shared serum components is a promising result to follow up on. Therefore, this protocol is both practically useful now and conceptually foundational for future chemically defined solutions.

For sure, in the future we will intend to derive fully defined solutions, allowing also GMP production of naive hPSCs, but we foresee that a cocktail of recombinant Collagens and Fibronectins would become an expensive solution, interesting only for companies and not research laboratories.

Taken together, serum coating offers a significant and enabling advance, particularly by reducing technical and financial barriers that currently restrict many labs from working with

naive hPSCs. We believe this approach will expand the reach and impact of research utilising naive hPSCs.

Major points:

Overall the manuscript lacks consistency and is too limited in the ways serum batches and cellular assays are conducted.

1. The number of days/passages naive hPSCs spent in feeder-free culture conditions is variable depending on the experiments conducted. The authors claim that naive hPSCs are expanded for 25 passages in serum coating condition in the abstract, but most of the experiments for example from Figure 1 are conducted after 4 passages on serum coating. Even if the authors explain their choice, they should conduct more experiments with later passages, in line with the abstract.

We unfortunately did not make clear the extent of experimental validation of serum coating in our previous submission. To clarify the passage number at which the different experiments have been performed, we introduced Table 2, which lists all cell lines used for each type of experiment, including the passage number and all serum batches used.

In summary, all experiments were performed between passages 4 and 25 in feeder-free conditions. Experiments in Figure 1 were performed after more than 10 passages in feeder-free conditions.

TABLE 2 - Complete overview of experiments and experimental details of serum-cultured cells:

Experiment	Figure(s) or panel(s)	Line(s)	Passage(s) on serum coating	FBS batch(es)	Group(s)
Low-density conversion	1A S1A-B, D-E	HPD06	p0-p4	Gibco 10270106 (2342201)	Martello
		HPD03	p0-p4		
Long-term maintenance	1B-E, G S1C, F-G S2B	HPD06	>14	Gibco 10270106 (2342201, 2412072) and Gibco A5256701 (2749488, 2740171, 2740173, 2453915)	Martello
		HPD03	>14		
		H9	>20	Sigma Aldrich F7524 (19C111) and Biowest S1600 (S00KI20001)	Leeb
		Shef6	>20		
		HNES1	>10	Gibco A5256701 (B2873995RP)	Pasque
		H9	>10		
		SIG-1	>10	Sigma Aldrich F7524 (0001669689)	Zylicz
		H9	>10		
HNES1	>17				
Genetic engineering	S1G	HPD06	>20	Gibco A5256701 (B2772471RP)	Martello

Proteomics on serum batches	2 S2A	/	/	Gibco 10270106 (2412072) and Gibco A5256701 (B2772471RP)	Martello
				Biowest S1600 (S00KI20001)	Leeb
				Sigma Aldrich F7524 (0001669689)	Zylicz
				Gibco A5256801 (2575650H)	Rivron
Exome-Seq in maintenance	3A-B	HPD06	p17-18	Gibco A5256701 (2740173)	Martello
		HPD03	p17-18		
RNA-Seq in maintenance	3C-E S2C-D S3	HPD06	p14	Gibco 10270106 (2342201)	Martello
		HPD03	p14		
		Shef6	>20	Sigma Aldrich F7524 (19C111)	Leeb
Capacitation	4 S4	HPD06	>20	Gibco A5256701 (2740173, 2453915)	Martello
		Shef6	>20	Sigma Aldrich F7524 (19C111)	Leeb
EBS differentiation from capacitated cells	5A S5A	HPD06	>20	Gibco A5256701 (2740173)	Martello
TE induction	5B-C S5B	H9	p6	Biowest S1600 (S00KI20001)	Leeb
		Shef6	25		
		HNES1	>15		
		HNES1 GATA3::mKO	p8		
TSC differentiation	5D-E S5C-E	HPD06	>20	Gibco A5256701 (2749488, 2740171, 2740173, 2453915)	Martello
		HPD03	>20		
		SIG-1	>10	Gibco A5256701 (B2873995RP)	Pasque
Blastoid generation	6 S6	H9	>20	Biowest S1600 (S00KI20001)	Leeb
		H9	p4-11	Sigma Aldrich F7524 (0001669689)	Zylicz
		HNES1	p8-18		
		KOLF2.1J	p5-10	Gibco A5256801 (2575650H)	Rivron
		STCi003-A	p5-10		
		HNES1	p6	Gibco A5256801 (B2873995RP)	Pasque

2. From the beginning of the manuscript, it is difficult to know whether the results presented are obtained with a unique serum batch, or with independent serum batches tested within the same lab or in independent labs. It is therefore very difficult to evaluate accurately the consistency of the results.

In the Methods section of our first submission, we indicated the information about companies and catalogue numbers of serum batches used in each laboratory, as also mentioned by the Reviewer at point 4, below. However, we appreciate the suggestion that it is important to clearly show all batches tested and used in the study.

We tested over 30 serum batches from 7 suppliers, and 96% of them allowed expansion of naive hPSCs (new Table 3). Among them, 12 were used for experiments across the study by the 5 labs involved, as indicated in the new Table 2. Furthermore, we performed proteomics analysis on 5 different serum batches from the independent laboratories involved in the study (revised Figure 2).

We believe that the consistency of results should be easy to evaluate now, thanks to the reviewer's comment.

3. A large number of results rely on 2 independent experiments, which does not allow statistical analyses (Fig. 1A, B, C, G etc...).

Indeed, many experiments were conducted in duplicate for each cell line in each laboratory. However, the use of multiple cell lines in independent laboratories has allowed for the statistical analysis of the results, considering independent lines as biological replicates. Together, our data unambiguously show the effective long-term maintenance of naive hPSCs on serum-coated cell culture dishes.

4. The fact that five independent laboratories worked on the project is a strength, together with the testing of 8 naive hPSC lines. However, only few approaches were conducted by the 5 independent labs. Moreover, the methods indicate the number of independent serums that have been tested but this number is variable between labs (ranging from 3 independent serums to a unique one). Overall it does not appear clearly throughout the manuscript whether the results are systemically obtained with three independent serum batches or with a single one. This information is crucial to evaluate the impact of this work, especially as a technical report. In that form, it is far too vague to evaluate whether the findings presented are consistent between serum batches.

We thank the Reviewer for appreciating our cross-laboratory effort to establish and evaluate serum coating as an effective method to culture naive hPSCs.

In the resubmitted manuscript, we provide a comprehensive summary of experiments conducted in collaborating laboratories, along with serum batches and cell lines used for each experiment (Tables 2 and 3). The conversion and cultivation of naive hPSCs was performed in each of the labs. Downstream experiments showed in various manners, including NGS-based and functional assays, that naive identity was maintained using different and multiple serum batches. We believe that now it is clear that results were not obtained with a single serum batch, but with several different batches, which were not shared among the different labs.

Concerning the extent of replications and lines used in our study, we notice that even recent studies published in prominent journals about naive hPSC feeder-free conditions are based

on 1 to 3 independent lines¹⁻³ in a single laboratory, with no clear information about the number of batches tested.

We believe that our efforts are well above the accepted standards in the field; indeed, Reviewer 3 stated “A major strength of this study is the extensive validation across eight different PSC lines in five independent laboratories, reinforcing the robustness of the findings”.

5. Authors conduct label-free quantitative proteomics analysis of serum coating. Did the authors analyze a single serum coating among the different serum batches used by the 5 independent labs? A comparative analysis would have permitted to highlight the molecules that are enriched and shared between the different serum batches. In that form, authors indicate the presence of “Fibronectin, Vitronectin, Laminin and several Collagens”. This part of the manuscript just highlight that the authors did not identify a controlled cocktail of molecules that ensure naive hPSCs maintenance.

As suggested by the Reviewer, we performed proteomics on 5 different serum batches from the independent laboratories involved in the study. These analyses are very informative as they identified ECM proteins consistently present in all batches, which are Fibronectin and 5 different Collagens.

As discussed in the major point above, these novel results will be instrumental for future studies aimed at identifying the minimal combination of recombinant Collagens and Fibronectin sustaining human naive pluripotency. As a single ECM protein might not be sufficient for naive hPSC expansion, we will have to systematically test over 30 combinations of 2 or 3 ECM proteins. These efforts would represent a new research project, going well beyond the scope of the current study.

Although we agree that a fully defined ECM cocktail would be crucial for future translation applications based on naive hPSCs, we doubt it would become standard conditions for expansion in research laboratories, exactly like gelatin and Matrigel are the common choices for expansion of murine and hPSCs.

6. Transcriptomic analysis of 3 naive hPSC lines grown in serum coating or on feeders show that the serum-coating condition maintains the global naive pluripotent signature. PCA from Figure 3C is once again very vague. Some blue dots (corresponding to naive hPSC samples) cluster closer to primed samples but the precise nature of these samples is not clearly indicated. Moreover, authors should decipher whether significant DEGs emerge within each line grown in feeder or feeder-free conditions and work on these differences. They should also compare the effects of various serum batches coating on the transcriptomes of the same naive hPSC line. Without these experiments, it is difficult to conclude that serum coating constitutes a concrete alternative to feeder. Overall, the analysis of the RNA-seq dataset is very limited and should be deepened to reach informative conclusions. Here as well, details about the number of passages in serum-coating conditions should be provided.

In the resubmitted manuscript, we have improved the visualisation of the PCA in Figure 3C following the precious suggestions of the Reviewer. Now all dots can be easily assigned to a

given study. For instance, black symbols show samples from this study (3 independent naive hPSC lines, on serum or MEFs), while all other colours represent data from previously published datasets available from the literature. We now use different shapes to indicate coating regimes and different colours to indicate cell types.

Again, to increase transparency and clarity, Table 2 indicates all the passage numbers for each experiment/dataset.

The samples in the naive cloud near the primed hPSCs are from Bayerl *et al*⁴, cultured in HENSM medium. We have included them for completeness, as they have been cultured on a feeder-free substrate. However, as the Reviewer indicated, they are not transcriptionally in a fully naive pluripotent state and are more akin to a transition state between naive and primed pluripotency⁵.

To provide a more comprehensive analysis of transcriptional data, we derived differentially expressed genes (DEGs) between naive hPSCs cultured on serum coating and MEFs. This analysis revealed only a small number of DEGs, most of which were pseudogenes, without significant enrichment for any biological process. Notably, we could not detect differences in the expression of any pluripotency or differentiation markers. These data are now shown in Supplementary Figure 3.

7. Differentiation in EBs will clearly benefit from the detection of additional lineage-specific markers by Q-RTPCR. Moreover, detecting germ layer induction by immunofluorescence and/or Western blot is required to assess whether lineage-specific proteins are induced to a similar extent in feeder and feeder-free naive hPSCs.

We analysed expression of more marker genes for EBs (revised Figure 5A), which further highlighted that naive hPSCs cultured on serum efficiently differentiate to the three germ layers, similarly to naive hPSCs previously cultured on MEFs.

Reviewer #2

The authors replaced the mouse embryonic fibroblast (MEF) feeder cells with serum-based coating for culture of naive pluripotent stem cells (PSC) due to the limitations of MEF. They have established this culture in 8 different lines across 5 different laboratories. They show that naive PSC grown on MEF and serum-based coating are similar in terms of morphology and expression of pluripotency and naive markers. They performed proteomic analysis on the serum-coating and showed that Fibronectin, Vitronectin, Laminin, and several Collagens are present in the coating to support the cell culture. They also demonstrated that the naive PSC grown on serum-based coating do not accumulate pathogenic mutations and retain the lack of imprinting typically seen in naive cells. They further showed that naive PSC grown on serum-based coating recapitulate the embryonic differentiation trajectories and that they can efficiently differentiate into three germ layers and trophoblast stem cells (TSC). These cells are also shown to self-assemble into blastoids.

Overall comments:

1. Overall, this manuscript is well supported with evidence showing that naive PSC grown on serum-coating is similar to naive PSC grown on feeder MEF cells, in terms of morphology, transcriptomic profile, differentiation potential, and blastoid generation. The manuscript focuses its efforts on comparing the serum-based cells to MEF-based cells, as well as to other published data. Although the manuscript successfully adapted the naive PSC culture to feeder-free serum-based culture, the study only reports incremental advancements.

We appreciate the Reviewer's thoughtful comment, but respectfully disagree with the assessment that our study provides only an incremental advance. Serum coating is a simple solution with a major impact on the utility, feasibility and accessibility of naive human stem cell culture protocols, enabling many researchers to pursue advanced pluripotency studies with fewer financial and technical constraints.

Our aim was to provide the field with a robust, simple, and cost-effective alternative to feeder-based systems for the culture of naive hPSCs. The challenges inherent to feeder-based systems, such as cost, labour intensity, variability, and limited accessibility, continue to be a major barrier to the widespread adoption of naive hPSCs research.

Our protocol represents an impactful step forward for the stem cell community in three key ways:

1. Accessibility and Scalability:

Serum-coating dramatically reduces the cost, time and effort of maintaining naive hPSCs by several orders of magnitude. We now discuss this in the discussion section for clarity. Briefly, our serum-coating approach offers a ~100-fold cost reduction compared to commercial ECM substrates, enabling long-term naive hPSCs expansion without altering existing media or culture regimes. Batch testing of serum is a well-established, straightforward process, far less labour-intensive than MEFs production, and can be performed at scale. A single batch can support the coating of up to 100,000 6-well plates, making this method not only cost-effective but also highly practical for widespread use. Thereby, naive hPSC systems become more accessible to laboratories that do not have the resources or infrastructure to routinely generate and validate MEFs.

2. Robustness Across Lines and Laboratories:

We demonstrate that over 30 serum batches from 7 suppliers, used across 5 independent labs and 8 hPSC lines, consistently support the maintenance of naive hPSCs. This high success rate of more than 96% attests to the reproducibility and reliability of the system.

3. Enabling Downstream Research:

By simplifying the culture requirements, our protocol lowers the technical threshold to enter the field. This has the potential to enable new labs, expand collaborative

networks, and stimulate innovation in areas such as human development, disease modelling, and regenerative medicine. This is an important enabling step that allows more researchers to access powerful models of human development.

We hope that, on balance, the Reviewer appreciates that serum coating offers a significant and enabling advance, especially by reducing technical and financial barriers that currently restrict many labs from working with naive hPSCs.

2. The manuscript needs to be proofread further as there are some inconsistencies between the actual figures and what was referred in the text.

We proofread the manuscript and made sure figures and text are consistent. We also corrected all points listed below.

Major comments:

1. Although the culture is adapted to feeder-free serum-based coating, it is important to note that fetal bovine serum (FBS) used in this serum coating also has limitations. They are prone to batch-to-batch variations and also contains undefined components. Therefore, quality control (QC) on the serum is necessary, and this may translate to substantial workload similar to preparing feeder cultures.

Batch testing of serum is a straightforward process, far less labour-intensive than MEFs production, and can be performed at scale. A single batch of 50 litres of serum can support the coating of up to 100,000 full 6-well plates, making this method not only cost-effective but also highly practical for widespread use. Feeder production, in contrast, is much more work and cost-intensive. Of note, a single round of MEFs production will typically involve the expansion of MEFs to more than 100 15-cm dishes to generate sufficient MEFs to coat only 400 6-well plates.

Indeed, serum needs to be batch tested, which is also true for feeders, although the high success rate of 32 out of 33 tested serum batches (96%) indicates that the large majority of sera will work for our protocol. Of note, we now provide in Table 3 the list of all batches used with success, representing a useful resource for the community.

Therefore, using serum coating provides a significant advantage for culturing naive hPSCs.

2. When culturing HPD06 and HPD03 at low density, there was a lag phase. First of all, was it fairly compared to MEF-based cells also seeded at low density? Secondly, the lag phase was not clearly explained, especially since it was only observed in these 2 cell lines.

We thought it could be interesting, in light of the potential expansion and application of this new methodology in many more laboratories, to highlight that some cell lines may experience an adaptation lag phase in the first passages of the conversion. In our experience, this lag phase is characterised by a transient reduction in the proliferation rate and clonogenic capacity

of the naive hPSCs (revised Supplementary Figure 1E-F), with no impact on morphology (revised Figure 1A) and the expression and protein levels of general and naive pluripotency markers (revised Supplementary Figure 1B-C).

We experienced this behaviour with the first two cell lines we tested, when cells were moved from MEFs to serum for starting the conversion at the same density as the routine maintenance of MEFs. However, both lines spontaneously recovered proliferation and clonogenic capacity to levels comparable to the same cell line cultured on MEFs by passage 8 (revised Figure 1E-F) without any further treatment or manipulation.

For these first converted cell lines, we also noticed that the conversion does not present a lag phase when plating a higher amount of cells compared to the one routinely chosen for maintenance on MEFs for the first couple of passages. By doing so, the adaptation to feeder-free conditions occurs swiftly. Of note, we therefore chose to keep the confluence a bit higher than usual for the first passage to convert all the other cell lines, and indeed, we did not experience any lag phase. This is indicated in the first section of Results (See Page 6, lines 182 - 183).

To clarify this point, we rearranged and better explained this aspect in the First Section of the Results. We also modified the proliferation graphs in order to highlight that the lag phase can occur in the first passages of the conversion (p0-p4, revised Supplementary Figure 1E), but spontaneously recovered afterwards (p14-p18, revised Figure 1E).

3. When the genomic sequences were compared to reference genome, could the shared variants between MEF and serum-based conditions be due to the patient/donor-specific variants?

Indeed, the shared variants that we detected can be due to intrinsic genetic variability compared to the reference genome. Moreover, cells have been cultured on MEFs before starting the feeder-free conversion experiment. Therefore, some of those shared variants could have been accumulated during the previous culture. See Page 7, lines 216 - 220.

4. in fig. 5B, to claim the serum-based cells differentiated at even better efficiency than the MEF-based cells, more experiments need to be done as the evidence is not strong enough.

We agree with the Reviewer and changed our statement to “We conclude that naive hPSCs cultured on serum differentiate to the three germ layers with an efficiency similar to naive hPSCs previously cultured on MEFs” (lines 315-317). We added new markers for the germ layers (revised Figure 5A), and detected consistent behaviour with the ones shown before.

5. in fig. 1D and 1E, it is shown that the two cell lines grown on MEF and serum are comparable. However, in sup. fig. 1D and 1E, the same cell lines are shown to have a lag phase when adapting to serum culture from P0 onwards. This is confusing.

See reply to Major Point 2.

6. in fig. 4G, it is shown that the naive PSC (serum-based) follow the similar differentiation trajectories to the human embryo development. However, looking at the heatmap from fig. 4F, the embryo development does not show a similar gene expression profile as the rest of the *in vitro* cultures.

A comparison at the fold change level between *in vitro* and *in vivo* pluripotency transitions at the individual transcript level is difficult to visualise in a scaled heatmap. We have therefore decided to remove this data item from the manuscript. Data integration, now shown in Figure 4G, and comparative analysis of key markers, shown in Supplementary Figure 4E, highlight that global transcriptomes align and key markers are regulated in a comparable manner between *in vivo* and *in vitro* culture. In sum, our data clearly show that cells cultured on serum coating differentiate along the epiblast lineage with overlapping kinetics compared to MEF-cultured cells.

7. in fig. 5D, the genes do not display an increasing trend but rather a fluctuation in the gene expression levels are seen across the different passages. What could possibly contribute to such drastic changes, especially in GATA2 gene expression?

Figure 5D shows loss of naive pluripotency transcripts and acquisition of TE/TSCs marker expression when exposing naive hPSCs to TSC culture conditions. The two cell lines used acquire TSC identity (indicated by GATA2 and GATA3 expression) with different kinetics. These are, however, consistent between starting from MEFs or serum coating and therefore indicate clone-specific variation rather than differences in coating protocols. Some fluctuation between passages and also between clones is observed, but it is relatively minor and not unexpected. Overall, all tested cell lines and conditions reach levels of GATA2 and GATA3 similar to those observed in stably cultured TSCs. These qPCR data are further corroborated by the IF analysis shown in Figure 5E. In addition, we have now included more TE differentiation experiments, shown in Figure 5C (lines 317 - 322 and 518 - 522). They clearly show that naive hPSCs, regardless of their previous culture on MEFs or serum coating, efficiently differentiate into TE. We conclude that serum coating allows the efficient differentiation of naive hPSCs into TE and TSCs.

Minor comments:

1. in line 300, it should be fig. 5C instead of fig. 4C.

2. in line 312, the authors could be referring to fig. 5C instead of fig. 4A.

Typos in the text highlighted in Minor Comments 1 and 2 have been corrected.

3. in fig. 4B, it would be good to show the comparison of serum-based and MEF-based for the cell line Shef6 as well.

Current Figure 4B compares Shef6 and HPD06 on serum, while Figures 4C, E-G compare Shef6 on serum against two lines on MEFs. In all cases, the capacitation trajectories are consistent across lines and conditions.

4. in fig. 6B and sup. fig. 5C, it would be good to show the same cell lines for the staining of GATA3, NANOG, CDX2, and OCT4.

We added more cell lines in the mentioned Figures (revised Figure 6 and Supplementary Figure 6). In particular, we added naive HNES1 and H9 hESCs-derived blastoids from different groups.

We also characterised the blastoids using multiple markers. For example, blastoids derived from HNES1 hESCs in two different laboratories were stained for GATA2 or CDX2 and NANOG or OCT4.

Reviewer #3

The manuscript by Rossignoli and colleagues reports a set of findings showing that human PSCs cultured under the PXGL regimen—which supports their self-renewal in the naive state—but where feeders were replaced with fetal calf serum (FCS), retain all their molecular and functional characteristics. Growth rate, morphology, and, most importantly, the expression of naive pluripotency markers are maintained. Genetic integrity is also preserved, and even slightly improved compared to the same cells cultured on feeders. Finally, PSCs cultured in PXGL + FCS retain their ability to differentiate along the same embryonic and extra-embryonic trajectories and to form blastoids with similar efficiency and characteristics as cells cultured in PXGL + feeders.

Overall, the manuscript is very clear and the data is solid. A major strength of this study is the extensive validation across eight different PSC lines in five independent laboratories, reinforcing the robustness of the findings.

However, the culture system developed in this study has specific weaknesses that must to be thoroughly addressed:

1. The primary concern is the uncontrolled variability between fetal calf serum (FCS) batches. Since the entire study was conducted using a single batch, it remains uncertain whether other laboratories would achieve the same level of performance with serum from a different source. This raises significant questions about the protocol's scalability. Is the system robust enough to tolerate inherent variations in serum composition? How was this specific batch selected? Could the authors identify any key serum components in this batch that could guide the selection of future batches worldwide?

We thank the Reviewer for appreciating the methodological strength of our approach and the advantage of independently evaluating our protocol across 5 laboratories. We apologise for failing to highlight the scalability and robustness of our protocol better. In our multi-lab

approach, we have used and evaluated more than 30 serum batches on multiple commonly used hESCs and hiPSCs. In our first submission, we already provided information about the different catalogue numbers and companies each laboratory obtained the serum from. However, we agree this is a crucial point; therefore, to highlight this better, we have introduced Tables 2 and 3.

- Table 2: a detailed overview of all experiments performed, including cell lines and passage numbers, but also the different serum batches used;
- Table 3: a list of all serum batches we tested during the time of this work, comprehensive of brand name and catalogue and lot numbers.

The successful testing rate (over 96% of batches working) highlights that the system is robust enough to tolerate most inherent variations in serum composition.

To expand the evaluation of serum composition, we ran a further proteomics experiment including four more serum batches, highlighting the molecules (5 Collagens and Fibronectin) that are shared between serum batches used by different laboratories (revised Figure 2). As only one serum batch failed to sustain naive hPSC, we are not able to identify with statistical confidence which components were potentially harmful or missing.

2. Another equally critical issue is the quality of feeder-free cultures for cell transfection experiments. In the discussion, the authors suggest that these experiments are best performed on feeders. I see this as a cautionary note regarding the protocol's robustness. Moreover, the inability to use this protocol for transfection and antibiotic resistance selection experiments is particularly unfortunate, as these are precisely the scenarios where feeders pose the greatest challenges. Many laboratories lack easy access to multi-resistant transgenic mice, which further compounds the issue. This limitation could significantly hinder the protocol's broader adoption. The authors should be more explicit about the challenges they encountered in their genome-editing experiments conducted without feeders.

We agree that this is an important point. We want to clarify that performing stable genetic manipulation of serum-cultured naive hPSCs is possible without the need to re-optimize protocols previously used on MEFs. We could swiftly generate lines stably expressing constructs of interest, without the need for multi-resistant MEFs, which are not easily available to all laboratories. We added an example of this in the revised Supplementary Figure 1 (Supplementary Figure 1G, lines 184 - 185 and 499 - 507).

We should mention that the recovery of colonies after fluorescence-activated cell sorting was higher on feeders, as also observed during the genetic engineering of mESCs. We note that shuttling serum-adapted naive hESCs between serum coating and MEFs works without the need for re-adaptation to feeder-free conditions. Therefore, MEFs could be used only for the recovery of clonal colonies after complex low-efficiency genetic manipulations. We added a discussion section on this aspect (lines 411 - 418).

Other points:

3. Lines 78-81: *This sentence is ambiguous and may misleadingly suggest that human blastocysts spontaneously give rise to PSCs in the primed state, while mouse blastocysts give rise to PSCs in the naive state. In reality, both mouse and human blastocysts can generate PSCs in the primed state when cultured in an FGF2 ± Activin A medium. Similarly, both can generate PSCs in the naive state when cultured under a naive state regimen. This sentence should be revised for accuracy.*

We agree with the Reviewer that this sentence may be misleading. We clarified the concept as follows (now lines 83-88):

“Conventional hPSCs cultured without feeders on Matrigel or vitronectin-coated plates in Essential 8 (E8) or mTeSR media including FGF2 and TGFβ are in a primed pluripotent state more akin to the post-implantation epiblast. This state is highly similar to the one of mEpiSCs in terms of growth factor dependence, transcriptional and epigenetic regulation. The ability to convert mEpiSCs into mESC through culture conditions or transient overexpression of naive-specific TFs such as Klf4 provided the paradigm for the derivation of naive hPSCs.”

4. Lines 160–167: *The data on the HPD06 and HPD03 lines are unclear. It is stated that cells exhibit a significant slowdown in proliferation rate and reduction of clonogenicity for four passages after transitioning to feeder-free culture conditions. However, it is later mentioned that "upon long-term culture in feeder-free conditions," they proliferate at the same rate as feeder-grown cells. What exactly does "long-term" mean? How many passages are required for the cells to regain their original proliferation rate in feeder-free conditions? Do cells also regain their original clonogenicity? Additionally, panels 1D and S1D appear to share the same time scale (p0 to p4), which is inconsistent and needs clarification.*

We thought it could be interesting in light of the potential expansion and application of this new methodology in many more laboratories, to highlight that some cell lines may experience an adaptation lag phase in the first passages of the conversion. In our experience, this lag phase is characterised by a transient reduction in the proliferation rate and clonogenic capacity of the naive hPSCs (revised Supplementary Figure 1E-F), with no impact on morphology (revised Figure 1A) and the expression and protein levels of general and naive pluripotency markers (revised Supplementary Figure 1B-C).

We experienced this behaviour with the first two cell lines we tested, when cells were moved from MEFs to serum for starting the conversion at the same density as the routine maintenance of MEFs. However, both lines spontaneously recovered proliferation and clonogenic capacity to levels comparable to the same cell line cultured on MEFs by passage 8 (revised Figure 1E-F) without any further treatment or manipulation.

For these first converted cell lines, we also noticed that the conversion does not present a lag phase when plating a higher amount of cells compared to the one routinely chosen for maintenance on MEFs for the first couple of passages. By doing so, the adaptation to feeder-free conditions occurs swiftly. Of note, we therefore chose to keep the confluence a bit higher than usual for the first passage to convert all the other cell lines, and indeed, we did not experience any lag phase.

To clarify this point, we rearranged and better explained this aspect in the First Section of the Results. We also modified the proliferation graphs in order to highlight that the lag phase can occur in the first passages of the conversion (p0-p4, revised Supplementary Figure 1E), but spontaneously recovers afterwards (p14-p18, revised Figure 1E).

5. Figure 3B: including the percentage values (numbers) for the Tier I to IV fractions would be very helpful in better highlighting the differences. For instance, the Tier II fraction seems slightly larger in the "serum-specific" condition compared to the "MEFs-specific" control condition for the HPD06 line.

We replaced the boxplot with pie charts to more clearly introduce percentage values in Figure 3B as suggested.

HPD06 line accumulated 3 Tier II variants, 24 Tier III variants and 329 Tier IV variants on serum coating, while it accumulated 3, 120 and 813 variants, respectively, on MEFs. In the two different conditions, the HPD06 line accumulated 3 different Tier II variants. As the total number of detected variants on serum coating is about a third compared to MEFs culture, the percentage of Tier II variants in feeder-free increases even if the total number is equal (3).

HPD03 line instead accumulated 2 Tier II variants, 22 Tier III variants and 287 Tier IV variants on serum coating, compared to MEFs, where they accumulated 4, 73 and 432 variants, respectively. For this line, detected Tier II variants in feeder-free were half of and different from the variants detected upon MEFs culture.

Therefore, the culture in feeder-free conditions accumulated fewer total variants, without clear evidence for an enrichment for likely pathogenic variants (Tier II).

References:

1. Huang, T. *et al.* Inhibition of PRC2 enables self-renewal of blastoid-competent naive pluripotent stem cells from chimpanzee. *Cell Stem Cell* **32**, 627-639.e8 (2025).
2. Cesare, E. *et al.* 3D ECM-rich environment sustains the identity of naive human iPSCs. *Cell Stem Cell* **29**, 1703-1717.e7 (2022).
3. Szczerbinska, I. *et al.* A Chemically Defined Feeder-free System for the Establishment and Maintenance of the Human Naive Pluripotent State. *Stem Cell Reports* **13**, 612–626 (2019).
4. Bayerl, J. *et al.* Principles of signaling pathway modulation for enhancing human naive pluripotency induction. *Cell Stem Cell* **28**, 1549-1565.e12 (2021).
5. Dekel, C. *et al.* Stabilization of hESCs in two distinct substates along the continuum of pluripotency. *iScience* **25**, 105469 (2022).

Final response from Reviewers

Our comments are highlighted in green.

Reviewer #1

I initially reviewed this manuscript for [journal name]. I would first like to acknowledge the considerable effort the authors have made to improve the scientific rigor of the study and to clarify their findings. In this reformatted version, they have clarified the experimental conditions and designs, notably by including summary tables that are indispensable for transparency. They have also performed additional experiments, such as proteomic analyses across a broader panel of cell lines and serum batches, which at least partially enhances the robustness of the work.

Despite these improvements, I remain skeptical about the study. I regret that several key concerns regarding the robustness of core findings have not been adequately addressed. In particular, the central claim relating to the genomic stability of naive human pluripotent stem cells (hPSCs) maintained under the serum-coating condition lacks the systematic and comprehensive experimental support that would be required to substantiate it. Only two cell lines—both from the same laboratory (Martello)—have been assessed for their mutational profiles, which is insufficient to draw generalizable conclusions.

Stable maintenance of genomic stability of human naïve PSCs is a well-known issue. We want to note that we did not make any claims towards having solved genomic stability issues by using serum coating.

This point about genomic stability was not raised in the first round of peer-review. If we were asked to perform the exome sequencing on multiple lines, we would have done it, although, in our opinion, two independent cell lines provide very clear interpretable data supporting our claims.

Overall, this is a perfect example of moving the goal post during the revision process, which precludes meaningful and productive peer review.

The authors refer to previous publications as indirect validation of their method, noting that “1–3 independent lines were used in a single laboratory.” However, it is important to emphasize that the positioning of the first two cited studies differs substantially from the current work. Those publications identified important regulators of human naive pluripotency, such as PRC2 and ECM components, but they were not framed as resources or protocols intended for widespread use by the stem cell community. The third reference, from 2019, is indeed more analogous in its positioning, but I would raise similar concerns about robustness and scope were I reviewing it for a high-impact journal.

We are quite puzzled about this comment.

We have never referred to previous publications as “indirect validation of our method”. Simply put, the “gold standard” in the most recent papers about human naïve PSCs, which we referred to, is using 1 or 2 lines in a single lab, without any information about how many batches of a given reagent were used.

As clearly shown in Tables 1 – 3, we were extensively characterised 8 lines in 5 labs, using 32 different batches of serum. Thus, our study surpassed by one order of magnitude this “gold standard”, but the reviewer does not appreciate this.

The reviewer’s subjective view about what type of paper is fit for what type of journal and what level of evidence is required to achieve a certain impact factor are against our understanding of the scientific process.

More broadly, I remain unconvinced that the manuscript represents a substantive advance for the stem cell field. The main contribution appears to be the cost-effectiveness of the proposed method, rather than the development of a fully validated or mechanistically novel culture system. While the authors argue that working with MEFs is technically demanding, I would point out—as they are likely aware—that commercially available immortalized fibroblast lines (e.g., STO cells from ATCC) can be easily and reliably used as feeders. G418-resistant derivatives (e.g., SNL cells) are also readily available. Therefore, I am not persuaded that the complexity or accessibility of MEF-based systems represents a major barrier for new laboratories entering the field of human naïve pluripotent stem cell research.

We made very clear that feeder-free conditions allow to save time and money, and even more importantly solve the issues of contaminations by feeder cells (please see how we detected contamination of murine transcripts in most published datasets of human naïve PSCs expanded on mouse feeders, see Figure 3d and S2d).

Just to give a simple example, any metabolomic analyses on a co-culture of human naïve PSCs and mouse feeders is basically pointless, as it would not be clear what cell produces/uses what metabolite.

In conclusion, while the authors have made genuine efforts to improve the manuscript, particularly in terms of clarity and transparency, the study still falls short in terms of robustness and depth of validation. The claims, especially those concerning genomic stability, remain insufficiently supported by rigorous data. Although the proposed method may be of practical interest due to its cost-efficiency, I am not convinced that the scope and depth of the current study are commensurate with the standards expected for publication in 2025 in [journal name].

Reviewer #2

The manuscript focuses its efforts on comparing the serum-based cells to MEF-based cells, as well as to other published data. Although the manuscript successfully adapted the naive PSC culture to feeder-free serum-based culture, the study only reports incremental advancements. Although the authors presented 3 reasons for the advancement over MEF-based methods, the serum-based culture still has some inherent limitations such as batch to batch variation of serum. Therefore, it is questionable whether there will be wide adoption of this method. The reviewer does not find this manuscript suitable for publication in [journal name].

We are pleased to see that the robustness of our results is not questioned, but now the issue is about the wide adoption of our method, which would be limited by the batch-to-batch variations of serum.

This point was already raised during the first round and we both discussed the advantages of feeder-free cultures and provided additional evidence, by showing that 32 out of 33 batches worked in our 5 labs (see table 3). We also analysed the composition of 5 batches of serum, finding shared proteins.

As a comparison, Matrigel and Geltrex are used by hundreds of labs around the world, but we all know there is batch variability and their composition / critical components is still unclear.

Reviewer #3

I consider that the authors have adequately addressed my comments raised in the first submission.

We are pleased to see that also this reviewer found that all scientific issues raised during the peer-review were addressed.

Dear Dr Martello,

Thank you again for the submission of your amended manuscript (EMBOJ-2025-123014) to The EMBO Journal. Please accept again my apologies for the unusual protraction with the assessment process due to delayed expert input. We have carefully evaluated your manuscript and the point-by-point response provided to the referee concerns that were raised during review at a different journal. In addition, and as mentioned before, we decided to involve an arbitrating expert to evaluate the revised version of your work, with respect to technical robustness, conceptual advance and overall suitability of your work for publication in The EMBO Journal.

As you will see from the arbitrating comment enclosed below, the advisor is in favour of the work stating the interest and value of your methods development and therefore supportive of publication at The EMBO Journal.

We are thus pleased to inform you that we can offer to swiftly move forward towards acceptance of this work at The EMBO Journal as a methods resource.

We now need you to take care of a number of minor issues related to formatting and data annotation, which I will share shortly in a separate message, together with additional changes and requests by our production team and information regarding Source Data provision.

Please submit a revised version of the manuscript at your earliest convenience using the link enclosed below, addressing the advisors' comments.

As you might remember from previous experience at EMBO Press, every paper at our journals includes a 'Synopsis', displayed on the html and freely accessible to all readers. The synopsis includes a 'model' figure as well as 2-5 one-short-sentence bullet points that summarize the article. I would appreciate if you could provide this figure and the bullet points.

Thank you again for giving us the chance to consider your manuscript for The EMBO Journal, I look forward to hearing from you and receiving your final revised version of the manuscript.

Best regards,

Daniel Klimmeck

EMBOJ-2025-123014

Arbitrating advisor's comment:

I have read the revised manuscript as well as the first and second rebuttal by the authors. I agree with your assessment that this study does provide a valuable addition to the human PSC field as it provides an affordable and readily accessible method to grow naive human PSCs capable of in vitro differentiation and blastoid formation while maintaining a naive transcriptional program. I am impressed by the number of cell lines and serum batches the authors have tested as well as the number of independent laboratories that validated key results. This makes the manuscript in my view a compelling and impactful resource appropriate for EMBOJ and should be of broad interest to the field. Thus, I recommend moving forward with this manuscript.

Please remember: Digital image enhancement is acceptable practice, as long as it accurately represents the original data and conforms to community standards. If a figure has been subjected to significant electronic manipulation, this must be noted in the figure legend or in the 'Methods' section. The editors reserve the right to request original versions of figures and the original images that were used to assemble the figure.

Dear Dr Martello,

Further to below, please find the mentioned additional formatting changes required for the final revision enclosed at the bottom of this message.

Please let us know any time should there be additional questions related.

Best regards,

Daniel Klimmeck

>> Please provide the main manuscript text as .docx file.

>> Limit the abstract to maximally 175 words.

>> Limit the title to maximally 100 characters.

>> Resubmit the revised manuscript with an adjusted manuscript type 'Method'.

>> Author Contributions: Remove the author contributions information from the manuscript text. Note that CRediT has replaced the traditional author contributions section as of now because it offers a systematic machine-readable author contributions format that allows for more effective research assessment. and use the free text boxes beneath each contributing author's name to add specific details on the author's contribution.

More information is available in our guide to authors.
<https://www.embopress.org/page/journal/14602075/authorguide>

>> Provide a completed Author Checklist.

>> Please correct the headings of order of the sections in the manuscript text to: Abstract / Keywords / The Paper Explained / Introduction / Results / Discussion / Methods / Data Availability / Acknowledgements / Disclosure and Competing Interests Statement / References / Main Figure Legends / Tables / Expanded View Figure Legends

>> Figures in separate files: Figures should be removed from the manuscript text and uploaded as individual, high resolution figure files. Legends should be placed after the References.

>> References: adjust the reference format to EMBO Journal format, 10 authors et al, and place References after the Disclosure and competing interests statement, before figure legends.

>> Data availability section: please ensure the GEO and PRIDE datasets are made publicly accessible.

>> Please indicate redisplay of data from Fig 1B in the figure legend of Supplemental Figure S1A,B.

>> Add a Reagents and Tools table to the Methods section, as a separate file using the existing template in the Guide For Authors, listing key reagents, experimental models, software and relevant equipment.

>> Please provide a complete set of source data and a source data checklist for the study as requested by the separate e-mail. Source data: source data should be uploaded as one (zipped) file per figure.

>> Consider additional changes and comments from our production team as indicated below:

- Figure legends:

1. Please indicate the statistical test used for data analysis in the legend of figure S3 C
2. Please note that the box plots need to be defined in terms of minima, maxima, centre, in the legend of figure S1 A
3. Please note that the box plots need to be defined in terms of minima, maxima, centre, bounds of box and whiskers, and percentile in the legends of figures S4 E
4. Please note that information related to n is missing in the legend of figure S3 C
5. Please note that n=2 in figures 1D, E; 4A, B, D; 5A, C, D; S1 E, F; S4 F, S5 E

Dear Dr Martello,

Thank you again for the submission of your amended manuscript (EMBOJ-2025-123014) to The EMBO Journal. Please accept again my apologies for the unusual protraction with the assessment process due to delayed expert input. We have carefully evaluated your manuscript and the point-by-point response provided to the referee concerns that were raised during review at a different journal. In addition, and as mentioned before, we decided to involve an arbitrating expert to evaluate the revised version of your work, with respect to technical robustness, conceptual advance and overall suitability of your work for publication in The EMBO Journal.

As you will see from the arbitrating comment enclosed below, the advisor is in favour of the work stating the interest and value of your methods development and therefore supportive of publication at The EMBO Journal.

We are thus pleased to inform you that we can offer to swiftly move forward towards acceptance of this work at The EMBO Journal as a methods resource.

We now need you to take care of a number of minor issues related to formatting and data annotation, which I will share shortly in a separate message, together with additional changes and requests by our production team and information regarding Source Data provision.

Please submit a revised version of the manuscript at your earliest convenience using the link enclosed below, addressing the advisors' comments.

As you might remember from previous experience at EMBO Press, every paper at our journals includes a 'Synopsis', displayed on the html and freely accessible to all readers. The synopsis includes a 'model' figure as well as 2-5 one-short-sentence bullet points that summarize the article. I would appreciate if you could provide this figure and the bullet points.

Thank you again for giving us the chance to consider your manuscript for The EMBO Journal, I look forward to hearing from you and receiving your final revised version of the manuscript.

Best regards,

Daniel Klimmeck

EMBOJ-2025-123014

Arbitrating advisor's comment:

I have read the revised manuscript as well as the first and second rebuttal by the authors. I agree with your assessment that this study does provide a valuable addition to the human PSC field as it provides an affordable and readily accessible method to grow naive human PSCs capable of in vitro differentiation and blastoid formation while maintaining a naive transcriptional program. I am impressed by the number of cell lines and serum batches the authors have tested as well as the number of independent laboratories that validated key results. This makes the manuscript in my view a compelling and impactful resource appropriate for EMBOJ and should be of broad interest to the field. Thus, I recommend moving forward with this manuscript.

Please remember: Digital image enhancement is acceptable practice, as long as it accurately represents the original data and conforms to community standards. If a figure has been subjected to significant electronic manipulation, this must be noted in the figure legend or in the 'Methods' section. The editors reserve the right to request original versions of figures and the original images that were used to assemble the figure.

The authors addressed the remaining editorial issues.

Dear Dr Martello,

Thank you for submitting the revised version of your Methods manuscript. I have now evaluated your amended manuscript and concluded that the remaining minor concerns have been sufficiently addressed.

I am thus pleased to inform you that your manuscript has been accepted for publication in the EMBO Journal.

Best regards,

Daniel Klimmeck

Daniel Klimmeck, PhD
Senior Editor
The EMBO Journal
EMBO
Postfach 1022-40
Meyerhofstrasse 1
D-69117 Heidelberg
contact@embojournal.org

Please note that it is The EMBO Journal policy for the transcript of the editorial process (containing referee reports and your response letters) to be published as an online supplement to each paper. If you should prefer removal of any referee-only figures included in the point-by-point response(s), e.g. because they may still be used for future publication or because they have been reproduced from published work by others, please do let us know immediately via response email.

More information is available here: <https://link.springer.com/partners/embo-press/editorial-policies#Peer%20review>